# GATA2 mitotic bookmarking is required for definitive haematopoiesis

Rita Silvério-Alves [1,2,3,4], Ilia Kurochkin [1,2], Anna Rydström [1],
Camila Vazquez Echegaray [1,2], Jakob Haider [1,2], Matthew Nicholls [5],
Christina Rode[5], Louise Thelaus[1,2], Aida Yifter Lindgren[1,2],
Alexandra Gabriela Ferreira [1,2,3,4], Rafael Brandão[6], Jonas Larsson [1,2],
Marella F. T. R. de Bruijn [5], Javier Martin-Gonzalez [6] &
Carlos-Filipe Pereira [1,2,3] ✉

In mitosis, most transcription factors detach from chromatin, but some are retained and bookmark genomic sites. Mitotic bookmarking has been implicated in lineage inheritance, pluripotency and reprogramming. However, the biological significance of this mechanism in vivo remains unclear. Here, we address mitotic retention of the hemogenic factors GATA2, GFI1B and FOS during haematopoietic specification. We show that GATA2 remains bound to chromatin throughout mitosis, in contrast to GFI1B and FOS, via C-terminal zinc finger-mediated DNA binding. GATA2 bookmarks a subset of its interphase targets that are co-enriched for RUNX1 and other regulators of definitive haematopoiesis. Remarkably, homozygous mice harbouring the cyclin B1 mitosis degradation domain upstream *Gata2* partially phenocopy knockout mice. Degradation of GATA2 at mitotic exit abolishes definitive haematopoiesis at aorta-gonad-mesonephros, placenta and foetal liver, but does not impair yolk sac haematopoiesis. Our findings implicate GATA2-mediated mitotic bookmarking as critical for definitive haematopoiesis and highlight a dependency on bookmarkers for lineage commitment.

Mitosis entails nuclear envelope breakdown and chromatin condensation, leading to RNA polymerase and transcription factor (TF) detachment from chromosomes, bringing transcription down to residual levels[1–3]. Nevertheless, lineage-specific transcriptional patterns must be re-established to preserve cell identity after each cell division. Recently, retention of TFs at mitotic chromatin has emerged as a novel mechanism to convey transcriptional memory from mother to daughter cells[4,5]. In addition to the ability to decorate mitotic chromatin, several TFs were shown to mark specific genomic sites, a mechanism termed "mitotic bookmarking"[6–8]. Mitotic retention and bookmarking have been investigated in differentiated cell-types[9–12], embryonic stem cells (ESCs)[13–16], and more recently in adult stem cells[17]. Depleting TFs at mitosis-to-G1 (M-G1) transition in cell cultures retarded the reactivation of bookmarked genes' transcription and impaired cell fate acquisition[9,11,12,14,16,18]. However, the importance of the mitotic bookmarking mechanism for in vivo lineage commitment during the development of a living organism remains to be addressed.

[1]Molecular Medicine and Gene Therapy, Lund Stem Cell Center, Lund University, BMC A12, 221 84 Lund, Sweden. [2]Wallenberg Centre for Molecular Medicine, Lund University, BMC A12, 221 84 Lund, Sweden. [3]CNC - Centre for Neuroscience and Cell Biology, University of Coimbra, Largo Marquês do Pombal, 3004-517 Coimbra, Portugal. [4]Doctoral Programme in Experimental Biology and Biomedicine, University of Coimbra, Largo Marquês do Pombal, 3004-517 Coimbra, Portugal. [5]MRC Molecular Haematology Unit, MRC Weatherall Institute of Molecular Medicine, Radcliffe Department of Medicine, University of Oxford, OX3 9DS Oxford, UK. [6]Core Facility for Transgenic Mice, Department of Experimental Medicine, University of Copenhagen, Blegdamsvej 3B, 2200 Copenhagen, Denmark. ✉e-mail: filipe.pereira@med.lu.se

Developmental haematopoiesis is a stepwise process, starting with a first wave in the yolk sac at E7.5, which generates primitive erythrocytes, macrophages and megakaryocytes, followed by a pro-definitive (second) wave of progenitors, including definitive erythro-myeloid progenitors (EMPs) and a third wave resulting in the specification of hematopoietic stem cells (HSCs)[19]. Definitive HSCs emerge in the dorsal aorta of the aorta–gonad–mesonephros (AGM) region and placenta at embryonic day (E) 10.5[20], through an endothelial-to-hematopoietic transition (EHT)[21,22]. Emergent HSCs migrate to the foetal liver and proliferate before colonising the bone marrow, where they remain mainly quiescent[20]. As cell cycle is tightly connected with the specification and maintenance of haematopoietic stem and progenitor cell (HSPC) fate, haematopoiesis provides an attractive biological system to address the role of TF-mediated mitotic bookmarking in vivo.

We have previously demonstrated direct reprogramming of mouse[23,24] and human fibroblasts[25,26] to HSPCs by overexpressing the TFs GATA2, GFI1B and FOS. These TFs induced a dynamic hemogenic process, recapitulating EHT and HSC ontogeny[24]. Here, we used fluorescence live-cell imaging combined with chromatin immunoprecipitation sequencing (ChIP-seq) to identify hemogenic factors with bookmarking ability. We then generated a mouse model to assess the role of GATA2-mediated bookmarking in vivo during hematopoietic commitment.

## Results

### Mitotic retention of hemogenic factors
To address whether hemogenic TFs display mitotic retention with the potential impact on the specification of definitive haematopoiesis in vivo (Fig. 1a), we analysed the subcellular localisation of GATA2, GFI1B and FOS after nocodazole arrest in transduced human dermal fibroblasts (HDFs) and HEK 293T cells (Fig. 1b, Supplementary Fig. 1a-c). GATA2 protein was detected in the nuclear fractions (soluble nucleus and chromatin-bound), in contrast to GFI1B, which was mainly present in the cytoplasmic and soluble nucleus fractions. FOS was almost exclusively present in the cytoplasmic fraction in both mitotic and asynchronous cells, indicating an overall weak binding to chromatin throughout the cell cycle[25]. These results suggest that mitotic retention is determined by the intrinsic DNA binding ability of each transcription factor. In addition, we overexpressed each TF fused to an mCherry fluorescent protein and addressed mitotic retention by live-cell imaging. GATA2 colocalised with chromatin during all phases of mitosis, while GFI1B was enriched at later stages (starting at anaphase) and FOS was completely excluded from chromatin in human fibroblasts (Fig. 1c, Supplementary Movies 1–3), mouse fibroblasts (Supplementary Fig. 1d) and K562 cells (Supplementary Fig. 1e). Positioning mCherry at the N- or C-terminal regions resulted in similar mitotic retention patterns (Supplementary Fig. 1f). Mitotic chromatin enrichment was significantly higher for GATA2 (Fig. 1d). Nevertheless, the stage-specific retention of GFI1B at anaphase highlights differential retention kinetics of mitotic bookmarking factors (Fig. 1c, Supplementary Fig. 1d, e). To exclude bias derived from overexpression, we verified mitotic retention of endogenous TFs in K562 cells (Fig. 1e, Supplementary Fig. 1g). As expected, the three TFs showed similar retention profiles, confirming robust GATA2 mitotic retention. Importantly, TF mitotic retention ability did not correlate with absolute mRNA or protein levels of endogenous or overexpressed TFs (Fig. 1f–h). Since GATA2, GFI1B and FOS cooperate during hemogenic reprogramming[25], we reasoned that co-expression could increase the retention of excluded factors. However, neither GFI1B nor FOS showed increased chromatin enrichment when co-expressed with GATA2 or with the two additional factors (Fig. 1i, j, Supplementary Fig. 1h). These results suggest that mitotic retention is an intrinsic feature of GATA2 that is not dependent on TF cooperation, protein abundance or cellular context.

### GATA2 mitotic retention requires DNA binding
We proceeded to dissect the protein domains required for GATA2 mitotic retention. The DNA-binding domain of GATA2 comprises an N-terminal zinc finger (N-ZF) and a C-terminal zinc finger (C-ZF) with homologous sequences, but different functions (Fig. 2a). The N-ZF has been implicated in stabilising DNA-protein complexes, whereas the C-ZF recognises and binds GATA consensus sequences[27]. To define domains required for mitotic retention, we generated mCherry-GATA2 deletion constructs lacking N-terminal, N-ZF, C-ZF and nuclear localisation signal (NLS). Interestingly, GATA2 was reduced from chromatin-bound protein fraction when the C-ZF, but not the N-ZF was deleted (Fig. 2b–d Supplementary Fig. 2a). The deletion of the NLS also led to the reduction of mitotic retention as assessed by imaging and subcellular fractionation followed by western blotting. This may reflect the requirement of active nuclear import, as previously described for SOX2[13]. However, since we detected GATA2 in the interphasic nucleus in the absence of the NLS (Fig. 2b), it is also possible that this deletion disturbs adjacent C-ZF functions. To confirm the requirement of C-ZF for mitotic retention, we selected GATA2 point mutations commonly found in leukemic and Emberger syndrome (ES) patients that influence DNA-binding affinity (Fig. 2a)[28–30]. C-ZF mutations associated with Acute Myeloid Leukaemia (AML) and/or ES that reduce DNA-binding affinity[28,30], including R396Q, R398W, T354M, R361L, and C373R, showed reduced GATA2 mitotic retention by fluorescent microscopy (Fig. 2e, Supplementary Fig. 2c–j, Supplementary Movie 4), suggesting that DNA-binding is necessary for GATA2 mitotic chromatin retention. L359V, which is described to increase the DNA-binding affinity of GATA2[31] and R362Q, which has a modest impact in binding affinity[28], did not display impaired mitotic retention (Fig. 2e, Supplementary Fig. 2e, g, Supplementary Movie 5). Quantification of chromatin-associated GATA2 by western blotting revealed decreased GATA2 chromatin binding in both mitotic and asynchronous cells for GATA2 mutations that have reduced DNA-binding affinity, particularly T354M, R361L and C737R (Fig. 2e, f, Supplementary Fig. 2b). In addition to DNA-binding perturbations, mutations in GATA2 may introduce complex conformational changes and modifications in protein stability, making it difficult to reveal mitosis-specific effects with this approach. Interestingly, both L359V and C373R mutants failed to efficiently reprogram HDFs to hemogenic cells (Fig. 2g), indicating that an intact C-ZF is critical for GATA2's reprogramming function and a different approach is necessary to uncover the role of GATA2 in mitosis. In agreement with its less important function for DNA binding, N-ZF mutations did not impact GATA2 mitotic retention (Supplementary Fig. 3). Taken together, these data suggest that GATA2 mitotic retention requires DNA-binding mediated by the C-ZF domain.

### GATA2 bookmarks key HSPC regulators
Next, we examined the genome-wide occupancy of endogenous GATA2 in asynchronous and fluorescence-activated cell sorting (FACS)-purified mitotic K562 cells using ChIP-seq. K562 cells were double-fixed (Supplementary Fig. 4a–d) prior to sorting to reduce potential artefacts caused by formaldehyde-only fixation[13,32]. Our analysis showed that GATA2 binds to a subset (1,598 peaks) of interphase target sites during mitosis, confirming bookmarking activity (Fig. 3a). The overall number of mitotic peaks was comparable to the pluripotency regulator ESRRB (1980 peaks)[15]. Bookmarked sites accounted for 15.3% of interphase genes including *GATA2* auto-regulation, the key regulator of definitive haematopoiesis *RUNX1*[33] and the HSC marker *CD9*[34] (Fig. 3b, Supplementary Fig. 4e, Supplementary Data 2). Since mitotic-unique peaks were scarce (42), had low read coverage, and were not located at genes with a known function in haematopoiesis, these were not included in further analyses. To investigate the differences in binding affinity of GATA2, we performed K-means clustering of asynchronous peaks, which resulted in three clusters (Fig. 3c). Most mitotic (bookmarked) peaks (71%) were assigned to

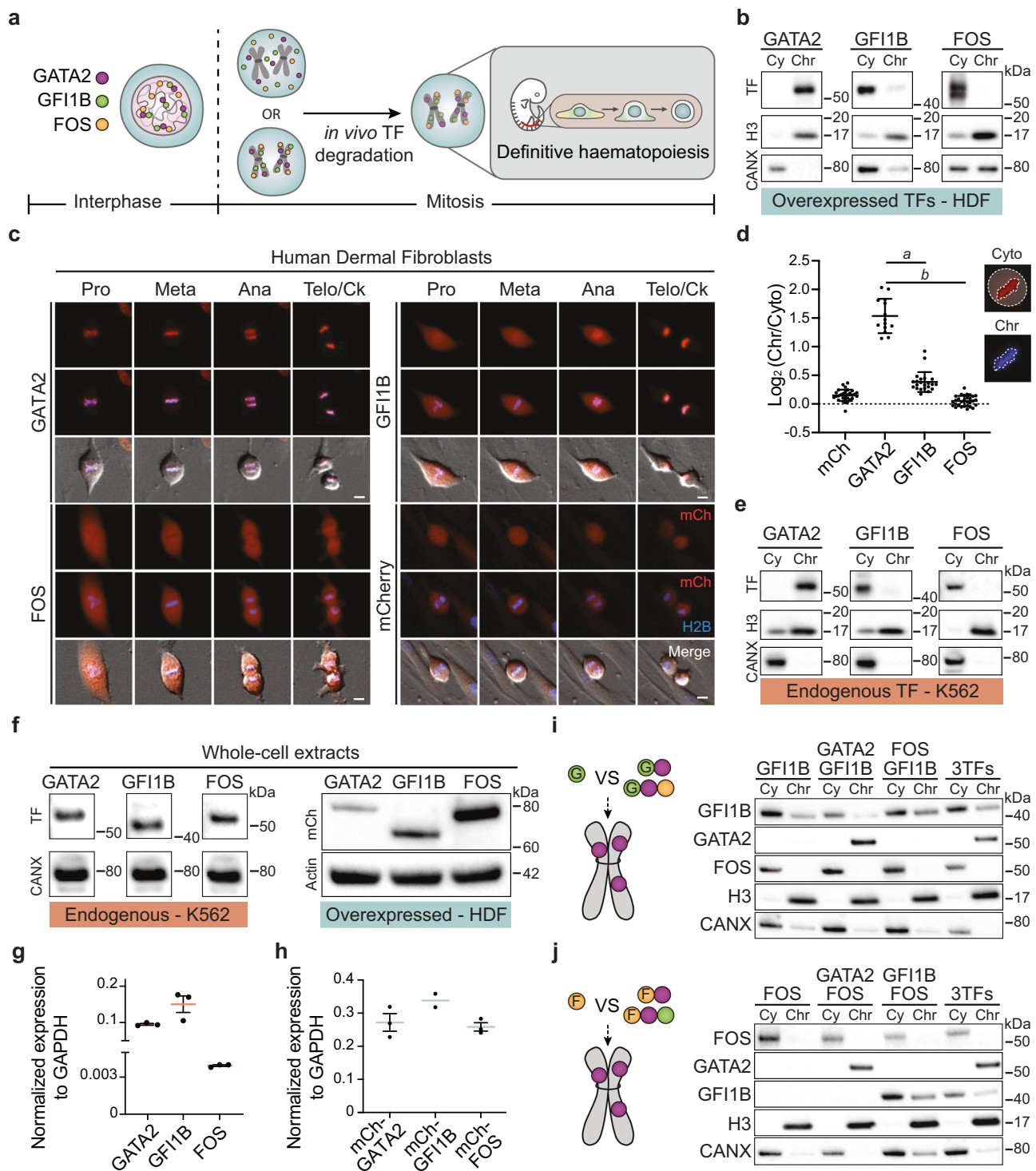

cluster 1, which contained a small fraction (9.5%) of GATA2 peaks in asynchronous cells with the highest peak intensities, suggesting that GATA2 bookmarks sites with high TF affinity. Cluster 3 accounted for only 3% of overlapped peaks (44 peaks) and, therefore, was excluded in further analyses. Next, we looked at the density of binding sites, by calculating the number of GATA2 binding motifs per peak in each group (Fig. 3d). Interestingly, mitotic peaks contained a significantly higher number of GATA2 motifs when compared to asynchronous peaks (KS test, $p$-value < 0.05). This observation implies that GATA2 mitotic bookmarking is influenced by a pre-existing motif structure, where increased number of binding sites translates into higher chromatin engagement in mitosis. We then performed

peak enrichment analysis based on asynchronous, mitotic, and mitotic clusters (Supplementary Fig. 4f, Supplementary Data 2). Gene Ontology Biological Processes showed co-enriched terms including "immune system development" ($p$-value = $2.4 \times 10^{-7}$, $1.2 \times 10^{-3}$ and $8 \times 10^{-4}$ in asynchronous, mitotic and cluster 1, respectively), "hematopoietic or lymphoid organ development" ($p$-value = $1.2 \times 10^{-6}$, $4.8 \times 10^{-3}$ and $3.4 \times 10^{-3}$) and "hemopoiesis" ($p$-value = $1.0 \times 10^{-5}$, $2.7 \times 10^{-2}$ and $2.7 \times 10^{-2}$). Moreover, motif enrichment analysis revealed that mitotic peaks were enriched for GATA, RUNX, and ETS motifs (Fig. 3e, Supplementary Fig. 4g), suggesting that GATA2 cooperates in mitosis with hematopoietic regulators. Therefore, we investigated TF cooperation in asynchronous and mitotic cells with other important

**Fig. 1 | GATA2 is retained at mitotic chromatin independently of cell context.**
**a** Experimental approach to address mitotic bookmarking function of the hemo-genic reprogramming transcription factors (TFs) GATA2, GFI1B and FOS, and their role in vivo for definitive haematopoiesis. **b** TF expression in the cytoplasmic (Cy) and chromatin-bound (Chr) protein fractions of mitotic human dermal fibroblasts (HDFs) expressing the indicated TF. Histone 3 (H3) and calnexin (CANX) were used as loading controls. kDA, kilodaltons. **c** Live-cell images of HDFs overexpressing mCherry (mCh)-TFs fusion proteins (red) during mitosis (Pro – prophase, Meta – metaphase, Telo/Ck – Telophase/Cytokinesis). Histone 2B (H2B)-mTurquoise (blue) signal shows DNA content. Mitotic events: n(GATA2) = 48, n(GFI1B) = 24, n(FOS) = 30, n(mCherry) = 137. Scale bars, 10 μm. **d** Quantification of the ratio between signal intensity of chromatin-retained and cytoplasm-localised TFs in metaphasic HDFs. n(mCherry)=30, n(GATA2) = 14, n(GFI1B) = 22, n(FOS) = 30. Mean ± SD is represented. Statistical significance was analysed by one-way ANOVA followed by Bonferroni's multiple comparison test. *a* and *b*, p < 0.001.

**e** Endogenous expression of GATA2, GFI1B and FOS in cytoplasmic and chromatin-bound fractions of mitotic K562 cells. H3 and CANX were used as loading controls. **f** Protein expression from asynchronous HDFs (overexpressed TFs, right panel) and K562 (endogenous TFs, left panel) whole-cell extracts. HDFs were transduced with mCh-TFs and the same number of mCh⁺ cells were FACS-purified for western blotting. CANX and Actin were used as loading controls. **g** Transcript levels of each endogenous TF in K562 cells. n = 3 biological independent samples per condition. Expression was normalised to GAPDH. **h** Transcript levels for overexpressed mCh-TFs in mCh⁺ FACS-purified HDFs. n(mCh-GATA2) = 3 technical replicates, n(mCh-GFI1B) = 2 technical replicates. (mCh-FOS) = 3 technical replicates. **g, h** Mean ± SD is represented. Expression was normalised to GAPDH. **i, j** Protein levels of GATA2, GFI1B and FOS in Cy and Chr fractions of mitotic HEK 293T cells when GFI1B (**i**) or FOS (**j**), were transduced individually or in combination with one or two additional TFs. Western blots were performed twice. Source data are provided as a Source Data file.

factors for haematopoiesis and HSPC specification, including the "heptad" TFs TAL1, LYL1, RUNX1, ERG, and FLI1[35], plus PU.1 (SPI1), MYB[35], PBX[36], GFI1B, FOS, HES1[37], MEIS1[35] and HLF[36] (Fig. 3f). We observed a high degree of motif co-occurrence at GATA2 mitotic peaks particularly for PU.1, "heptad" TFs, RUNX1 and FOS, implying that TF complexes and cooperativity with GATA2 is maintained during mitosis. This was confirmed by integration with available ChIP-seq data[38,39], which demonstrated a high degree of overlap between GATA2 mitotic peaks and the binding sites for RUNX1, PU.1, FOS, and TAL1 (Fig. 3g). We then inspected the genomic distribution of asynchronous and mitotic GATA2 peaks. Overall, GATA2 genomic distribution was retained in mitosis, with preference for promoters and active enhancers (Fig. 3h, Supplementary Fig. 4h). Interestingly, we observed a 1.8- and 2.2-fold binding decrease at "weak enhancer" (EnhWk, marked by H3K4me1) and "bivalent enhancer" (EnhBiv, marked by H3K4me1 and H3K27me3) chromatin states respectively, attributed to lower mitotic retention at sites decorated with H3K4me1 (Fig. 3i, j, Supplementary Data 2). Besides the potential biological role, the reduction of H3K4me1 also confirms that bookmarked sites are not derived from contamination with asynchronous cells. Furthermore, we integrated our data with histone marks and chromatin accessibility data (DNase-seq and ATAC-seq) available for K562 cells from ENCODE project[38] (Fig. 3i, j, Supplementary Data 2). In addition to H3K4me1, H3K36me3 and H4K20me1, which are associated with transcriptional elongation, also showed lower abundance at mitotic peaks[40]. Regarding accessibility to nucleases, we did not find major differences between mitotic and asynchronous peaks, especially when looking at cluster 1 mitotic peaks, suggesting that chromatin accessibility does not represent a barrier for mitotic bookmarking[13]. Altogether, our results show that GATA2 bookmarks critical regulators of HSPC specification and function.

## Bookmarking is critical in vivo

Next, we utilised the mitotic degradation (MD) domain of cyclin B1[11,14,16] to assess the role of GATA2-mediated mitotic bookmarking. Substitution of arginine for an alanine (R42A) inactivates the domain (MD$_{mut}$)[11] and results in similar protein levels throughout the cell cycle. We assessed protein levels of MD- or MD$_{mut}$-GATA2 constructs before and after release from nocodazole arrest, using degradation of cyclin B1 as a control for mitotic degradation. While cyclin B1 was quickly degraded after nocodazole release, we observed induced GATA2 protein degradation over time, with the highest impact on MD-GATA2 degradation at 4 hours, increased protein levels after 6 hours and returning to MD$_{mut}$-GATA2 levels in asynchronous cells (Fig. 4a, Supplementary Fig. 5a). For better resolution of protein degradation during specific phases of cell cycle, we fused the MD or MD$_{mut}$ domains to an mTurquoise fluorescent protein and observed mTurquoise signal reduction by flow cytometry at both mitosis and G1 phases (Supplementary Fig. 5b–d).

To gain initial insight regarding the role of GATA2 at M-G1 transition for HSPC generation, we induced hemogenic reprogramming in HDFs with MD-GATA2 or MD$_{mut}$-GATA2, in combination with GFI1B and FOS[25,26] (Fig. 4b). We observed that expression of the surface marker CD9, an early marker of hemogenic reprogramming[25,26] which is also bookmarked by GATA2 (Supplementary Fig. 4e), was delayed when GATA2 was degraded at M-G1 transition (Fig. 4c), highlighting the importance of GATA2 mitotic bookmarking for hemogenesis. To uncover the transcriptional impact of degrading GATA2 at M-G1 transition, we employed single-cell RNA-sequencing (scRNA-seq) at early stages of hemogenic reprogramming. We profiled 32,773 cells derived from untransduced HDFs (two independent donors), and 4 and 6 days after induction of hemogenic reprogramming. We first confirmed that cells derived from the two donors undergo similar transcriptional changes using Uniform Manifold Approximation and Projection (UMAP) (Supplementary Fig. 5e–g) and in agreement with flow cytometry data, *CD9* expression was delayed when reprogramming was induced with MD-GATA2 (Supplementary Fig. 5h). To investigate the differences between the transcriptional program in MD-GATA2 and MD$_{mut}$-GATA2 reprogrammed cells, we performed differential gene expression analysis (Supplementary Data 3) focusing on GATA2 bookmarked targets (Fig. 4d). This analysis revealed that 28 and 75 genes were downregulated, and 18 and 37 genes were upregulated at day 4 and 6 respectively, when GATA2 was degraded at mitotic exit, pointing to a role of GATA2 at M-G1 in the activation of the hemato-poietic program during reprogramming. Downregulated genes with MD-GATA2 included *ZBTB16*, an epigenetic regulator of HSC fate[41] and *TEAD1* which helps driving hematopoietic specification[42]. Reassuringly, genes that require GATA2 at mitotic exit during reprogramming were predicted to be regulated by GATA2 with ChIP Enrichment Analysis (ChEA, Supplementary Data 3), and contained GATA2 motifs, as well as motifs of other hematopoietic TFs (Fig. 4e). We observed high enrichment with the "heptad" TFs members, especially at day 6, suggesting their cooperative action also in mitosis to impose hemogenic fate.

To address the role of GATA2 bookmarking in vivo, we generated a mouse model where the MD domain was inserted upstream the *Gata2* gene via CRISPR-Cas9 (Fig. 4f, Supplementary Fig. 6a). The insertion of MD in both *Gata2* alleles resulted in lethality, as homozygous pups could not be generated from two independent injections of edited ESCs (Supplementary Fig. 6b) or by crossing heterozygous mice (Fig. 4g). Remarkably, MD homozygous mice died at the onset of definitive haematopoiesis, between E10.5 and E11.5 (Fig. 4g, h, Supplementary Fig. 6c), phenocopying *Gata2* knockout mice[43]. When we compared the expected Mendelian ratios, we found higher frequencies of wild-type (WT) mice at the expense of heterozygous mice, particularly at E10.5 (observed 44% vs expected 25%), E13.5 (observed 45%) and postnatally (observed 49%) (Fig. 4g), suggesting a more profound impact in MD-*Gata2* mice when compared to *Gata2* full

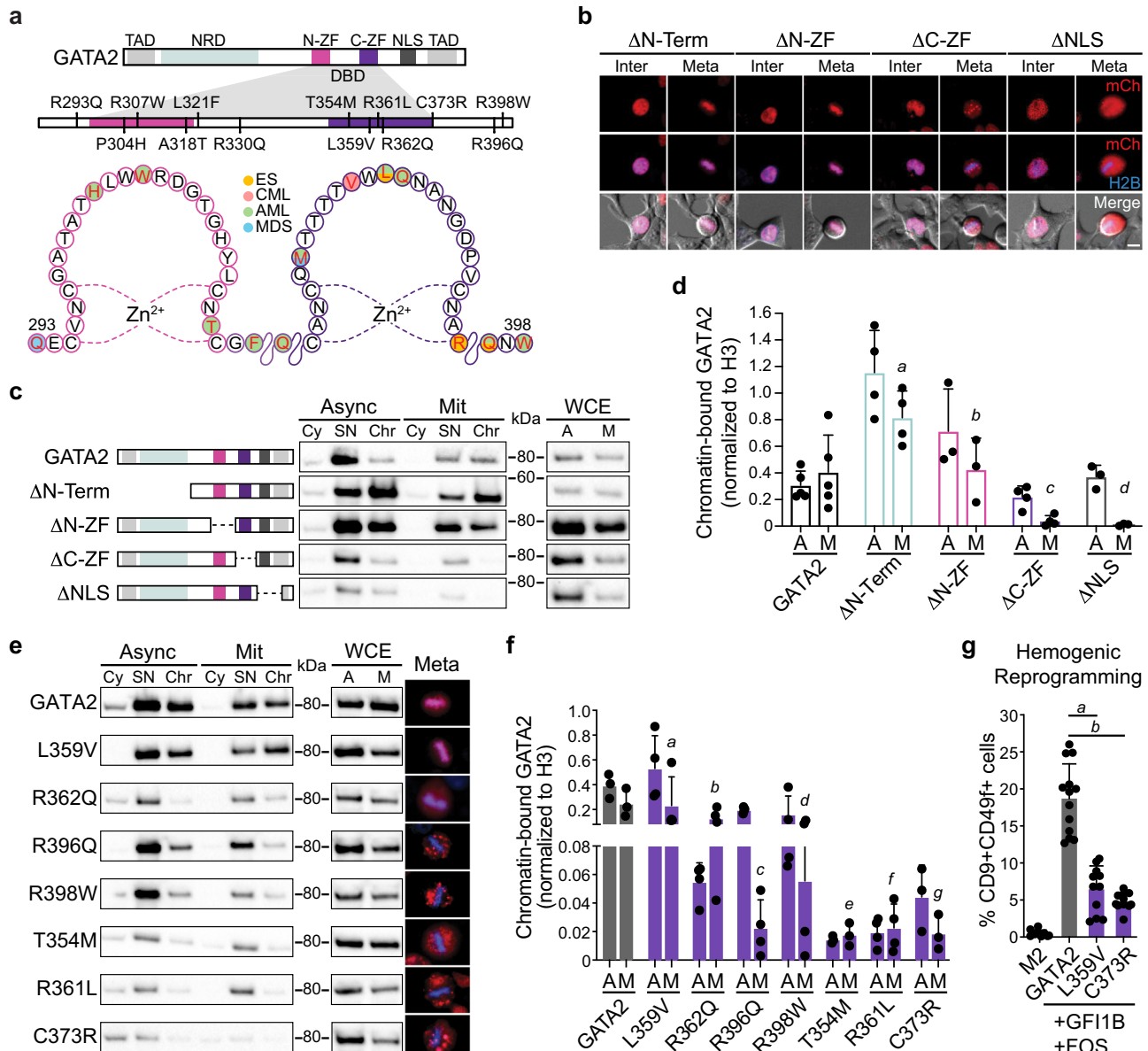

**Fig. 2 | Point mutations in GATA2 C-terminal zinc finger associated with leukaemia and Emberger syndrome reduce mitotic retention. a** Representation of GATA2 domains highlighting leukaemia and Emberger syndrome (ES)-associated point mutations in the N- and C- terminal zinc fingers (ZF) of DNA-binding domain (DBD). TAD – transactivation domain. NRD – negative regulatory domain. NLS – nuclear localisation signal. CML – chronic myeloid leukaemia. AML – acute myeloid leukaemia. MDS – myelodysplastic syndrome. **b** Live-cell images of 293T cells overexpressing mCherry (mCh)-GATA2 deletion (Δ) constructs (red) excluding the N-terminal (amino acids 1-235), N-ZF (287–342), C-ZF (243–379) or NLS (380–440) in interphase (Inter) and metaphase (Meta). Mitotic events: n(ΔN-Terminal) = 395, n(ΔN-ZF) = 375, n(ΔC-ZF) = 100, n(ΔNLS) = 52. Scale bar, 10 μm. **c**, Protein expression in whole-cell extracts (WCE), cytoplasmic (Cy), soluble nucleus (SN) and chromatin-bound (Chr) fractions of both asynchronous (A, Async) and mitotic (M, Mit) 293T cells overexpressing deletion constructs. Representative blots were acquired with the same exposure time for asynchronous and mitotic cells. kDA, kilodaltons. **d**, Western blot quantifications of GATA2 deletions at Chr fraction in asynchronous and mitotic cells after normalising to H3. n(GATA2) = 5, n(ΔN-Term)

= 4, n(ΔN-ZF) = 3, n(ΔC-ZF) = 4, n(ΔNLS) = 3. Statistical analysis performed by two-way ANOVA followed by Fisher's LSD test and comparisons of mitotic retention of each deletion to GATA2 are shown. *a, p* = 0.005; *b, p* = 0.89, *c* and *d, p* = 0.01. **e** Protein expression of mCherry-fused GATA2 mutants in the Cy, SN and Chr protein fractions of asynchronous and mitotic 293T. Representative cells in metaphase are shown (right). Histone 2B (H2B)-mTurquoise signal (blue) indicates DNA content. Scale bar, 10 μm. **f** Quantification of GATA2 mutants at Chr fraction. n(GATA2) = 3, n(L359V) = 4, n(R362Q) = 4, n(R396Q) = 4, n(R398W) = 4, n(T354M) = 3, n(R361L) = 4, n(C373R) = 3. Statistical analysis performed by two-way ANOVA followed by Fisher's LSD test and comparisons of mitotic retention of each mutant to GATA2 are shown. *a, p* = 0.86; *b, p* = 0.17, *c* and *f, p* = 0.01; *d, p* = 0.03; *e* and *g, p* = 0.02. **g** Hemogenic reprogramming efficiency (% CD9⁺CD49f⁺ cells) with GATA2, L359V or C373R, plus GFI1B and FOS. M2rtTA (M2) was used as control. n(M2) = 10, n(GATA2) = 13, n(L359V) = 11, n(C373R) = 11. Mean ± SD is represented. Statistical significance was analysed by one-way ANOVA followed by Bonferroni's test. *a* and *b, p* < 0.001. Source data are provided as Source Data file.

knockout model which showed near-Mendelian ratios (29%)[43]. This might be explained by an allelic bias in MD-*Gata2* heterozygous mice resulting in preferential expression of MD over the WT allele (Supplementary Fig. 6d). Morphological analysis of E10.5 and E11.5 MD-*Gata2* embryos showed that MD/MD embryos were smaller and paler,

particularly at E11.5, with evident lack of blood (Fig. 4h, Supplementary Fig. 6e). This contrasted with insertion of MD_mut upstream the *Gata2* gene, which did not hamper embryonic development (Fig. 4h, Supplementary Fig. 6f, g). As an additional control, we have confirmed that the MD-GATA2 protein was expressed in vivo by western blotting on

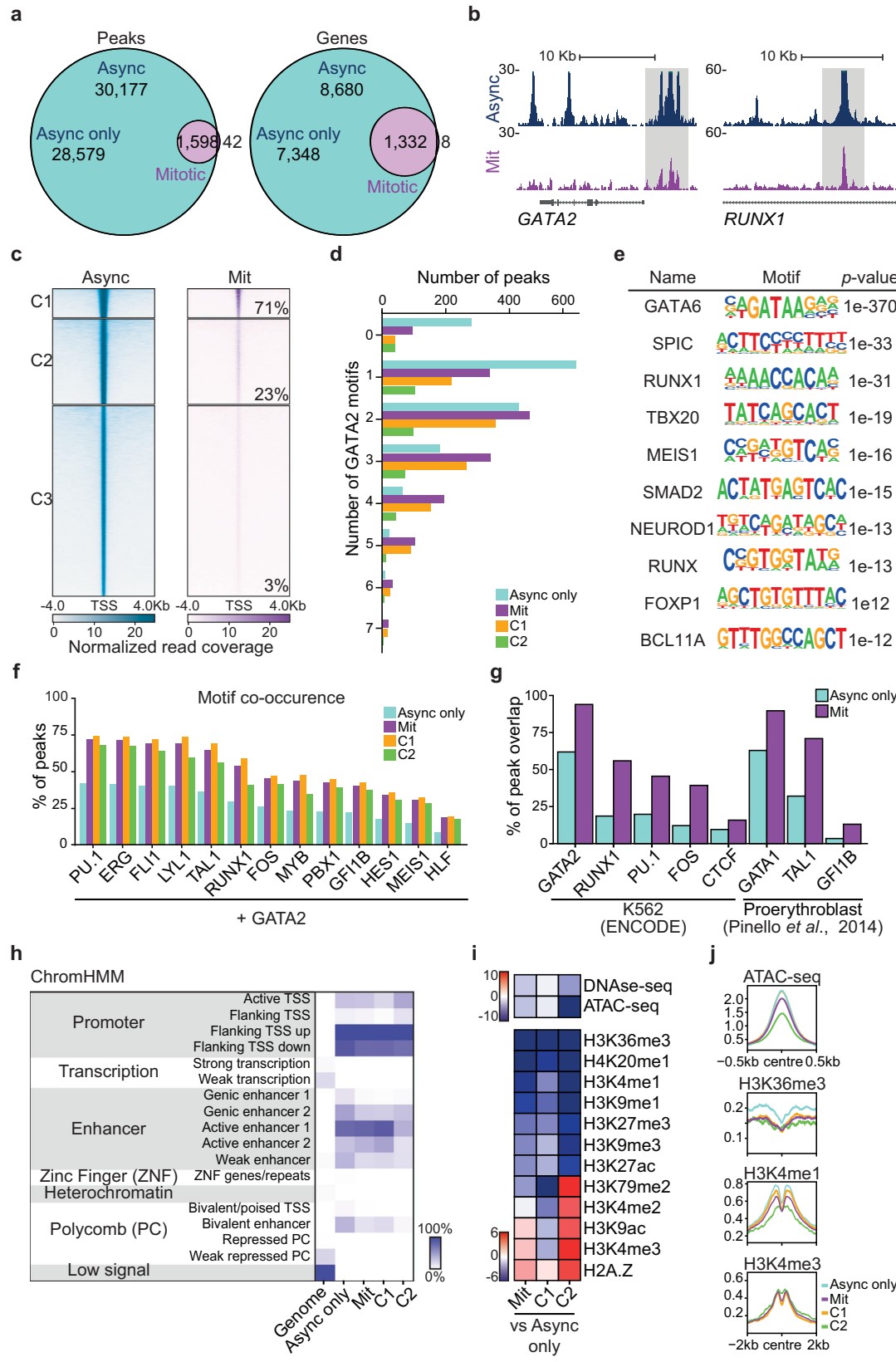

heterozygous E9.5 yolk sac-derived hematopoietic colonies (Supplementary Fig. 6h). Flow cytometry analysis of MD-*Gata2* embryonic erythroblasts showed premature erythroid maturation at E10.5 (Fig. 4i, Supplementary Fig. 6i), resulting in severe anaemia at E11.5 (Fig. 4j). Our results illustrate the critical importance of GATA2 at M-G1 transition for embryonic development.

**Definitive haematopoiesis requires bookmarking**

Definitive HSPCs emerge in intra-aortic hematopoietic clusters (IAHCs) positive for RUNX1 and the endothelial marker CD31[33,44]. To assess the impact of GATA2 degradation at M-G1 transition in definitive haematopoiesis, we first analysed embryos and cluster formation by whole-embryo mounting followed by immunohistochemistry. We observed

**Fig. 3 | GATA2 bookmarks a subset of interphasic target genes with roles in definitive haematopoiesis. a** Venn diagram showing the number of ChIP-seq GATA2 peaks and genes shared between asynchronous (Async) and mitotic K562 cells. Async only refers to non-bookmarked peaks and genes in asynchronous cells. **b** Gene tracks for GATA2 binding sites at *GATA2* and *RUNX1 loci* showing both mitotic (Mit) bookmarked (grey) and asynchronous unique peaks. Kb – kilobase. **c** K-means clustering of Async (left) and Mit peaks (right). The percentage of mitotic (bookmarked) peaks overlapping with asynchronous peaks in each cluster is shown. The 42 mitotic-unique peaks are not shown. **d** Number of GATA2 motifs in Async only peaks, mitotic peaks and mitotic clusters 1 (C1) and 2 (C2). **e** De novo motif enrichment analysis for GATA2 mitotic bookmarked target sites. Top ten

motifs are shown with respective *p*-values. **f** Percentage of GATA2 peaks where motifs for relevant HSPCs regulators are also present. **g** Percentage of overlap between GATA2 peaks and peaks for HSPC regulators from available ChIP-seq datasets. **h** Chromatin-state enrichment heatmap representing the percentage of genome occupancy of GATA2 per group of peaks. Scale represents the percentage of peaks at each genomic segment. TSS – Transcription start site. **i** Integration heatmap with histones marks, DNAse-seq and ATAC-seq data for K562 cells (ENCODE). Scale represents the accumulated sum differences across bins between Async only and mitotic peaks and clusters. **j** Histone marks and ATAC-seq profiles at peak summit (centre). Source data are provided as Source Data file.

an abolishment of hematopoietic clusters in the ventral region of the dorsal aorta and severe reduction at the dorsal region in MD homozygous embryos at E10.5 (Fig. 5a, b). Also, although the number of wild-type and heterozygous clusters was similar, heterozygous clusters were overall smaller. Interestingly, at E9.5, RUNX1 was unchanged, particularly in the vitelline artery where expression is stronger at this developmental stage[33] (Supplementary Fig. 7a). To assess HSPC function, we performed colony-forming unit (CFU) assays. At E9.5 the number of colonies obtained from yolk sacs were comparable between genotypes (Fig. 5c), suggesting that yolk sac progenitors are not affected by the loss of GATA2 at M-G1 transition. Indeed, the percentage of CD41+c-kit+Ter119- s in the yolk sac was similar between groups (WT/WT, 5.40 ± 1.43%; MD/WT, 3.88 ± 1.00%; MD/MD, 4.08 ± 0.45%) (Supplementary Fig. 7b). This result comes in contrast to *Gata2* full or conditional knockout models where both AGM and yolk sac haematopoiesis were shown to be impaired[43,45]. When we investigated E10.5 embryos, we observed a decrease in the number of hematopoietic colonies derived from AGM, placenta and foetal liver by 2.9-, 4.6- and 3.2-fold, respectively, when compared to WT (Fig. 5d). This effect was even more striking at E11.5, with colony numbers reduced by 16- and 8-fold in AGM and placenta, respectively, when compared to MD$_{mut}$ homozygous mice (Fig. 5e). These results reflect the lack of RUNX1+CD31+ hematopoietic clusters at E10.5 and illustrate the requirement of GATA2 at mitotic exit for definitive haematopoiesis. We then investigated the generation of HSPCs by transplantation of E11.5 placenta cells into sublethal irradiated CD45.1 or CD45.1/2 recipient mice (Supplementary Fig. 8a). While wild-type and heterozygous mice engrafted irradiated recipients, placental HSPCs from MD homozygous mice did not show long-term engraftment (6 months) in peripheral blood or bone marrow (Fig. 5f–h, Supplementary Fig. 8b). Short-term engraftment (4 weeks) was observed in 2 out of 8 mice (4.1% and 4.4% CD45.2+ cells), but these cells did not contribute to either myeloid or lymphoid lineages (Supplementary Fig. 8c). Since we could not observe significant differences in IAHC numbers or in embryonic HSPC function between wild-type and heterozygous mice, we evaluated adult heterozygous HSC function. Therefore, we performed competitive transplantation with bone marrow LSK-SLAM HSCs (Lineage-Kit+Sca-1+CD150+CD48-) from adult mice (Supplementary Fig. 8d, e). GATA2 haploinsufficiency (heterozygous from full *Gata2* knockout) translates into reduced engraftment of bone marrow HSCs[46]. Likewise, we observed reduced engraftment capacity of MD-*Gata2* heterozygous HSCs from adult bone marrow (Fig. 5i–k), suggesting a role for GATA2 mitotic bookmarking not only in HSC specification but also in HSC maintenance. Altogether, these results demonstrate that GATA2 is essential in vivo at M-G1 transition for definitive haematopoiesis. We propose that GATA2 remains bound to key hematopoietic genes during mitosis through its C-ZF domain, cooperating with HSPC regulators to allow faithful commitment of definitive HSPCs during embryonic development (Fig. 5l).

## Discussion

We show the hemogenic TF GATA2 remains bound to chromatin throughout mitosis, independently of cellular context or absolute

protein levels, and bookmarks important HSPC regulators. Abolishing GATA2 during M-G1 transition arrests embryonic development at the onset of definitive haematopoiesis. Unexpectedly, we also found that GFI1B was retained in a mitotic phase-specific manner starting from anaphase. The ability to bind to mitotic chromatin has been mainly associated with non-specific binding mediated by electrostatic interactions between TFs and DNA[12,17,47], where amino acid polarity and absolute charge per DNA-binding domain were predicted to play a major role[42]. While differences in electrostatic forces might explain the differences in enrichment capabilities of TFs, the kinetics of this process requires further exploration. The pro-neural TF ASCL1 was shown to decorate mitotic chromatin in late telophase[17], due to nuclear import at the time of nuclear envelope reassembling. The association of TFs at anaphase, as we report for GFI1B, illustrates stage-specific mitotic chromatin retention and may be controlled by independent mechanisms.

GATA2 binds DNA through its zinc finger domains. Removal of the C-terminal zinc finger (C-ZF) of the DNA-binding domain reduced mitotic retention of GATA2, while deletion of the N-terminal region or the N-terminal zinc finger (N-ZF) of the DNA binding domain did not. This may be due to the different functions of the zinc fingers[27], but nonetheless suggests that mitotic retention requires DNA binding, in agreement with previous studies[13,15,17]. We used mutations commonly found in patients with leukaemia, myelodysplastic syndrome (MDS) and ES, that were reported to interfere with DNA-binding affinity of GATA2, to implicate the requirement for the C-ZF, but not the N-ZF for mitotic retention. Patients with N-ZF mutations have better clinical outcomes, than patients with C-ZF mutations[48] and N-ZF mutants reduce, but do not abolish, chromatin occupation and transcriptional activation by GATA2[29]. C-ZF mutations impact GATA2 DNA-binding affinity at different levels: L359V (CML) shows increased affinity[31], R362Q (AML) has modest to no impact in binding affinity, T354M and R398W (AML/MDS) show moderate reduction in affinity to DNA and the ES mutations R361L, C373R and R396Q result in little to no DNA binding, as evaluated by EMSA[28,30]. However, the nature of these interactions is not completely clear (specific versus non-specific binding), as C-ZF and contiguous residues of GATA proteins form both direct hydrogen bonds and Van der Waals contacts with DNA bases[49]. Of note, NLS deletion (amino acids 380-440) includes the R396 and R398 residues, which may explain the impact observed in mitotic retention. Although GATA2 mitotic retention mediated by DNA binding is probably regulated by electrostatic forces, as described for many TFs[12,17,47], sequence-specific interaction cannot be discarded, particularly since we show that GATA2 binds to a subset of its interphase targets in mitosis, making it a bona fide mitotic bookmarking factor. Our findings underscore a potential unappreciated role of disruption of mitotic retention in human disease.

Among GATA2 bookmarked genes, we found several members of the "heptad" TFs (*ERG*, *GATA2*, *LMO2*, *LYL1*, *RUNX1*) that regulate gene expression and function in HSPCs[35]. Indeed, GATA2-bound sites are enriched for GATA, PU.1 and RUNX1 motifs, suggesting cooperation in mitosis between GATA2 and other key hematopoietic regulators. Bookmarked sites showed biologically

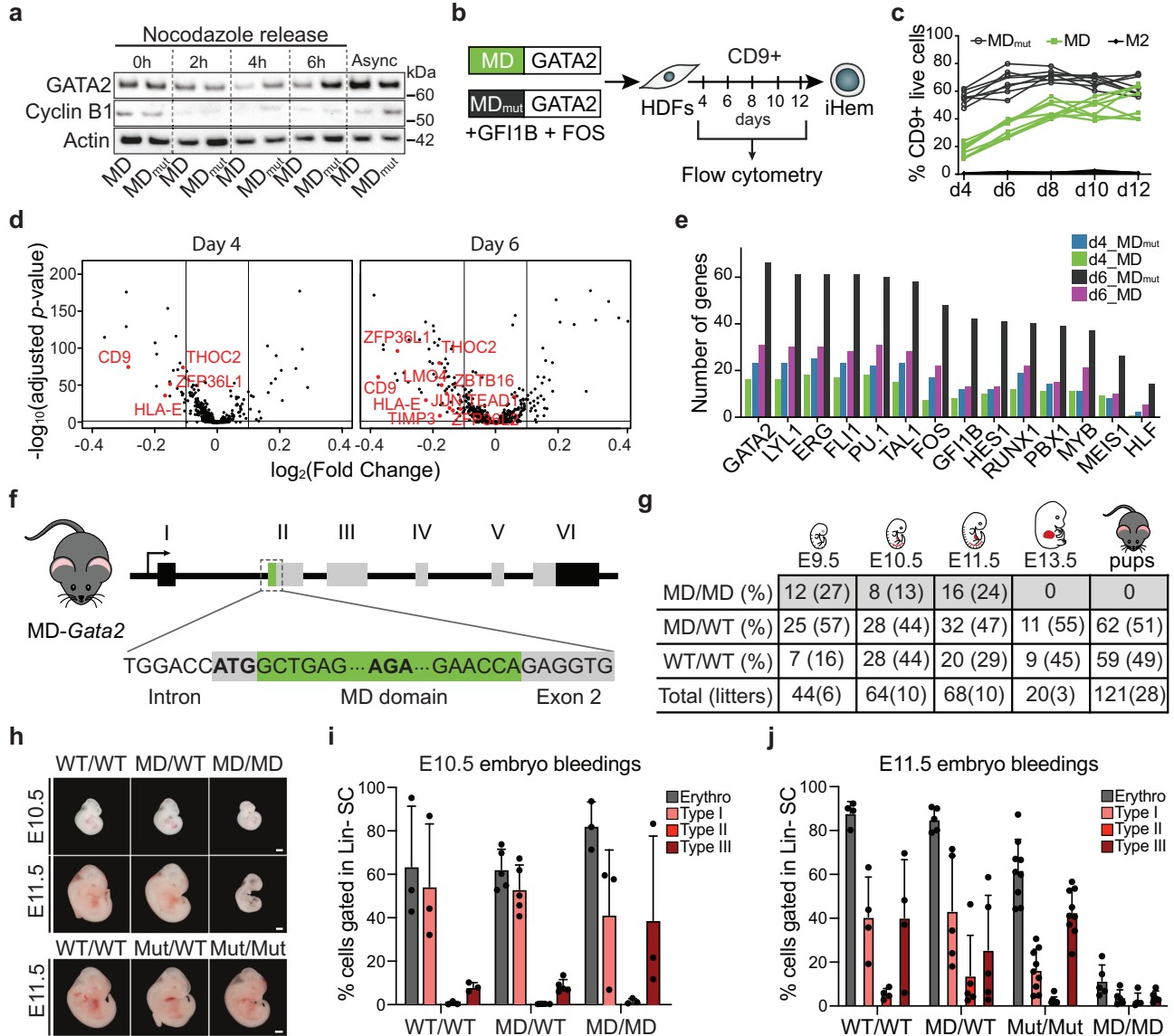

**Fig. 4 | GATA2 is required at the mitosis-to-G1 transition in vivo for embryonic development. a** Western blot analysis of GATA2 and cyclin B1 proteins in HEK 293T expressing GATA2 fused to the mitotic degradation (MD) domain of cyclin B1 or an inactive form (MD$_{mut}$) before (0 h) and 2, 4 and 6 hours (h) after release from nocodazole arrest. Actin shown as loading control. Async – Asynchronous cells. Western blots were performed twice. kDA, kilodaltons. **b** Direct reprogramming strategy to convert human dermal fibroblast (HDFs) into induced hemogenic cells (iHem). HDFs were transduced with lentivirus encoding MD- or MD$_{mut}$-GATA2, plus GFI1B and FOS factors, and the kinetics of CD9 activation was evaluated by flow cytometry. **c** Quantification of CD9 expression from day (d) 4 to d12. M2rtTA (M2) was used as control. **d** Volcano plots showing differential gene expression of GATA2 bookmarked genes at day 4 and 6 of reprogramming with MD-GATA2 and MD$_{mut}$-GATA2. Relevant genes downregulated in MD-GATA2 condition (left) are highlighted in red. **e** Number of bookmarked differentially expressed genes between MD-GATA2 and MD$_{mut}$-GATA2 conditions at d4 and d6, with binding sites

for HSPC regulators. Upregulated genes in MD$_{mut}$-GATA2 condition at d4 and d6 (d4_MD$_{mut}$ and d6_MD$_{mut}$, respectively), and upregulated genes in MD-GATA2 at d4 and d6 (d4_MD and d6_MD, respectively) are shown. **f** Schematic representation of the mouse model developed to assess mitotic degradation of GATA2 in vivo by inserting the MD domain upstream *Gata2* gene. **g** Frequency of homozygous (MD/MD), heterozygous (MD/WT) and wild-type (WT/WT) embryos at embryonic day (E) 9.5, E10.5, E11.5, E13.5 and pups, after crossing heterozygous mice. **h** Representative images of MD-*Gata2* embryos at E10.5 and E11.5, and control MD$_{mut}$-*Gata2* embryos at E11.5. Scale bars, 1 mm. **i, j** Flow cytometry quantification of E10.5 (**i**) and E11.5 (**j**) erythroblasts after whole-embryo bleeding. Graphs show percentage of total erythroblasts (Erythro) or immature (type I) to mature (Type III) erythroblasts gated within lineage negative (Lin-) live single cell (SC) population. n(E10.5 WT/WT) = 3, n(E10.5 MD/WT) = 5, n(E10.5 MD/MD) = 3 n(E11.5 WT/WT) = 4, n(E11.5 MD/WT) = 5, n(E11.5 Mut/Mut) = 9, n(E11.5 MD/MD) = 5. Mean ± SD is shown. Source data are provided as Source Data file.

relevant Gene Ontology terms for hematopoietic lineage development. Accordingly, TF co-occupancy analysis shows motifs with high co-occurrence at GATA2 peaks, including PU.1 and RUNX1 among other "heptad" TFs, pointing towards the maintenance of cooperative events in mitosis. Importantly, a gradual increase in RUNX1 and PU.1 expression together with a downregulation of endothelial genes was observed in hemogenic endothelial cells of the AGM, suggesting a role for PU.1 in

hematopoietic fate specification, in addition to the well-described role in controlling myeloid/erythroid lineage differentiation[50]. Moreover, the cooperative action between GATA2 and RUNX1 is of fundamental importance for the specification of HSCs, as supported by the phenotype of Gata2$^{+/-}$:Runx1$^{+/-}$ double heterozygous mice[35].

In mitosis, we observed a reduction of GATA2 at sites marked with H3K4me1 associated with enhancers, as well as H3K36me3 and

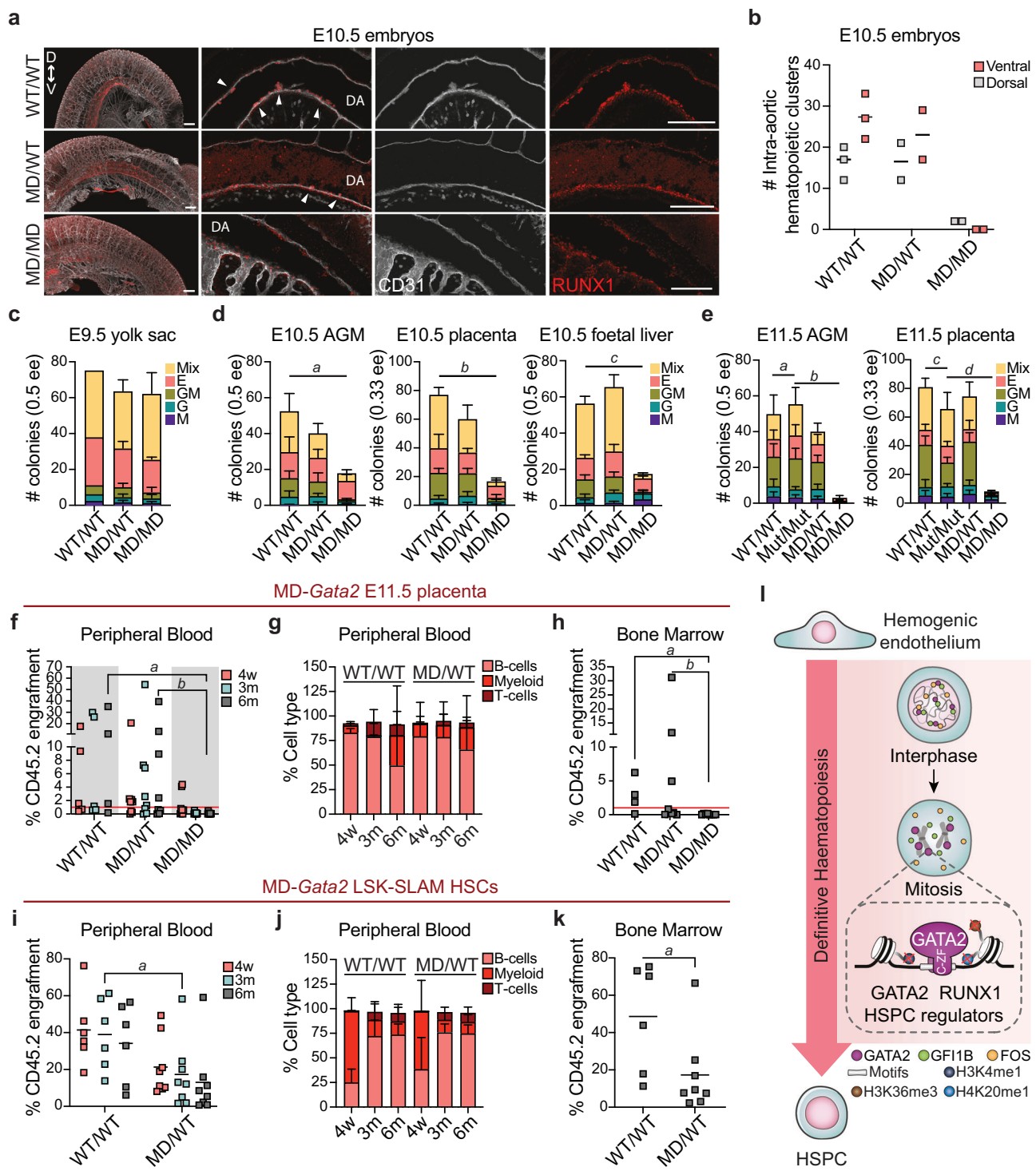

H4K20me1, associated with transcriptional elongation. The reduction in histone marks associated with elongation may reflect an overall reduction in transcription in mitosis. As for H3K4me1, decreased binding at bivalent and weak enhancers, while increasing retention at active enhancer and promoter sites might allow gene expression flexibility upon G1 re-entrance, while maintaining important active genes marked. This supports the idea that M-G1 transition might serve as a time-window for adjustments in future gene expression profiles[51,52]. Moreover, we showed that mitotic bookmarking is correlated with binding affinity to peaks with higher density of GATA2 motifs. Whether this pre-existing motif structure represents a transversal feature of other TF bookmarkers needs to be addressed in future studies.

*Gata2*-null mice die between E10.5 and E11.5, before the emergence of definitive HSCs, from severe anaemia[43]. E9.5 GATA2[-/-] embryos had primitive erythroid cells in circulation, but impaired yolk sac haematopoiesis. Definitive EMPs arise in the yolk sac from hemogenic endothelium between E8.25 and E10, before homing the foetal liver at E10.5, from where they sustain embryonic haematopoiesis, including definitive erythropoiesis[53]. Our data show embryo lethality similarly to the *Gata2* knockout model, however, MD-*Gata2* homozygous E9.5 embryos were indistinguishable from their wild-type counterparts, and yolk sac progenitors generated hematopoietic colonies in numbers comparable to those of wild-type and heterozygous yolk sacs. Moreover, yolk sac suspensions expressed similar

**Fig. 5 | GATA2 bookmarking is required for definitive haematopoiesis.**
**a** Immunohistochemistry images representing E10.5 wild-type (WT/WT), hetero-zygous (MD/WT) and homozygous (MD/MD) intra-aortic hematopoietic clusters expressing RUNX1 (red) and CD31 (white) in the ventral (V) or dorsal (D) sides of the dorsal aorta (DA). White arrowheads indicate clusters. Scale bars, 150 μm.
**b** Number (#) of intra-aortic hematopoietic clusters per genotype. n(WT/WT) = 3, n(MD/WT) = 2, n(MD/MD) = 2 embryos. Mean is shown. **c–e** Colony-forming units for E9.5 yolk sac (**c**), for E10.5 aorta-gonad-mesonephros (AGM), placenta and foetal liver (**d**) and for E11.5 AGM and placenta (**e**) cell suspensions. n(WT/WT) = 2-5, n(MD/WT) = 3–4, n(MD/MD) = 3–5, n(Mut/Mut) = 5 embryos. Mean ± SD is represented. Macrophage (M), granulocyte (G), granulocyte/macrophage (GM), erythroid (E) and mixed colonies (Mix) are shown per embryo equivalent (ee).
**d, e** Statistical significance for the total number of colonies was calculated by one-way ANOVA followed by Bonferroni's multiple comparison test. **d**, *a*, $p = 0.01$; *b*, $p < 0.001$, *c*, $p = 0.002$. **e**, *a*, *b*, *c* and *d*, $p < 0.001$. **f**, Percentage of donor chimerism (CD45.2$^+$) 4 weeks (w), and 3 and 6 months (m) after transplantation with E11.5 placenta cells. Red line indicates 1% chimerism. n(WT/WT, 4w, 3m) = 5, n(WT/WT,

6m)=4, n(MD/WT, 4w, 3m, 6m) = 8, n(MD/MD, 4w, 3m, 6m) = 8. *a*, $p = 0.011$; *b*, $p = 0.016$. **g**, Quantification of donor contribution to myeloid and lymphoid lineages. Only mice with donor chimerism above 1% were considered. n(WT/WT, 4w, 3m, 6m) = 3, n(WT/WT, 6m) = 4, n(MD/WT, 4w, 3m) = 5, n(MD/WT, 6m) = 4. **h** Percentage of donor chimerism in bone marrow 6 months after transplantation. n(WT/WT) = 4, n(MD/WT) = 8, n(MD/MD) = 8. *a*, $p = 0.023$; *b*, $p = 0.028$. **f, h** Statistical significance at 6 months was calculated with Kruskal-Wallis test followed by uncorrected Dunn's test. **i** Percentage of donor chimerism 4 weeks, 3 months and 6 months after competitive transplantation (see Supplementary Fig. 7, Methods). n(WT/WT) = 6, n(MD/WT) = 8. *a*, $p = 0.029$. **j** Quantification of donor contribution to blood lineages. n(WT/WT, 4w, 3m, 6m) = 6, n(MD/WT, 4w, 3m, 6m) = 8. **k** Percentage of donor chimerism in bone marrow 6 m after trans-plantation. n(WT/WT) = 6, n(MD/WT) = 8. *a*, $p = 0.020$. **i, k** Statistical significance was calculated with two-tailed Mann–Whitney test at 3 months (**i**) or at 6 months (**k**). **l** Proposed model for the role of GATA2 bookmarking for definitive haema-topoiesis. HSPC, hematopoietic stem and progenitor cell. Source data are pro-vided as Source Data file.

levels of EMP markers. Nevertheless, these progenitors did not con-tribute to blood output between E10.5 and E11.5, consequently leading to anaemia. This suggests that GATA2 might not be necessary at M-G1 transition for the generation of yolk sac progenitors, but an impact in their maintenance or function cannot be disregarded. Additionally, HSPC generation was severely reduced in the embryo proper at E10.5 and E11.5, suggesting a requirement of GATA2-mediated bookmarking during this developmental stage. Even though definitive EMPs and HSPCs arise through an EHT, hemogenic endothelia in the yolk sac and in the AGM region (and major arteries) are molecularly distinct and governed by different pathways[54]. For example, Notch control of *Gata2* expression is required for the generation of hematopoietic clusters in the AGM, but not for definitive yolk sac haematopoiesis[37,55]. In agreement, we found the Notch1 target *HES1* and the Notch ligand *JAG1*, both expressed in the AGM region at E10.5[37] to be bookmarked by GATA2, providing a direct link between GATA2 mitotic bookmarking and HSC specification via Notch pathway regulation. GATA2 haploinsufficient mice (*Gata2*$^{+/-}$) show a 2.7- and 1.6-fold decrease in the number of CFU colonies derived from E10 AGM and YS, respectively[45]. Moreover, *Gata2*$^{+/-}$ bone marrow HSPCs showed reduced engraftment capacity[56]. In contrast, embryonic MD-GATA2$^{+/-}$ HSPCs generated similar num-bers of CFUs when compared to wild-type HSPCs, and had similar numbers of IAHCs. These could be explained by either a differential requirement for mitotic bookmarking of blood progenitors through-out development or different cell cycle dependencies. Nevertheless, engraftment of irradiated recipients by bone marrow heterozygous LSK-SLAM HSCs was reduced, reinforcing the requirement of GATA2 mitotic bookmarking for proper function of definitive HSCs.

It has been previously reported that GATA2 levels are higher at S-phase and lower at G1/S and G2/M[57]. Using a *Gata2Venus* mouse model, GATA2 protein levels were shown to oscillate in cells under-going EHT[58]. A limitation of our study is that MD-GATA2 protein degradation remains to be characterised in detail throughout the cell cycle in vivo. This characterisation is important given that GATA2 degradation persists for several hours beyond mitotic exit, potentially interfering with GATA2 function in G1. To track MD-Gata2-induced degradation in vivo it would be useful to generate an MD-*Gata2Venus* mouse model, merging our approach with the one reported by Eich et al.[58]. Nevertheless, in that study GATA2 was detected in IAHC cells undergoing mitosis[58], providing a window of opportunity for GATA2-mediated mitotic bookmarking. The impact of GATA2 degradation at M-G1 in the displacement of multimolecular TF complexes from mitotic chromatin, and the consequences for the development of leukaemia remain to be investigated.

In summary, we provided evidence that definitive haematopoiesis is dependent on mitotic bookmarking, supporting a requirement for this mechanism for lineage commitment and blood specification.

Overall, we have established the critical role of GATA2 at M-G1 tran-sition for in vivo hematopoietic development.

## Methods
### Ethical statement
The research presented in this work complies with ethical regulations at Lund University. Animal work was approved by the Malmö - Lund Animal Experimentation Ethics Committee (Malmö - Lunds djurför-söksetiska nämnd).

### Generation of mouse models
MD-*Gata2* and MD$_{mut}$-*Gata2* mouse models were generated in colla-boration with the Core Facility for Transgenic Mice from the University of Copenhagen. Briefly, the MD domain of cyclin B1 or a non-functional mutated (R42A) domain (MD$_{mut}$) were inserted after the ATG start codon, located at the exon 2 of the *Gata2* gene in ESCs via transfection of two vectors: a pX458-GFP vector containing Cas9 and the 20-nucleotide gRNA (GACACAGTAGTGGACCATGG) for the target loca-tion in the *Gata2* gene, together with either the MD or MD$_{mut}$ double-stranded DNA template sequences synthesised in a pMX plasmid (GeneArt, Thermo Fisher Scientific) (Supplementary Data 1). GFP$^+$ cells were FACS-purified, individual clones were isolated and genotyped using a combination of PCR and Sanger sequencing. Five positive ESC clones were expanded and injected into mouse morulae, and later blastocysts were transferred to surrogate females. A heterozygous mouse colony was established for the MD-*Gata2* model. Sperm from MD-*Gata2* male founders was frozen in straws and used to rederive the mouse line at the animal facility at Lund University. For genotyping, tail biopsies or embryonic tissues were individually collected, and diges-ted with QIAamp DNA Mini Kit (Qiagen) or KAPA Express Extract Kit (Roche). Purified genomic DNA was amplified by PCR with Phusion Hot Start II High-Fidelity PCR Master Mix (Thermo Fisher Scientific), according to the manufacturer's instructions (primers in Supplemen-tary Data 1). The same pair of primers was used to genotype CRISPR-modified ESCs. Amplified products were run in a 1% agarose gel and visualised in a ChemiDoc (Bio-Rad) with Image Lab software (version 5.2.1).

### Animal handling
Animal experiments were performed according to the ethical permit protocol 11845/2019 approved by the Malmö - Lund Animal Experi-mentation Ethics Committee (Malmö - Lunds djurförsöksetiska nämnd). All animals (*Mus musculus*), including MD-*Gata2* (129S2;C57BL/6 N), C57BL/6 J (B6), B6.SJL-*Ptprc*$^a$ *Pepc*$^b$/BoyJ (B6.SJL) and C57BL/6JxB6.SJL mice were kept under a 12-hour light/dark cycle with access to food and water *ad libitum*. Temperature was kept at 22 °C and relative humidity at 55%.

## Molecular cloning

For live-cell imaging experiments, mCherry-GATA2/GFI1B/FOS, GATA2/GFI1B/FOS-mCherry sequences and mCherry alone were subcloned into the doxycycline-inducible pFUW-tetO[25] or constitutive pHAGE-MCS lentiviral vectors. For live-cell imaging and western blots, GATA2 deletion constructs[54] were subcloned into the pFUW-tetO fused with mCherry sequence downstream. For live-cell imaging and western blots with mCherry-GATA2 mutants, point mutations were inserted with specific primers in the GATA2 sequence, using QuikChange Lightning Site-Directed Mutagenesis Kit (Agilent) and cloned together with mCherry into the constitutive SFFV lentiviral vector[59], using In-Fusion HD Cloning Kit (Takara), according to the manufacturer's instructions. For reprogramming experiments, individual TFs and GATA2 mutants (L359V and C373R) were subcloned into the constitutive FUW-hUbc (FUW) lentiviral vector[23] using In-Fusion Cloning. A similar method was used to fuse the MD or MD$_{mut}$ sequences upstream mTurquoise fluorescent protein, for protein degradation experiments at different cell cycle phases, or GATA2, for reprogramming experiments, and cloned into the FUW vector. H2B-mTurquoise sequence used to detect DNA in live-cell imaging experiments and mTurquoise alone, also used in protein degradation experiments were subcloned into FUW. Fusion proteins lack the stop codon of the first sequence. Plasmids and cloning primers used in this study can be found in Supplementary Data 1.

## Cell culture

HEK 293T cells (CRL-3216, ATCC), B6 mouse embryonic fibroblasts (MEFs) and HDFs (ScienCell) were maintained in Dulbecco's modified Eagle's (DMEM) medium supplemented with 10% (v/v) FBS, 2 mM L-glutamine and 10 µg/mL penicillin/streptomycin (pen/strep). K562 cells (CCL-243, ATCC) were cultured in RPMI 1640 supplemented with 10% FBS, 2 mM L-Glutamine, 1 mM MEM non-essential amino acids, 1 mM sodium pyruvate and pen/strep. All cells were maintained at 37 °C and 5% (v/v) CO$_2$.

## Lentiviral production

HEK 293T cells were transfected with a mixture of 10 µg transfer plasmid, 7.5 µg packaging plasmid expressing the viral packaging proteins and 2.5 µg envelope plasmid encoding the VSV-G protein, together with 30 µg/mL polyethyleneimine. Viral supernatants were harvested after 36, 48, and 72 hours, filtered (0.45 µm), concentrated 100-fold with Lenti-X Virus Concentrator (Takara), and stored at -80 °C. TF combinations were pool produced.

## Viral transduction and hemogenic reprogramming

Cells were cultured until 60-70% confluency and transduced with pFUW-tetO plus FUW-M2rtTA (1:1), FUW or SFFV lentiviral particles in medium with 8 µg/mL polybrene. Spin infection (800 g for 1 hour at 32 °C) was performed for K562 cells. Medium was replaced the next day and doxycycline (1 µg/mL) added when pFUW-tetO was used. For hemogenic reprogramming of HDFs, cells were transduced twice with FUW lentivirus encoding the 3 TFs or modified versions. Day 0 was considered the day of the second transduction. Cells were split 1:2 at day 4 and cultured in MyeloCult H5100 (StemCell Technologies) supplemented with 1 mM hydrocortisone (StemCell Technologies) and 1X antibiotic-antimycotic (Thermo Fisher Scientific). Medium was changed twice a week.

## Mitotic arrest

Cells for western blot were arrested in pro-metaphase 40 to 48 h post-transduction or after the addition of doxycycline with 0.2 µg/mL nocodazole (Sigma) in complete growth medium for 12–14 h. HDFs and HEK 293T arrested cells were collected by mitotic shake-off and K562 were FACS-purified. In nocodazole arrest and release experiments, cells were arrested as described before, collected by mitotic shake-off and washed with PBS before plating. Cells were collected 2, 4, and 6 h after nocodazole release for western blot analysis. Untreated cells (asynchronous) and unreleased treated cells were also collected. HDFs were also stained for H3 serine 10 phosphorylation (H3S10p) to confirm the accumulation of cells in mitosis. Briefly, cells were spun (Shandon Cytospin 2, Marshall Scientific) onto Polysine-coated slides (Thermo Fisher Scientific) for 10 min at 1,400 rpm, fixed with 1% formaldehyde (FA) for 10 min, washed, permeabilized with 0.4% Triton X-100 and blocked with 5% normal goat serum in PBS. Cells were then incubated with anti-H3S10p primary antibody (1:400) overnight at 4 °C, washed, and incubated with goat anti-rabbit AF488 secondary antibody (1:400) for 45 min, protected from light. Slides were mounted (DAPI-containing mounting medium, Vector Laboratories) and visualised under an inverted Olympus IX70 (Olympus) microscope. See Supplementary Data 1 for antibody information.

## Subcellular protein fractionation, whole-cell protein extraction and western blotting

400,000 sorted K562 or 1-2 million transduced HEK 293T (asynchronous or mitotic) and HDFs (mitotic) were processed with the Subcellular Protein Fractionation kit for Cultured Cells (Thermo Fisher Scientific), according to the manufacturer's instructions to extract the cytoplasmic, soluble nucleus and chromatin-bound protein fractions. For whole-cell extracts, 200 µL of RIPA buffer supplemented with 1X Halt Protease Inhibitor Cocktail (Thermo Fisher Scientific), 1 mM phenylmethylsulfonyl fluoride (PMSF) and 5 mM sodium fluoride (NaF) were added per $1 \times 10^6$ cells. Samples were vortexed and placed on ice every 5 min for a total time of 20 min and centrifuged at 4000 g for 5 min. Protein fractions were diluted 1:2 in Laemmli buffer (Bio-Rad) with 5% 2-Mercaptoethanol (Sigma) and boiled at 98 °C for 10 min. Samples were run in Bolt 4-12%, Bis-Tris (Invitrogen) SDS-PAGE gel, using a Mini Gel Tank (Thermo Fisher Scientific) and Blot MOPS SDS running buffer (Invitrogen). The transfer was performed in an iBlot 2 (Thermo Fisher Scientific) dry system for 7 min. Membranes were incubated overnight at 4 °C with unconjugated primary antibodies against GATA2 (1:1000), GFI1B (1:1000), FOS (1:1000), mCherry (1:1000), Cyclin B1 (1:1000), Histone 3 (1:5000), Calnexin (1:1000) or Actin (1:2000) and with anti-rabbit horseradish peroxidase–conjugated (1:10,000) or anti-mouse horseradish peroxidase–conjugated (1:10,000) secondary antibodies (Supplementary Data 1). Membranes were incubated with ECL prime (Amersham) for 5 min and revealed in a ChemiDoc. Similar numbers of cells were used between conditions. Blots from different membranes from the same experiment were acquired and processed in parallel with similar parameters. Uncropped blots can be found in the Source Data file and in the Supplementary Information file. Bands were quantified using ImageJ (v2.1.0).

## Real-time quantitative PCR (RT-qPCR)

Total cellular RNA was extracted using TRIzol reagent (Thermo Fisher Scientific) following the manufacturer's instructions. The yield and purity of RNA samples were assessed by absorbance in Nanodrop (Thermo Fisher Scientific). 1 µg of total RNA was treated with DNAse I (Roche) and retro-transcribed using SuperScript IV reverse transcriptase (Invitrogen) and Random Primers (Invitrogen), according to the manufacturer's instructions. Quantitative PCR was performed using Maxima SYBR Green/ROX Master mix (Thermo Fisher Scientific) and specific primers (Supplementary Data 1) in a QuantStudio1 Real-Time machine (Applied Biosystems). Raw data was analysed with LinReg PCR software (v2021.2)[60] and N0 fluorescence values were calculated using the same program. Gene expression was normalised to the housekeeping gene GAPDH.

## Live-cell imaging

Widefield live-cell images of HDFs, MEFs and K562 were obtained 48 hours after transduction or addition of doxycycline in 96-well plates (Ibidi), using the Celldiscoverer 7 automated microscope (Zeiss). Random individual positions in wells were imaged for 24-48 h (37 °C, 5% $CO_2$) in 5-10 min intervals and representative snapshots of each mitotic phase were saved from time-lapse data. The same exposure times for mCherry, Pacific Blue (H2B-mTurquoise) and phase-gradient contrast channels were used across conditions. Single mCherry-TF and TF-mCherry images from HEK 293T cells were imaged with an Olympus IX70. Celldiscoverer images were initially processed with ZEN 2 (blue edition, version 3.1) and later with ImageJ (v2.1.0). Images obtained from Olympus IX70 were processed with Adobe Photoshop CS6. Cell counting was performed using the ImageJ plugins StarDist[61] (for segmentation) and Trackmate[62] (for cell tracking) on the time-lapse series. Only mCherry-positive cells were considered for the quantification of mitotic events, and at least 5 different fields were used for quantification. Quantifications do not represent the absolute number of mitotic events or mitotic rate in the wells.

## Chromatin retention analysis

Quantification of mitotic chromatin enrichment was calculated as the ratio between the mean grey value (fluorescence intensity) in the chromatin region of interest (ROI) and the cytoplasm in cells overexpressing mCherry-TFs. Chromatin ROI was defined by delimiting chromosomes marked with H2B-mTurquoise and cytoplasm ROI was defined by delimiting the whole cell perimeter and subtracting chromatin ROI. Cells were analysed with ImageJ (v2.1.0). Mean grey values for each ROI and $Log_2$ (Chr/Cyto) were calculated for statistical analysis.

## Sample preparation for ChIP-seq

**Asynchronous cells.** A total of $1 \times 10^7$ asynchronous K562 cells were crosslinked with 1 mL 2 mM PBS-disuccinimidyl glutarate (DSG) (Sigma) for 50 min at room temperature (RT) with occasional shaking[15]. Then, cells were spun down, resuspended in 10 mL PBS and crosslinked again with 1 mL FA solution (11% FA, 0.1 M NaCl, 1 mM EDTA, 50 mM, HEPES in $H_2O$, FA final concentration 1%), with agitation for 15 min at RT – double fixation (DSG + FA). FA was quenched with 0.125 M glycine for 5 min. Cells were centrifuged at 800 g at 4 °C for 10 min and washed twice with PBS 0.5% Igepal, supplemented with 100 mM PMSF. Cell pellets were snap-frozen and stored at −80 °C.

**Mitotic cells.** A total of $1 \times 10^8$ K562 nocodazole-arrested cells were crosslinked in 2 mL 2 mM PBS-DSG for 50 min at RT, spun down and fixed with 1% FA, as described for asynchronous cells. Cells were stained for FACS with anti-phospho-Ser/Thr-Pro MPM-2 primary antibody (1:170) and anti-mouse AF647 secondary antibody (1:170) (Supplementary Data 1), as described by Campbell et al.[63], with the difference that FA was quenched with 0.5 M glycine. Approximately $5 \times 10^6$ mitotic (MPM-2⁺) K562 were sorted in several rounds, after doublet exclusion resulting in <10% contamination with asynchronous cells. After each sorting, cell pellets were snap-frozen, stored at −80 °C and pulled together for ChIP-seq.

## ChIP-seq

Samples were sent to Active Motif (Carlsbad, CA) for ChIP-Seq. Chromatin was isolated with a lysis buffer, followed by disruption with a Dounce homogeniser. Lysates were sonicated and DNA sheared to an average length of 300-500 bp with Active Motif's EpiShear probe sonicator. Genomic DNA (input) was prepared for asynchronous and mitotic K562 cells by treating aliquots of chromatin with RNase, proteinase K and heat for de-crosslinking, followed by SPRI beads clean up (Beckman Coulter) and quantitation by Clariostar (BMG Labtech). 40 μg of chromatin was precleared with protein A agarose beads (Invitrogen).

Genomic DNA regions of interest were isolated using 5 μg of antibody against GATA2 (Santa Cruz, sc-9008). Complexes were washed, eluted from the beads with SDS buffer, and treated with RNase and proteinase K. Crosslinks were reversed by incubation overnight at 65 °C, and ChIP DNA was purified by phenol-chloroform extraction and ethanol precipitation. Quantitative PCR (qPCR) reactions were carried out in triplicate on specific genomic regions using SYBR Green Supermix (Bio-Rad). The resulting signals were normalised for primer efficiency by carrying out qPCR for each primer pair using Input DNA. Illumina sequencing libraries were prepared from ChIP and Input DNAs on an automated system (Apollo 342, Wafergen Biosystems/Takara). After a final PCR amplification step, the resulting DNA libraries were quantified and sequenced on Illumina's NextSeq 500 (75 nt reads, single end).

## ChIP-seq data analysis

ChIP-seq analysis was performed on raw FASTQ files. FASTQ reads were aligned to the human genome (hg38) using the BWA v0.7.12 algorithm[64]. Only uniquely mapped reads (mapping quality ≥ 25) were used for further analysis. Alignments were extended in silico at their 3′-ends to a length of 200 bp and assigned to 32-nt bins along the genome. Mapped output files were processed through MACS v2.1.0[65] and Genrich v0.6.1 (https://github.com/jsh58/Genrich) software to determine peaks. Peaks that were on the ENCODE blacklist of known false ChIP-Seq peaks were removed. Peak annotation was performed using ChIPseeker R library[66]. For genome tracks, bigwig files were created from bam files with deepTools[67] and explored using the UCSC Genome Browser (https://genome.ucsc.edu/). Heatmaps were produced with deepTools in the reference-point mode where each feature was aligned at GATA2 summits and tiled the flanking up- and downstream regions within ±2 kb. Clusters were generated with K-means clustering ($n = 3$) in deepTools. Functional enrichment analysis for the peaks from asynchronous, mitotic cells and mitotic clusters was performed with GREAT software[68] using GO BP ontology. For motif discovery, findMotifsGenome.pl procedure with default parameters from HOMER[69] was used on ChIP-seq peaks separately. To find the number of GATA2 motifs in peaks we used GATA2 motif from HOCOMOCO v11, and scanned asynchronous, mitotic and mitotic clusters peaks using FIMO from MEME suite. To adjust for the uneven number of peaks in asynchronous versus mitotic cells, we randomly subsampled the number of peaks matching to mitotic condition 1000 times and then averaged the distribution. For chromatin state fold enrichment analysis, enrichment scores for GATA2 asynchronous and mitotic ChIP-seq peaks were calculated using the ChromHMM Overlap Enrichment[70]. ChromHMM segmentation for K562 cell line, containing 18 different chromatin states, was downloaded from the Roadmap Epigenomics Consortium website (https://egg2.wustl.edu/roadmap/web_portal/). To calculate fold change enrichment between two conditions, we normalised each of them by subtracting the minimum value in the condition and then dividing by the maximum value. To compare histone marks distribution in asynchronous and mitotic GATA2 peaks we used all histones marks available for K562 cell line at ENCODE (https://www.encodeproject.org/), along with DNAse-seq and ATAC-seq (Supplementary Data 2). To produce profile matrix and plots, we used deepTools in the reference-point mode centred at GATA2 peak summits and tiled the flanking up- and downstream regions within ±2 kb for histones marks and ±500 bp for DNAse-seq and ATAC-seq. By summing the difference through all bins (bin size = 50 bp), which were estimated in the previous step, we calculated the difference between asynchronous and mitotic states. To investigate motif co-occupancy we downloaded TF motifs, associated with HSPC regulators from HOCOMOCO v11, and proceeded as described above, focusing on the peaks that contained GATA2 motifs, and motifs for other TFs. For integration with other ChIP-seq datasets for other HSPC regulators[38,39], we used bedtools intersect v2.30 (Supplementary Data 2). Replicates were merged by bedtools merge v2.30 when available.

## scRNA-seq sample preparation

A total of 20,000 live (DAPI-) single cells undergoing hemogenic reprogramming were FACS-purified at different time points (day 4 and day 6) following transduction with either MD-GATA2 plus GFI1B and FOS, or $MD_{mut}$-GATA2 plus GFI1B and FOS, using human dermal fibroblast (HDFs) from two different donors (2 biological replicates). Untransduced HDFs were used as reprogramming controls. Sorted samples (Supplementary Data 3) were loaded in a Chromium Next GEM Chip G (10x Genomics) and run in a Chromium Controller (10x Genomics), according to the manufacturer's instructions. scRNA-seq libraries were prepared with Dual Index Chromium Next GEM Single Cell 3′ Kit v3.1 (10x Genomics) according to the manufacturer's instructions. Library size and quality check was performed using High Sensitivity D1000 ScreenTape with High Sensitivity D1000 Reagents (Agilent) in a 4200 TapeStation System (Agilent), according to the manufacturer's protocol. Index libraries were pooled and sequenced on a NovaSeq 6000 System (Illumina) using the NovaSeq 6000 S4 Reagent Kit (200 cycles) v1.5 (Illumina).

## scRNA-seq data analysis

Thirty two thousand seven hundred and seventy-three single cells with an approximate median of UMI counts 60,000 per cell were generated (R1 read: technical, length: 28 bp; R2 read: biological, length: 90 bp). Paired-end sequencing reads were processed using publicly available 10x Genomics software – Cell Ranger v6.0.1[71]. Firstly, we used cellranger mkfastq to convert binary base call files to FASTQ files. Next, we applied cellranger count to FASTQ files and performed alignment to human (hg38, GRCh38.p14) genome assemblies using STAR v2.7.9a[72]. The sparse expression matrix generated by cellranger analysis pipeline was used as input to Seurat R library v4, and cells and genes that passed quality control thresholds were included according to the following criteria: 1) total number of UMIs detected per sample greater than 5000; 2) number of genes detected in each single cell greater than 1000; 3) percentage of counts in mitochondrial genes less than 7.5%. To account for technical variation, we performed batch integration. Firstly, we normalised each batch separately using "LogNormalize" with the scale factor of 10,000 and identified 5000 variable features. Next, we performed batch integration by finding corresponding anchors between the batch using 30 dimensions. We selected the first 30 principal components for subsequent UMAP visualisation. For differential expression analysis between cell states, we used Seurat v4 FindMarkers function using Wilcoxon test with logfc.threshold = 0.1, min.pct = 0.25, BH-adjusted p < 0.05. We further checked how many differentially expressed genes are bookmarked by GATA2 and contain motifs by other HSPC regulators, by scanning mitotic peaks using FIMO from MEME suite and associating them with differentially expressed genes.

## Embryo and hematopoietic tissue isolation

Uteruses from pregnant heterozygous MD-*Gata2* females (8 to 12-weeks old), at indicated developmental stages, were collected and embryos separated from maternal tissue in individual dishes containing PBS 2% FBS, using tweezers. Yolk sacs (E9.5), the caudal region of the embryo proper containing the aorta-gonad-mesonephros (AGM) regions (E10.5 and 11.5), foetal livers (E10.5) and placentas (E10.5 and 11.5) were isolated. To isolate the embryo caudal region (simplified as "AGM"), embryos were cut below the heart and abdominal tissue, including foetal liver, limbs and tail were removed. Embryos were imaged before dissection and hematopoietic tissues were collected after head dissection, which was used for genotyping. Tissues were kept in PBS 2% FBS on ice if not dissociated immediately.

## CFU assays

Embryonic tissues were dissociated in 0.1% (w/v) type I collagenase (Thermo Fisher Scientific), 1% P/S, 10% FBS in PBS (dissociation buffer).

Placentas in 2 mL of dissociation buffer were mechanically disrupted by passing three times through an 18 G needle and incubated at 37 °C, 5% $CO_2$ for 45 min. Following incubation, placentas were disrupted again with a 21 G needle and incubated for an additional 45 min. Finally, cell suspensions were passed through a 23 G needle, filtered (50 μm), washed and pelleted at 300 g for 5 min. Individual yolk sacs, AGMs and foetal livers in 1 mL dissociation buffer were mechanically disrupted by pipetting with a P1000 and incubated for 15 min at 37 °C, 5% $CO_2$, pipetted again with a P200 and incubated again for 15 min. After incubation, cell suspensions were filtered into a new tube, washed and pelleted. Red blood cells (RBC) were lysed with 1x BD Pharm Lyse (BD Bioscience) for 5 min at RT protected from light. After washing, cells were resuspended in Iscove's Modified Dulbecco's Medium (IMDM) (Cytiva) with 2% FBS and resuspended in MethoCult GF M3434 (StemCell Technologies). Placental cells were plated as 0.33 embryo equivalents (ee, triplicates) and yolk sacs, AGMs and foetal livers as 0.5 ee (duplicates). Hematopoietic colonies were scored after 6-7 days of culture.

## Bone marrow isolation for LSK and LSK-SLAM HSC sorting

To quantify allelic expression in MD heterozygous mice and to perform competitive transplantations, bone marrow (BM) was harvested from tibias and femurs by crushing. Cell suspensions were collected with PBS 2% FBS, filtered through a 40 μm cell strainer, washed and pelleted. RBC lysis was performed with BD Pharm Lyse for 8 min at RT protected from light. Whole BM was enriched by magnetic-activated cell sorting for c-kit, using CD117 MicroBeads (Miltenyi), according to the manufacturer's instructions for subsequent FACS.

## FACS

To purify mitotic populations for western blot, K562 cells were sorted according to DNA content, after mitotic arrest. Vybrant DyeCycle Violet DNA permeable stain (Invitrogen) was added directly to cells in growth medium to a final concentration of 5 μM and incubated at 37 °C for 30 min, protected from light. 400,000 K562 cells within G2/M (4 N) gate were sorted immediately after staining for subcellular protein fractionation. For western blot and RT-qPCR analysis, 100-200,000 mCh-TF transduced HDFs were sorted and whole-cell extractions or RNA were collected. For cell cycle-dependent protein degradation experiments, mAzamiGreen positive HEK 293T cells expressing the FUCCI[66] vector were sorted before transduction with mTurquoise lentivirus. mTurquoise positive cells were sorted and cultured before flow cytometry analysis. MD-*Gata2* BMs isolated for LSK sorting were stained with APC-c-kit (1:100), PE-Sca-1 (1:100), PE-Cy5-Lineage: CD3ε/B220/Gr1/Mac1/Ter119 (0.5:100 each) for 20 min on ice. Lineage⁻Sca1⁺c-kit⁺ (LSK) cells were sorted for mRNA extraction. MD-*Gata2* BMs isolated for LSK-SLAM HSC sorting were stained with APCe780-c-kit (1:50), BV421-Sca-1 (1:200), FITC-CD48 (1:200), PECy7-CD150 (1:200), PE-CD45.1 (1:100), APC-CD45.2 (1:100) and PE-Cy5-Lineage: CD3ε/B220/Gr1/Mac1/Ter119 (0.5:100 each) for 20 min on ice. Mitotic K562 for ChIP-seq analysis were sorted after double-fixation and MPM-2 staining as mentioned in "Sample preparation for ChIP-seq". Cells were sorted with either FACSAriaII or FACSAriaIII (BD Biosciences).

## mRNA isolation and RT-PCR

mRNA from BM LSK sorted cells was isolated with 300 μL TRIzol (Invitrogen), reverse transcribed to cDNA with SuperScript IV Reverse Transcriptase (Invitrogen) and amplified by PCR using Phusion Hot Start II, according to the manufacturer's instructions (primers in Supplementary Data 1). PCR products were run in a 1% agarose gel and visualised in a ChemiDoc.

## Transplantation and bleedings

For transplantations assays with embryonic tissue, E11.5 MD-*Gata2* placentas from all genotypes were dissociated as described for CFU

assays with the exception that RBC lysis was not performed, and cells were resuspended in 350 μL PBS 2% FBS. 11 to 14-weeks old B6.SJL (CD45.1) or C57BL/6JxB6.SJL (CD45.1/2) males were sub-lethally irradiated with 600 cGy (Gammacell 40 Exactor), 6 hours before injection. Mice were warmed for 10 min and each recipient was injected (0.5 mL 29 G syringe, Terumo) in the tail vein with 300 μL placenta cell suspensions (1 ee). Bleedings were performed at 4 weeks, 3, and 6 months after injections as follows: blood was collected from the tail vein, spun down and resuspended in 200 μL $NH_4Cl$. After a 5 min incubation at RT in the dark, cells were spun for 5 min at 300 g, washed in 200 μL PBS 2% FBS for flow cytometry analysis. At the 6-month mark, mice were sacrificed, and BM was collected from the left leg and ilium by crushing to assess donor chimerism. For competitive transplantations, 200 FACS-purified LSK-SLAM HSCs (Lineage$^-$Sca-1$^+$Kit$^+$CD150$^+$CD48$^-$) from 11 to 13-weeks old male competitor CD45.1 SLJ mice and from 9 to 13-weeks old male MD/WT or WT/WT MD-*Gata2* mice (CD45.2) were mixed 1:1 and injected into lethally irradiated (900 cGy) 9 to 13-weeks old female recipients (CD45.1/2), together with 200,000 support whole bone marrow cells (CD45.1/2). Blood and bone marrow analyses were performed as described before.

## Whole-embryo bleedings
E10.5 and E11.5 embryos involved by the yolk sac were carefully separated from placenta at RT with fine tweezers to avoid damaging the yolk sac, washed with PBS and placed in individual wells with PBS to bleed. At this point, yolk sacs were open to allow blood to spread in the well for 10 min. After that, both embryo and yolk sac were removed, blood suspensions were passed through a 40 μm strainer into a FACS tube and centrifuged 5 min at 300 g before flow cytometry analysis.

## Flow cytometry analysis
Cells undergoing hemogenic reprogramming were dissociated, pelleted, and incubated with PE-CD9 and PE-Cy7-CD49f antibodies diluted 1:100 in PBS 2% FBS at 4 °C for 20 min, together with mouse serum 1% (v/v). Single live (DAPI-) cells were analysed. Cells expressing FUCCI vector and mTurquoise fluorescent proteins were analysed directly after collection. Mouse-derived blood for donor-contribution analysis was stained with FITC-CD45.1 (1:100), PE-CD45.2 (1:100), APC-B220/CD3ε (0.4:100 each), PECy5-B220/Mac1/Gr1 (0.4:100 each), for 20 min on ice. BMs isolated from mice 6 months after transplantation were treated with BD Pharm Lyse to remove RBC, washed, filtered, and stained for lineage with PeCy5-Ter119/B220/Gr1/Mac1/CD3ε (1:400 each), plus PE-CD45.1 (1:100) and APC-CD45.2 (1:100) antibodies, prior to analysis. Blood cell suspensions from whole-embryo bleedings were incubated with lineage antibodies PE-Cy5-B220/Gr1/Mac1/CD3ε (1:400 each), 7AAD (dead cell exclusion) and the erythroblast development markers FITC-CD71 and APC-Ter119[67] (1:100 each). For EMP analysis, single-cell E9.5 yolk sac suspensions were stained with DAPI, PE-Ter119, APC-eF780-c-kit, FITC-CD41 and APC-CD16/32 (1:100 each), as previously reported[68]. Nocodazole arrest efficiency of HEK 293T and HDFs was assessed by propidium iodide (PI) staining after fixation with 70% ice-cold ethanol. Prior to analysis, ethanol was washed, and cells resuspended in PI buffer (50 μg/mL PI, 100 μg/mL RNAse A, 0.5% of 10% NP-40) for 20 min on ice and 10 min at RT. To check mitotic arrest of K562 for ChIP-seq, asynchronous and nocodazole treated cells were double fixed, stained with MPM-2 (see "Sample preparation for ChIP-seq") and resuspended in PI buffer prior to analysis. Cells were analysed in LSR FORTESSA, LSR FORTESSA x20 or LSRII (BD Biosciences). Flow cytometry results were collected using BD FACSDiva (v9.0) and analysed using FlowJo software (FLOWJO LLC, version 10.6.1).

## Whole-embryo mounting and immunohistochemistry
E9.5 and E10.5 embryos were separated from maternal tissue in PBS (+ Ca/Mg) 10% FBS and heads were isolated for genotyping. Then, embryos were fixed in 4% FA for 60−90 min and rinsed 3 times in 1 mL of PBS for 5 minutes (each wash) at RT. Samples were placed in successively higher concentrations of methanol in PBS on ice for 10 min (50%, 75% and 100%) and stored at -20 °C prior to immunostaining. Embryos were rehydrated in 50% methanol and washed in PBS and incubated in staining buffer (0.4% Triton-X, 2% FCS in PBS) overnight at 4 °C on a shaking rack. Each embryo was incubated for 8 h with gentle rotation at RT with 100-200 μL antibody solution containing either goat anti-CD31 and rabbit anti-RUNX1, or goat anti-CD31 and rabbit anti-RUNX1, plus rat anti-c-kit primary antibodies (1:200 each). Then, embryos were washed in staining buffer overnight at 4 °C on a shaking rack and incubated with the secondary antibody AF594 anti-rabbit (1:800), AF647 anti-goat (1:800) and AF488 anti-rat (1:400) solution for 8 h, gently rotating at RT. See Supplementary Data 1 for more information on antibodies. After a final wash in 1 mL staining buffer overnight at 4 °C, embryos were dehydrated in 50% and 100% methanol and then cleared in 50% and 100% BABB (one part benzyl alcohol with two parts benzyl benzoate). Embryos were then mounted into Fastwells (Sigma) pre-attached to a slide with a coverslip and sealed with another coverslip. Confocal scans were performed using an upright LSM 900 microscope (Zeiss) and the resulting images were processed and analysed using Imaris x64 v9.5.1 (Oxford Instruments). Hematopoietic clusters (CD31$^+$RUNX1$^+$ or CD31$^+$RUNX1$^+$c-kit$^+$) were manually counted. CD31$^+$RUNX1$^+$c-kit$^+$ clusters were only used for quantification purposes.

## Statistics and reproducibility
Comparisons between groups were performed by one-way ANOVA followed by Bonferroni's multiple comparison test, two-way ANOVA followed by Fisher's LSD test or Kruskal-Wallis test followed by uncorrected Dunn's test, or two-tailed Mann-Whitney (for non-parametric data) with GraphPad Prism 9 software. α = 0.5. Seurat v4 FindMarkers function using two-sided Wilcoxon Rank Sum test with logfc.threshold = 0.1, min.pct = 0.25 and adjusted p-value, based on bonferroni correction <0.05, was used for differential expression analysis between cell states. See figure legends for more details. Exact p values are shown when relevant. No statistical method was used to predetermine sample size. No data were excluded from the analyses. Mice were randomly assigned to test groups in transplantation experiments. In vitro experiments were not randomised. CFU colonies were scored blindly.

## Reporting summary
Further information on research design is available in the Nature Portfolio Reporting Summary linked to this article.

## Data availability
Data supporting this work is available upon request. Source data are provided with this paper. ChIP-seq and scRNA-seq data has been deposited in the Gene Expression Omnibus (GEO) with accession codes GSE207551, and GSE221691, respectively. Processed data was mapped to the human genome assembly (hg38, GRCh38.p14). Public datasets used in this study can be accessed under accession numbers GSM733680, GSM7336569, GSM733658, GSM733692, GSM733714, GSM733776, GSM733778, GSM733651, GSM733675, GSM733786, GSM733653, GSM733777, GSM803540, GSE96253, GSM803384, GSM777644, GSM1010820, GSE170378, GSE172523 for Histones ChIP-seq, TFs ChIP-seq, DNAse-seq and ATAC-seq in K562 from ENCODE project; GSM1278240, GSM1278241, GSM1278242 for TFs ChIP-seq in Proerythroblast from Pinello *et al.*, 2013[39]. ChIP-seq and scRNA-seq data generated in this study are provided in Supplementary Data 2 and

Supplementary Data 3, respectively. All other data is provided in the Source Data file. Source data are provided with this paper.

## Code availability

This paper does not report original code. Any additional information required to reanalyse the data reported in this paper is available upon request.

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

## Acknowledgements

We thank the members of the Cell Reprogramming in Haematopoiesis and Immunity Laboratory for useful discussions. We thank Dr. Sjaak Philipsen (Erasmus MC, Netherlands) for kindly providing the GATA2 deletion constructs, as well as Dr. Gerd Blobel (University of Pennsylvania, USA) for the MD-GATA1 and MDmut-GATA1 plasmids and sequences. We also thank Dame Amanda Fisher (Imperial College London, UK) for her critical input. We thank the Center for Translational Genomics and Clinical Genomics Lund (SciLifeLab) for providing sequencing services and Lund University Bioimaging Center for microscopy assistance. We also thank Lund Stem Cell Center FACS Facility for cell sorting, and the Centre for Comparative Medicine for animal facilities, particularly Rosa-Linda Meza and Sofia Ekbjörn for handling the mouse models used in this study. This project has received funding (C.-F.P.) from the Olle Engkvist Foundation (194-0694), FCT (2022.02338.PTDC) and Plano de Recuperação e Resiliência de Portugal pelo fundo *NextGenerationEU* (C644865576-00000005). We would like to acknowledge the Knut and Alice Wallenberg foundation, the Medical Faculty at Lund University and Region Skåne for financial support. R.S.-A received funding from the Royal Physiographic Society of Lund. Research in the de Bruijn group was supported by a program in the MRC Molecular Haematology Unit Core award (MC_UU_00016/02). C.V.E. is supported by a Marie Curie postdoctoral fellowship (101067501). R.S.-A and A.G.F. are supported by FCT scholarships with references PD/BD/135725/2018 and SFRH/BD/133233/2017, respectively.

## Author contributions

R.S.-A. and C.-F.P. were responsible for conceptualisation and experimental design. R.S.-A., J.H., L.T. and A.L. cloned fusion proteins, performed live-cell imaging, immunofluorescence, and western blotting. C.V.E. performed RT-qPCR experiments and quantification of mitotic events. R.S.-A. performed hemogenic reprogramming experiments and flow cytometry analysis. R.S.-A. and A.G.F. FACS-purified cells for protein degradation and ChIP-seq experiments. J.M.-G. and R.B. designed and generated the MD/MDmut-GATA2 mouse models. R.S.-A., A.R. and J.L. performed transplantation assays and interpreted the data. R.S.-A. isolated and dissociated embryonic tissues. M.N., C.R. and M.F.T.R.B. performed whole-embryo microscopy and M.N. quantified hematopoietic clusters. I.K. analysed ChIP-seq and scRNA-seq data. R.S.-A. and C.-F.P. wrote the manuscript. All authors contributed to data interpretation and manuscript revision.

## Funding

## Competing interests

The authors declare no competing interests.
