## [Peer Review File · Nature Communications]

GATA2 mitotic bookmarking is required for definitive haematopoiesisREVIEWER COMMENTS

Reviewer #1 (Remarks to the Author):

In this manuscript, Silverio-Alves and team investigate the potential role of GATA2 in hematopoiesis through mitotic bookmarking using an interesting toolset of cell lines and engineered mouse strains. By using live cell-imaging as well as western blot analysis on mitotic chromatin extracts, they demonstrate that GATA2 remains bound on chromatin throughout cell division whereas some of its partners in the hemogenic process either dissociate from chromatin during mitosis (FOS), or just bind from anaphase onwards (GFI1B). Furthermore, they uncover that GATA2 binding on chromatin is dependent on its C-terminal zinc finger domain, and point mutations in this domain found in AML and Emberger syndrome (ES) patients also impair chromatin binding. The mitotic retention of GATA2 on chromatin is also validated by ChIP-seq analysis, which identifies more than 1,000 mitotic binding sites, enriched for promoters and active enhancers and motifs of known hematopoietic regulators, such as RUNX1 and PU1. Finally, they evaluate the functional impact of GATA2 bookmarking both in the context of hemogenic reprogramming of fibroblasts and in vivo by generating GAT2 fusions with the cyclin B1 degron system (either MD or an inactive mutant), which selectively degrades the protein upon mitotic exit. Extensive characterization of homo/het and WT mouse embryos reveal severe phenotypes and embryonic lethality, supporting an important role of GATA2 during M-to-G1 transition in hematopoiesis.

The manuscript is nicely structured starting from the characterization of GATA2 binding to chromatin in cells, the impaired binding of GATA2 carrying leukemogenic mutations and - more importantly- assessing the bookmarking function in vivo. Most experiments are well-described and many of the conclusions are sufficiently supported by the data. However, there are several aspects that are relatively underdeveloped and require additional characterization and analysis to support the author conclusions.

Major concerns and suggestions:

-Assessing the relative mitotic retention of various TFs and mutants. Most of the conclusions regarding the degree of mitotic retention of the factors of interest are based on ectopically expressed proteins either by imaging or WB analysis of subcellular extracts. However, it is possible that the absolute levels of overexpression might result in unspecific binding and impact the degree of mitotic retention. For example, in Figure 2b it is clear that the mutants with the lowest total levels (DC-ZF and DNLS) are the ones with the lowest degree of retention. Therefore, the authors should present WB analysis of total, cytoplasmic and chromatin extracts both in asynchronous and mitotic cells in for each TF and each mutant that they interrogate by overexpression. This will be also important for properly evaluating the relative retention of each factor on mitotic vs interphase chromatin. For example, is FOS really excluded from the chromatin specifically during mitosis or does it have an overall weak binding throughout cell cycle?

-Genome-wide characterization of GATA2 mitotic binding. The authors describe in detail their elegant approach to isolate mitotic K562 cells at high purity for comparative ChIP-seq analysis with asynchronous cells. The results seem of good quality and the detected mitotic peaks seem reliable. However, the presented analysis is rather superficial and more can be done to gain insights into the potential mechanisms that ensure mitotic retention versus eviction of GATA2. The analyses based on the relative ranking of each peak or gene is a bit confusing. An actual differential analysis (e.g presented as MA plot) or at least K-mean clustering of all detected peaks followed by Gene ontology, motif analysis and enrichment analysis using the plethora of published ChIP-seq datasets in K562 would be very informative and help address the following important questions: (i) What is special between the mitotically-retained vs lost sites (other than the h3k4me1 levels)? Relative strength/affinity of GATA2 binding? Density of binding sites? Overall accessibility? Co-

occupancy with other factors? For the latter, examining the degree of overlap with published FOS or GF11b binding sites would also be of special interest to help interpret for the different bookmarking behaviors (e.g. is mitotic GATA2 overlapping preferentially with GF11b promoting its rapid recruitment during anaphase?).

In vivo interrogation of GATA2 bookmarking function. In principle, the in vivo experiments are the most interesting and novel, since there is no current publication-to my knowledge-that examines the role of TF bookmarking in vivo. For this reason, the engineered mouse strains are very useful and the observed phenotypes are very exciting. However, there are critical controls missing and might lead to overinterpretation of the results.

(1) The authors report that mice homozygous for GATA2-MD degron phenocopy to a large degree the GATA2 mutant, arguing that the presence of GATA2 specifically during M-to-G1 transition is essential for many of its functions. How are we sure that the engineered protein is behaving as expected? WB analysis either of total embryos or specific hemogenic tissues comparing MD, MDmut and WT mice should be provided. Also, a detailed characterization of the degree of degradation and kinetics of recovery should be performed using ectopic GATA2-MD and MDmut constructs either by imaging and/or WB in synchronized cells and upon mitotic exit. Also, given that IF of endogenous GATA2 worked nicely for K562 cells, the authors should try to confirm the mitotic localization of endogenous GATA2 in vivo in any of the contexts that they interrogated (e.g. using fixed cells from the CFU assays) as well as the degree and stages of degradation.

(2) All phenotypic comparisons are done between MD and WT strains. I believe the fairest characterization (after proving similar levels of MD and MDmut GATA2) is between homozygous MD and MD-mut. Key phenotypes should be confirmed in that context.

Minor points and other suggestions:

-The authors mention that mitotic genes are enriched among GATA2 mitotic targets "as expected". Why is GATA2 expected to target/regulate mitotic genes?

-Many of the figure panels appear very small and difficult to read. For example for live-cell imagine, larger cells and spindle images could help to assess better the localization of the proteins over the mitotic cycle. Also, a detailed quantitation of all presented imaging experiments is crucial to evaluate the relative degree of mitotic retention of each factors/mutant of interest.

-Number of replicates/nuclei needs to be reported in each panel.

-To gain some mechanistic insights into the links between GATA2 and RUNX1 (or other hemogenic regulators), a relative straightforward experiment would be gene expression analysis at different cell types comparing MDmut and MD genotypes and integration of the differential genes with the ChIP-seq results. The same could be done for early timepoints during hemogenic reprogramming using either MD-GAT2 or MDmut.

Reviewer #2 (Remarks to the Author):

The manuscript by Silverio-Alves et al. reports the results of experiments examining the chromatin binding and retention aspects of the GATA2 hematopoietic transcription factor during mitosis. The authors perform experiments in a number of different cell lines for the binding/retention of GATA2 and deletion mutants of GATA2, examine the chromatin versus cytoplasmic localization by fluorescence microscopy and ChIP, and compare GATA2 with localization of other transcription factors (GF11B, FOS, RUNXa). The choice of these other factors is based on their previous studies in inducing the differentiation of hemogenic endothelium and hematopoietic cells from human iPSC. The concept of mitotic bookmarking is exciting, novel and unexplored in the field of hematopoietic development. This study sets out to provide in addition to beautiful dynamic imaging data and ChIPseq data, more clear developmental evidence for mitotic bookmarking during endothelial to hematopoietic

transition by constructing a mouse in vivo model with a cyclin-B mitosis degradation domain upstream of the Gata2 gene. The data are well presented and provide support for GATA2 bookmarking, which is localized to the C-terminal zinc finger region.

Specific comments

GATA2 retention could be a special feature of this transcription factor as compared to the others – the results could be strengthened by providing some evidence on the relative rate of transcription of each of the genes in the cell lines and the half-life of each of the transcription factor proteins. The levels of both RNA transcripts and protein of the manipulated transcription factor genes in the cell lines should be provided and along with indications how these levels compare to the endogenous levels.

Questions arise over the ubiquitination sites that may be lacking in the specific GATA2 mutants. Is degradation of any of the GATA2 mutants employed in the study altered? Also, can the rate of translation be affected?

No information is provided for the level of GATA2 mutant protein expression in the transfected cells tested in Figure 2. Have western blots of whole cell extracts been performed and GATA2 levels measured?

In the section titled GATA2 bookmarks key HSC regulators, numbers of binding peaks and genes are provided. How do these numbers compare with the numbers published in the Wilson et al 2010 Cell Stem Cell paper for the mouse hematopoietic cell line?

Concerning the mouse model, it is unclear why the authors state that the MD homozygous mice phenocopy the Gata2 knockout mice. It is clear that the two models differ greatly in the heterozygous mice. And if there is 'an allelic bias in MD-Gata2 heterozygous mice resulting in preferential expression of MD over the WT allele' please explain why there is not a defect in the CFU-C numbers in the AGM, placenta and fetal liver. Wouldn't you expect them to be decreased as in the Gata2 heterozygous mutants?

What do you find in whole mount stained MD-Gata2 heterozygous embryos? Are the numbers of emergent cluster cells the same (number and size) as in the WT or the MD homozygous embryos?

No reference is provided to the data in Eich et al. 2018 JEM concerning the levels and oscillations of Gata2 in the aortic cells undergoing endothelial to hematopoietic transition. A discussion of how these results may fit into the concept of bookmarking should be elaborated.

The authors should avoid using the term HSC (page 3, for example) and use HSPC instead.

Reviewer #3 (Remarks to the Author):

The authors investigated the role of transcription factor retention during mitosis (also known as mitotic bookmarking) during developmental hematopoiesis using a mouse model system that they generated. They first show that GATA2 transcription factor remains localized to the chromatin throughout mitosis in transduced human dermal fibroblasts and mouse embryonic fibroblasts, while two other candidate TFs GFI1B and FOS are either present only in late mitosis, or not at all, respectively. Next, the authors show that the GATA2 C-terminal zinc finger site is required for the protein's mitotic localization to the chromatin. The authors conclude that since several mutated forms of GATA2 found in AML and ES patients are unable to localize to the chromatin, the DNA binding is important for the protein's function. The authors established a mouse model expressing a modified GATA2 protein that is

degraded during mitosis and found that the homozygous progeny is unable to survive past definitive hematopoiesis and the embryos were lacking blood when analysed. The authors attributed the defects to the lack of GATA2 at mitotic exit and suggested that the daughter cells must be born with the GATA2 at key sites to promote hematopoiesis.

Overall the data is of high quality and the authors include a number of very nice controls that support their findings. However, there are a few places in the text where the authors overstate their conclusions. Thus, there are a few issues with the study that should be addressed prior to publication.

Main Points:

Fig. 1f: On page 4, lines 17-20, the authors state “To exclude bias derived from overexpression, we verified mitotic retention of endogenous TFs in K562 cells. As expected, the three TFs showed a similar retention profile as demonstrated in fibroblasts and HEK293T cells (Fig. 1e,f, Extended Data Fig. 1e).” However, Fig. 1f involves K562 cells overexpressing TFs, rather than endogenous. Referencing this subpanel along with Fig. 1e and the preceding sentences is misleading. It gives the impression to the reader that the data in Fig. 1f is from endogenously tagged TFs. The text should be changed to clearly identify which experiments include endogenous proteins and which include over-expressed proteins.

Fig. 1g,h: Given the negative result, the authors should include blotting for the co-expressed proteins to ensure they are being expressed and whether they are in the cytoplasmic and chromatin fractions.

Fig. 2b: The Δ C-ZF mutant looks like it still binds Chr, but there is just less total protein than the other conditions. If the Δ C-ZF mutant was not able to bind Chr, then I would expect an increase in the protein in the Cytoplasmic fraction. In other words, I would have expected a reduction in the size of the band in the Chr fraction and a concomitant increase in the band in the Cytoplasmic fraction in the Δ C-ZF mutant condition (See Fig. 2d, C373R mutant for a perfect example of this). An alternative interpretation of the data in Fig. 2b is that rather than C-ZF being necessary for Chr binding, that the C-ZF is needed for protein stability. If there is less total Δ C-ZF-GATA2 protein in the cell compared to WT-GATA2, then the bands in the IP will appear weaker given equal ability to bind Chromatin. In order to distinguish between this alternative interpretation the authors could try a number of things. Can the authors provide a loading control to help normalize the bands between conditions? Can the authors include a darker exposure of this western blot so the reader can see if there is an increase in the size of the bands in the Cyto fraction? Or perhaps the authors can measure the turnover rate of each mutant to ensure they are all equally stable by doing a Cycloheximide time course experiment? Similar comments apply also to Fig. 2d.

Fig. 2c: The Δ C-ZF mutant overexpression has many foci in the cells in mitosis, which are not observed in interphase cells (as judged by the cells on the edge). The foci could be due to poor protein folding due to the deletion. Similar foci are observed in Fig. 2d for the C373R mutant, which the authors state in the Discussion section impacts protein folding. Could the authors comment on the appearance of the foci and whether this occurs in other cells in mitosis and whether these foci are due to poor protein folding?

Extended Data Fig. 5: The authors validated the destruction of mTurquoise signal when fused to the MD domain but then go on to fuse the MD domain to GATA2 for further experiments. It would be better to validate the MD-GATA2 protein so that they can show the exact timing of its degradation in mitosis. It is difficult to extrapolate the exact timing of protein degradation based on the FACS data. We suggest that the authors fuse the MD and the MDmut to the existing mCherry-GATA2 construct (or generate an mTurquoise-MD-GATA2 construct) and utilize live-cell imaging to demonstrate the loss of MD-GATA2 in mitosis throughout G1. This can replace the current Extended Fig. 5c as it is more relevant for their

future conclusions.

Discussion, Page 11, lines 22-25: The authors state “Removal of the C-terminal zinc finger (C-ZF) of the DNA-binding domain abolished mitotic retention of GATA2, while deletion of the N-terminal region or the N-terminal zinc finger (N-ZF) of the DNA-binding domain did not impact mitotic retention”. The authors are presumably referencing the results from Fig. 2b,c for this statement. Binding to the DNA was not “abolished” by removal of the C-ZF. A band corresponding to the Chr bound fraction is clearly visible in Fig. 2b. Thus, the binding was not abolished, but rather reduced. The authors should consider using the word “reduced” to better describe the results in the Discussion.

Discussion, Page 15, Line 9-10: The authors state “Overall, we have established the critical role of TFs at M-G1 transition for in vivo development”. The authors specifically test only for the role of GATA2 in in vivo development. While this study does include some data for Fos and GF11B, those are all from in vitro experiments. Thus, I think it better to revise this conclusion sentence to reflect that the authors only know about the role of a single TF in vivo from this study.

Minor points:

Fig. 1f: The GATA2 overexpression images lack the scale bar.

Fig. 4b: It would be a nice addition to include either here or in the extended data a similar table with the MDmut ratios.

Extended Data Fig. 5f: Its really hard to see which lines correspond to which conditions. I would suggest either color-coding the lines to the conditions, or make the shapes larger and more distinguishable.

GATA2 mitotic bookmarking is required for definitive haematopoiesis

Response to Reviewers:

We would like to thank the reviewers for their constructive comments. In the following sections, we address their individual concerns. We restate the original comments followed by our detailed responses. Where appropriate, we include Figures presenting new data that address issues raised by the referees.

Reviewer #1

In this manuscript, Silverio-Alves and team investigate the potential role of GATA2 in hematopoiesis through mitotic bookmarking using an interesting toolset of cell lines and engineered mouse strains. By using live cell-imaging as well as western blot analysis on mitotic chromatin extracts, they demonstrate that GATA2 remains bound on chromatin throughout cell division whereas some of its partners in the hemogenic process either dissociate from chromatin during mitosis (FOS), or just bind from anaphase onwards (GFI1B). Furthermore, they uncover that GATA2 binding on chromatin is dependent on its C-terminal zinc finger domain, and point mutations in this domain found in AML and Emberger syndrome (ES) patients also impair chromatin binding. The mitotic retention of GATA2 on chromatin is also validated by ChIP-seq analysis, which identifies more than 1,000 mitotic binding sites, enriched for promoters and active enhancers and motifs of known hematopoietic regulators, such as RUNX1 and PU1. Finally, they evaluate the functional impact of GATA2 bookmarking both in the context of hemogenic reprogramming of fibroblasts and *in vivo* by generating GAT2 fusions with the cyclin B1 degron system (either MD or an inactive mutant), which selectively degrades the protein upon mitotic exit. Extensive characterization of homo/het and WT mouse embryos reveal severe phenotypes and embryonic lethality, supporting an important role of GATA2 during M-to-G1 transition in hematopoiesis.

The manuscript is nicely structured starting from the characterization of GATA2 binding to chromatin in cells, the impaired binding of GATA2 carrying leukemogenic mutations and - more importantly- assessing the bookmarking function *in vivo*. Most experiments are well-described and many of the conclusions are sufficiently supported by the data. However, there are several aspects that are relatively underdeveloped and require additional characterization and analysis to support the author conclusions.

We would like to thank this reviewer for considering our manuscript well-structured and that the conclusions are supported by the data. We are also thankful for the helpful suggestions, which significantly improved our study.

Major concerns and suggestions:

1. Assessing the relative mitotic retention of various TFs and mutants. Most of the conclusions regarding the degree of mitotic retention of the factors of interest are based on

ectopically expressed proteins either by imaging or WB analysis of subcellular extracts. However, it is possible that the absolute levels of overexpression might result in unspecific binding and impact the degree of mitotic retention. For example, in Figure 2b it is clear that the mutants with the lowest total levels (DC-ZF and DNLS) are the ones with the lowest degree of retention. Therefore, the authors should present WB analysis of total, cytoplasmic and chromatin extracts both in asynchronous and mitotic cells in for each TF and each mutant that they interrogate by overexpression. This will be also important for properly evaluating the relative retention of each factor on mitotic vs interphase chromatin. For example, is FOS really excluded from the chromatin specifically during mitosis or does it have an overall weak binding throughout cell cycle?

We understand the reviewer's concern about absolute levels of expression and the possibility that very high levels of expression may result in unspecific binding in mitosis. To exclude this possibility, we performed western blotting (WB) for whole-cell lysates of HDFs overexpressing mCherry-GATA2, mCherry-FOS and mCherry-GFI1B fusion proteins (same number of FACS purified mCherry+ cells), as well as measured total endogenous expression in K562 cells. Additionally, we performed qRT-PCR to quantify endogenous and exogenous transcript levels. We did not find a correlation between absolute levels of expression and the degree of mitotic retention at the transcript or protein levels, supporting our conclusions that mitotic retention is an intrinsic property of each transcription factor and is not a reflection of expression levels. We have included these data in revised Fig. 1f- h (below) and text in page 4, lines 14-15.

Fig. 1. GATA2 is retained at mitotic chromatin independently of cell context. f, Protein expression from HDFs (overexpressed TFs) and K562 (endogenous TFs) whole-cell extracts. HDFs were transduced with mCh-TFs and the same number of mCherry+ cells were FACS sorted for western blotting. CANX and actin were used as loading controls. **g, h,** Transcript levels of each endogenous TF in K562 (**g**) and for overexpressed TFs in HDFs (**h**). Expression was normalized to GAPDH.

We also performed western blot analysis of total, cytoplasmic and nuclear fractions (soluble nucleus and chromatin-bound) in asynchronous and mitotic cells (HEK 293T cells) for overexpressed GATA2, GFI1B and FOS. These data confirmed the retention of GATA2 in the nuclear fractions in mitotic cells, but also in asynchronous cells. GFI1B was highly enriched at the cytoplasmic and soluble nucleus fraction, and FOS was mainly present in the

cytoplasmic fraction in both mitotic and asynchronous cells, suggesting that FOS has indeed an overall weak binding to chromatin throughout the cell cycle. This result suggested that mitotic retention is determined by the overall chromatin binding ability of individual transcription factors. See revised Supplementary Fig. 1b (below) and text in page 3, lines 20-25 and page 4, lines 1-2.

Supplementary Fig. 1 GATA2, GFI1B and FOS chromatin retention in mitotic cells. b, TF expression in cytoplasmic (Cy), soluble nucleus (SN) and chromatin-bound (Chr) protein fractions of asynchronous (Async) and mitotic (Mit) HEK 293T cells, after overexpression of the indicated TF. Calnexin (CANX) and histone 3 (H3) represent loading controls.

Finally, according to the reviewer suggestion we performed WB analysis of whole-cell extracts, as well as cytoplasmic and nuclear fractions (soluble nucleus and chromatin-bound) both in asynchronous and mitotic cells in for each GATA2 mutant and deletion. Our results confirmed that protein abundance itself does not explain mitotic retention. For instance, mutations L359V, R362Q and R398W show similar total protein levels, but very different mitotic retention profiles. We included these results in revised Fig. 2b, d (below) and loading controls for WB analysis in revised Supplementary Fig. 2a, b (below). We also modified results section text in page 5, lines 5-7 and 14-19. We trust that these data will help clarifying that bookmarking does not reflect transcription factor absolute levels both when comparing GATA2, GFI1B and FOS or mutated versions of GATA2.

Fig. 2. Point mutations in GATA2 C-terminal zinc finger associated with leukaemia reduce mitotic retention. b, GATA2 protein expression in whole-cell extracts (WCE), cytoplasmic (Cy), soluble nucleus

(SN) and chromatin-bound (Chr) fractions of both asynchronous (Async) and mitotic (Mit) HEK 293T cells when deletion (Δ) constructs were used excluding the N-terminal (amino acids (aa) 1-235), N-ZF (aa 287-342), C-ZF (aa 243-379) or NLS (aa 380-440). See Supplementary Fig. 2 for loading controls. **d**, Levels of mCherry-fused GATA2 C-ZF mutant proteins in the Cy, SN and Chr protein fractions of asynchronous and mitotic HEK 293T cells. A representative cell in metaphase expressing indicated mCherry-GATA2 mutant is shown (right). Histone 2B (H2B)-mTurquoise signal (blue) indicates DNA content. Scale bars = 10 μ m.

Supplementary Fig. 2. Chromatin retention of GATA2 is reduced by mutations in the C-terminal zinc finger (C-ZF). **a, b**, Western blot analysis of actin and histone 3 loading controls for mCherry-GATA2 deletion constructs (**a**) and mutant proteins (**b**) in asynchronous whole-cell extracts (WCE) and asynchronous and mitotic HEK 293T cell samples after subcellular fractionation. Cy, cytoplasmic protein fraction. SN, soluble nucleus protein fraction. Chr, chromatin-bound protein fraction.

2. Genome-wide characterization of GATA2 mitotic binding. The authors describe in detail their elegant approach to isolate mitotic K562 cells at high purity for comparative ChIP-seq analysis with asynchronous cells. The results seem of good quality and the detected mitotic peaks seem reliable. However, the presented analysis is rather superficial and more can be done to gain insights into the potential mechanisms that ensure mitotic retention versus eviction of GATA2. The analyses based on the relative ranking of each peak or gene is a bit confusing. An actual differential analysis (e.g. presented as MA plot) or at least K-mean clustering of all detected peaks followed by Gene ontology, motif analysis and enrichment analysis using the plethora of published ChIP-seq datasets in K562 would be very informative and help address the following important questions: (i) What is special between the mitotically-retained vs lost sites (other than the h3k4me1 levels)? Relative strength/affinity of GATA2 binding? Density of binding sites? Overall accessibility? Co-occupancy with other factors? For the latter, examining the degree of overlap with published FOS or GFI1b binding sites would also be of special interest to help interpret for the different bookmarking behaviors (e.g. is mitotic GATA2 overlapping preferentially with GFI1b promoting its rapid recruitment during anaphase?).

We thank the reviewer for considering our data of good quality and agree with the reviewer's comment regarding the depth of the analysis. We have now substantially improved our analysis and included the results in revised Figure 3 (see below). Following the reviewer suggestions we performed K-means clustering of all detected peaks followed by Gene ontology BP and motif analysis. K-means clustering of asynchronous peaks resulted in three distinct peak clusters (C1, C2 and C3). Most mitotic peaks (71%) were assigned to cluster 1.

GO BP showed co-enriched terms including “immune system development”, “hematopoietic or lymphoid organ development” and “hemopoiesis” and *de novo* motif analysis revealed that asynchronous and mitotic peaks are enriched for GATA, RUNX and ETS (SPIC) motifs, suggesting that GATA2 cooperates in mitosis with important hematopoietic regulators for the maintenance of HSPC functions. Regarding the density of binding site, we calculated the number of GATA2 binding motifs per peak in asynchronous only and mitotic groups (including the clusters). Interestingly, mitotic peaks contained significantly higher number of GATA2 motifs when compared to asynchronous peaks. This observation suggests that GATA2 mitotic bookmarking is at least partially determined by a pre-existing motif structure where increased number of binding sites is translated in higher TF affinity. Additionally, we expanded our data integration with more histone marks and chromatin accessibility (DNase-seq and ATAC-seq) available for K562 cells from ENCODE project. In addition to H3K4me1, H3K36me3 and H4K20me1 also showed lower abundance at mitotic peaks, which are both associated with active gene transcription. Regarding accessibility to nucleases, we have not found major differences between mitotic and asynchronous peaks, especially when looking at mitotic peaks from cluster 1, suggesting that chromatin accessibility does not represent a barrier for mitotic bookmarking. Next, we investigated TF co-occupancy at GATA2 peaks in asynchronous and mitotic conditions with other important factors for definitive haematopoiesis, including the “heptad” TFs TAL1, LYL1, RUNX1, ERG, and FLI1 and other important hematopoietic regulators such as PU.1, MYB, PBX1, GFI1B, HES1, MEIS1 and HLF. We can see high degree of motif co-occurrence at mitotic peaks particularly for the heptad TFs and PU.1. This was confirmed by overlapping available peak data for GATA2, RUNX1, PU.1, FOS, GATA1, TAL1 and GFI1B TFs showing higher peak overlap at mitosis when compared to asynchronous cells, suggesting that previously formed TF complexes are maintained in mitosis. However, we have not seen a preferential overlap with GFI1B when compared to the other HSPC regulators that could explain its rapid recruitment during anaphase as the reviewer suggests. We have included the results in the text (pages 6 and 7, and page 8, lines 1-3. And revised Fig. 3). We have also removed the ranked-up and ranked-down analysis that was included in the original manuscript that the reviewer found confusing. We believe that these new results represent major advances in our understanding of mitotic bookmarking and thank the reviewer for the valuable suggestions.

Fig. 3. GATA2 bookmarks a subset of interphasic target genes with roles in definitive haematopoiesis. a, Venn diagram showing the number of ChIP-seq GATA2 peaks and genes shared between asynchronous (Async) and mitotic (Mit) K562 cells. **b**, Gene tracks for GATA2 binding sites at GATA2 and RUNX1 loci showing both bookmarked (grey) and asynchronous unique peaks. Kb –

kilobase. **c**, K-means clustering of asynchronous (left) and mitotic peaks (right). The percentage of mitotic peaks in each cluster is shown. **d**, Gene Ontology (GO) biological processes (BP) for the top 1,000 gene-related peaks in asynchronous, mitotic and mitotic clusters 1 (C1), 2 (C2) and 3 (C3). Categories that contain more than 5 peaks per category are displayed. Coloured scale represents the adjusted p-value and the circle size the number of peaks per group. **e**, De novo motif enrichment for GATA2 target sites using asynchronous cells, mitotic and mitotic clusters. Top three motifs are shown with respective p-values. **f**, Number of GATA2 motifs in individual peaks in asynchronous only, mitotic and mitotic clusters 1 and 2. **g**, Chromatin-state enrichment heatmap representing the percentage of genome occupancy of GATA2 per group of peaks. Scale represents the percentage of peaks at each genomic segment. TSS – Transcription start site. **h**, Integration heatmap with histones marks, DNase-seq and ATAC-seq data for K562 cells (ENCODE). Scale represents the accumulated sum differences across bins between asynchronous peaks only and mitotic peaks. **i**, Histone marks and ATAC-seq profiles at peak summit (centre). **j**, Percentage of GATA2 peaks where motifs for relevant factors HSPCs regulators are also present. **k**, Percentage of overlap between GATA2 peaks and peaks for HSPC regulators from available CHIP-seq datasets.

3. In vivo interrogation of GATA2 bookmarking function. In principle, the in vivo experiments are the most interesting and novel, since there is no current publication-to my knowledge- that examines the role of TF bookmarking in vivo. For this reason, the engineered mouse strains are very useful and the observed phenotypes are very exciting. However, there are critical controls missing and might lead to overinterpretation of the results.

(1) The authors report that mice homozygous for GATA2-MD degron phenocopy to a large degree the GATA2 mutant, arguing that the presence of GATA2 specifically during M-to-G1 transition is essential for many of its functions. How are we sure that the engineered protein is behaving as expected? WB analysis either of total embryos or specific hemogenic tissues comparing MD, MDmut and WT mice should be provided. Also, a detailed characterization of the degree of degradation and kinetics of recovery should be performed using ectopic GATA2-MD and MDmut constructs either by imaging and/or WB in synchronized cells and upon mitotic exit. Also, given that IF of endogenous GATA2 worked nicely for K562 cells, the authors should try to confirm the mitotic localization of endogenous GATA2 in vivo in any of the contexts that they interrogated (e.g. using fixed cells from the CFU assays) as well as the degree and stages of degradation.

We would like to thank the reviewer for acknowledging the novelty and interest of our *in vivo* experiments. Indeed, to our knowledge this is the first demonstration of the relevance for this mechanism *in vivo*. We understand the reviewer suggestion to map GATA2 degradation *in vivo*, however these are very challenging experiments. We would like to clarify that the results shown in Fig. 1 regarding TF mitotic retention in K562 cells were not performed by IF, but by overexpressing fluorescent fusion proteins followed by live-cell imaging. Only the WBs shown in revised Fig. 1e were performed to assess mitotic retention of endogenous proteins. We understand that this may be confusing for the reader, so we have modified the text (page 4, lines 2-7 and 11-13) to clearly separate exogenous versus endogenous TF retention and moved the imaging data in K562 cells to supplementary information (Supplementary Fig. 1e). This makes the mapping of degradation *in vivo* very difficult to

address with the current mouse model. We would have to generate another mouse model such as inserting the MD in GATA2-Venus model¹ to properly map degradation kinetics *in vivo* which is beyond the scope of this paper. Nevertheless, and as suggested, we attempted to perform WB of total embryos (excluding head), but this was not successful (see below).

Western blot analysis of E9.5 heterozygous (MD/WT) embryos. Four embryos (Emb) were dissociated directly with RIPA buffer for 20 min on ice with vortexing every 5 min. K562 is shown as positive control. Single cell suspensions with collagenase before incubation with RIPA also did not result in GATA2 protein detection (not shown).

As an alternative, we have performed CFU assays with E9.5 yolk sac cell suspensions from heterozygous MD-GATA2 mice, followed by WB of hematopoietic colonies, where we showed that the MD-GATA2 protein is expressed *in vivo* (see text page 10, lines 7-9). We added this data to the revised Supplementary Fig. 6h. In the timeframe of the revision for the paper, we were only able to include heterozygous yolk sacs in this analysis as these experiments are dependent on successful timed-matings that are difficult to predict.

Supplementary Fig. 6. Characterisation of MD- and MD_{mut}-Gata2 mouse models. h, Western blot analysis of heterozygous MD-GATA2 E9.5 yolk sac-derived hematopoietic colonies after 6 days of culture. Bands corresponding to MD-GATA2 (MD) and wild-type (WT) proteins are shown.

Regarding the degree of degradation and kinetics of recovery, we have performed WB analysis of asynchronous and synchronized cells before and after nocodazole release as suggested. These data showed that degradation is happening with MD-GATA2 in the timeframe of 4 hours and gradually increases until levels go back to normal. We have inserted the data in revised Fig. 4a (below) and revised text (page 8, lines 8-15). In addition, we included a paragraph in the discussion with the limitations of our study in this regard: “A limitation of our study is that MD-GATA2 protein degradation *in vivo* remains to be characterized in detail throughout the cell cycle. To track MD-Gata2-induced degradation *in vivo* it would be useful to generate an MD-Gata2Venus mouse model, merging our approach with the one reported by Eich *et al.*⁵⁸” (page 15, lines 23-25 and page 16, line 1).

Fig. 4. GATA2 is required at the mitosis-to-G1 transition *in vivo* for embryonic development a, Western blot analysis for GATA2 protein levels in HEK 293T cells expressing GATA2 fused to the mitotic degradation (MD) domain of cyclin B1 or an inactive form (MD_{mut}) before (no release) and after release from nocodazole arrest. Actin was included as loading control. Hours (h) after nocodazole release are shown. Async – Asynchronous cells.

(2) All phenotypic comparisons are done between MD and WT strains. I believe the fairest characterization (after proving similar levels of MD and MDmut GATA2) is between homozygous MD and MD-mut. Key phenotypes should be confirmed in that context.

We agree with the reviewer comment, and we have included comparison of MD and MDmut GATA2 for E11.5 embryo bleedings (revised Fig. 4j) and E11.5 AGM and placenta CFU assays (see below and revised Fig. 5e). MDmut homozygous embryos do not show signs of anemia with erythrocyte production being comparable to WT/WT and MD/WT embryos. Moreover, Mut/Mut AGM and placenta HSPCs give rise to hematopoietic colonies in comparable numbers to WT embryos, which are significantly higher when compared to MD/MD embryos. We have revised the figures and figure legends as shown below.

Fig. 4. GATA2 is required at the mitosis-to-G1 transition *in vivo* for embryonic development. i, j, Flow cytometry quantification of E10.5 (i) and E11.5 (j) erythroblasts after whole MD-*Gata2* embryo bleeding. Bar graphs show percentage of total erythroblasts (Erythro) or immature (type I) to mature (Type III) erythroblasts gated within lineage negative (Lin-) live single cell (SC) population. E11.5 bleeding include the MD_{mut} (Mut/Mut) homozygous control. n(E10.5 WT/WT)=3, n(E10.5 MD/WT)=5, n(E10.5 MD/MD)=3 n(E11.5 WT/WT)=4, n(E11.5 MD/WT)=5, n(E11.5 Mut/Mut)=9, n(E11.5 MD/MD)=5. Mean±SD is shown.

Fig. 5. GATA2 bookmarking is required for definitive haematopoiesis. **e**, Colony-forming unit assays at E11.5 using AGM and placenta cell suspensions. Mean±SD is represented. Macrophage (M), granulocyte (G), granulocyte/macrophage (GM), erythroid (E) and mixed colonies (Mix) are shown per embryo equivalents (ee). Statistical significance for the total number of colonies was calculated by one-way ANOVA followed by Bonferroni's multiple comparison test. * $p < 0.05$, ** $p < 0.01$, *** $p < 0.001$, ns – nonsignificant.

Minor points and other suggestions:

-The authors mention that mitotic genes are enriched among GATA2 mitotic targets “as expected”. Why is GATA2 expected to target/regulate mitotic genes?

We agree with the reviewer comment, and we have removed peak ranking analysis as previously described in point 2. We have also removed associated text from the revised manuscript.

-Many of the figure panels appear very small and difficult to read. For example for live-cell imagine, larger cells and spindle images could help to assess better the localization of the proteins over the mitotic cycle.

Agreed. We have increased font size from 8 to 9 points and 10 to 11 points in all figures. Moreover, we have increased the size of imaging panels, namely in Fig. 1c, Fig. 2c, d, Supplementary Fig. 1d, e, f, Supplementary Fig. 2c-j, Supplementary Fig. 3a-f.

Also, a detailed quantitation of all presented imaging experiments is crucial to evaluate the relative degree of mitotic retention of each factors/mutant of interest.

The quantification of the degree of mitotic retention of each deletion and mutant was performed by WB analysis, which now includes the soluble nucleus fraction for additional resolution on protein subcellular distribution for each construct. Please see revised Fig. 2b, d and Supplementary Fig 2a, b. We also have included the quantification of number of nuclei observed in imaging experiments (see point below).

-Number of replicates/nuclei needs to be reported in each panel.

We have now included the number of mitotic events per condition in each panel, as we believe this parameter is more representative of the result shown by the imaging data. All observed dividing cells displayed the same mitotic retention behavior in each condition. Quantifications can be found in figure legends of Fig 1, Fig 2, Supplementary Fig. 2 and

Supplementary Fig. 3. Quantification method can be found in the “Methods” section under “Live-cell imaging”.

-To gain some mechanistic insights into the links between GATA2 and RUNX1 (or other hemogenic regulators), a relative straightforward experiment would be gene expression analysis at different cell types comparing MDmut and MD genotypes and integration of the differential genes with the ChIP-seq results. The same could be done for early timepoints during hemogenic reprogramming using either MD-GATA2 or MDmut.

We have followed the suggestion and performed single-cell RNA-seq at early time points of hemogenic reprogramming (day 4 and day 6) from 2 independent human fibroblasts donors’ when reprogramming was elicited with GF11B, FOS and MD-GATA2 or MDmut-GATA2. We profiled 32,773 cells and confirmed that cells derived from the two donors undergo similar transcriptional changes using Uniform Manifold Approximation and Projection (UMAP). In agreement with the FACS data, CD9 expression is delayed when reprogramming was induced with MD-GATA2. To investigate the differences between the transcriptional program in MD-GATA2 and MD_{mut}-GATA2 reprogrammed cells, we performed differential gene expression analysis focusing on GATA2 bookmarked targets. This analysis revealed that 28 and 75 genes were downregulated, and 18 and 37 genes were upregulated at day 4 and day 6 respectively, when GATA2 was degraded at mitotic exit, pointing out to a preferential role of GATA2 at M-G1 in the activation of the hematopoietic program through reprogramming. Downregulated genes with MD-GATA2 included ZBTB16, an epigenetic regulator of HSC fate and TEAD1 which helps driving hematopoietic specification. Reassuringly, genes that require GATA2 at mitotic exit during reprogramming were predicted to be regulated by GATA2 with ChIP Enrichment Analysis (ChEA), contained GATA2 motifs, as well as motifs the other hematopoietic TFs. We observed high enrichment with the heptad TFs members, especially at day 6, suggesting their cooperative action also in mitosis to impose the hemogenic fate. We added a new text section describing the main results in page 8, lines 21-25 and page 9, lines 1-15. We also revised Figure 4d, e and Supplementary Fig. 5d-g (below). We have included a Supplementary Table 3 with the differential expression analysis for d4 and d6 of hemogenic reprogramming with MD_{mut}-GATA2 or MD-GATA2 and ChIP Enrichment Analysis (ChEA) with the top 25 gene regulators.

Fig. 4. GATA2 is required at the mitosis-to-G1 transition *in vivo* for embryonic development. d, HDFs, and live transduced cells in early reprogramming stages were profiled by single cell RNA-seq.

Volcano plots showing differential gene expression of GATA2 bookmarked genes at day 4 and day 6 of reprogramming with MD-GATA2 and MD_{mut}-GATA2. Relevant genes downregulated in MD-GATA2 condition (left) are highlighted in red. **e**, Number of bookmarked differentially expressed genes between MD-GATA2 and MD_{mut}-GATA2 conditions at d4 and d6, with binding sites for HSPC regulators. Upregulated genes in MD_{mut}-GATA2 condition at d4 and d6 (d4_MD_{mut} and d6_MD_{mut}, respectively), and upregulated genes in MD-GATA2 at d4 and d6 (d4_MD and d6_MD, respectively) are shown.

Supplementary Fig. 5. Protein degradation driven by the mitotic degradation (MD) domain of cyclin B1 delays hemogenic reprogramming. **d-f**, Uniform Manifold Approximation and Projection (UMAP) analysis of 32,773 single transcriptomes of FACS sorted, live (Dapi-) human dermal fibroblasts (HDFs) and HDFs undergoing hemogenic reprogramming with overexpression of GFI1B, FOS and MD-GATA2 or GFI1B, FOS and MD_{mut}-GATA2 from two donors. **(d)** shows single cells coloured by donor (Donor 1 and Donor 2) **(e)** highlights untransduced HDFs and reprogrammed cells at day (d) 4 and d6 and **(f)** shows transduced cells with either MD-GATA2 (left) or MD_{mut}-GATA2 (right). **g**, Expression levels of the *CD9* gene in untransduced HDFs and in MD-GATA2 or MD_{mut}-GATA2 transduced cells at d4 and d6 of hemogenic reprogramming.

These data have generated mechanistic insights into the impact of degrading of GATA2 at M-G1 transition by identifying downstream targets that are downregulated at early reprogramming stages and may interfere with the specification of the hemogenic fate. We would like to thank the reviewer for suggesting this insightful experiment.

Reviewer #2

The manuscript by Silverio-Alves et al. reports the results of experiments examining the chromatin binding and retention aspects of the GATA2 hematopoietic transcription factor during mitosis. The authors perform experiments in a number of different cell lines for the binding/retention of GATA2 and deletion mutants of GATA2, examine the chromatin versus cytoplasmic localization by fluorescence microscopy and ChIP, and compare GATA2 with localization of other transcription factors (GFI1B, FOS, RUNXa). The choice of these other factors is based on their previous studies in inducing the differentiation of hemogenic endothelium and hematopoietic cells from human iPSC. The concept of mitotic bookmarking is exciting, novel and unexplored in the field of hematopoietic development. This study sets out to provide in addition to beautiful dynamic imaging data and ChIPseq data, more clear developmental evidence for mitotic bookmarking during endothelial to hematopoietic transition by constructing a mouse in vivo model with a cyclin-B mitosis degradation domain upstream of the Gata2 gene. The data are well presented and provide support for GATA2 bookmarking, which is localized to the C-terminal zinc finger region.

We would like to thank the reviewer for acknowledging the interest and novelty of our study, as well as for the helpful comments which increased the overall quality of this work.

Specific comments:

1. GATA2 retention could be a special feature of this transcription factor as compared to the others – the results could be strengthened by providing some evidence on the relative rate of transcription of each of the genes in the cell lines and the half-life of each of the transcription factor proteins. The levels of both RNA transcripts and protein of the manipulated transcription factor genes in the cell lines should be provided and along with indications how these levels compare to the endogenous levels.

We understand the reviewer's concerns about absolute levels of transcript and protein could cause increased or decreased binding in mitosis. We have performed WB and qRT-PCR in HDFs overexpressing GATA2, GFI1B and FOS as well as measured total endogenous expression in K562 cells. We have conclusively shown that mitotic bookmarking does not correlate with absolute protein or transcript levels and is a characteristic feature of GATA2. See Reviewer 1, point 1 for a detailed response.

Regarding protein half-life, we attempted to address this by performing a cycloheximide-chase assay with the fusion proteins mCherry-GATA2, mCherry-GFI1B and mCherry-FOS and transfect in HEK 293T cells (similar amounts of plasmid) and evaluate protein half-life by flow cytometry. However, since our proteins are being overexpressed, this assay was suboptimal for detecting protein degradation (see below) even when adding high concentration of drug without causing toxicity (250 µg/mL). Notwithstanding, since we show that protein abundance does not provide an explanation for mitotic retention kinetics, both when comparing across different transcription factors or mutated forms of GATA2 we find it unlikely that protein half-time would explain mitotic retention ability of each TF.

Cycloheximide-chase assay. HEK 293T cells were transfected with individual transcription factors fused to mCherry protein. 250 $\mu\text{g}/\text{mL}$ of Cycloheximide was added to cultures 40 hours after transfection and cells were collected at 2, 4, 6 or 8 hours (h) for flow cytometry analysis. DMSO treated cells were used as control. MFI, Mean fluorescent intensity.

2. Questions arise over the ubiquitination sites that may be lacking in the specific GATA2 mutants. Is degradation of any of the GATA2 mutants employed in the study altered? Also, can the rate of translation be affected?

Indeed, GATA2 protein has several ubiquitination sites mainly in the N-terminal region of the protein², which may explain why there is more mCherry- ΔN -terminal protein when compared to other deletion constructs, as assessed by WB of whole cell extracts (Fig. 2b). So, we agree with the reviewer, and it is indeed possible that mutations/deletions can affect the degradation kinetics of the GATA2 protein. However, we now show in revised Figure 2 quantification of total GATA2 protein in whole cell extracts for all deletions and mutants. As we detail in response to Reviewer 1, GATA2 mutants with very similar total expression show very different patterns of retention in mitosis, not supporting to the hypothesis that the rate of translation would explain binding to chromatin in mitosis.

3. No information is provided for the level of GATA2 mutant protein expression in the transfected cells tested in Figure 2. Have western blots of whole cell extracts been performed and GATA2 levels measured?

According to the reviewer suggestion (and also in response to Reviewer 1, point 1) we have performed WB analysis of whole-cell extracts for each mutant and deletion. Please see Reviewer 1, point 1 for a detailed response.

4. In the section titled GATA2 bookmarks key HSC regulators, numbers of binding peaks and genes are provided. How do these numbers compare with the numbers published in the Wilson et al 2010 Cell Stem Cell paper for the mouse hematopoietic cell line?

We thank the reviewer for the great suggestion that improved our manuscript by providing mechanistic insights into cooperativity with other HSPC regulators. We started by directly overlapping GATA2 peaks in asynchronous cells and mitosis with the peaks from Wilson *et al.*, 2010³ generated by ChIP-seq in asynchronous cells in a mouse hematopoietic cell line (see below). To perform a direct comparison the peaks from Wilson et al. were “lift-over” to hg38 genome version.

Percentage of peak overlap between GATA2 peaks in asynchronous (Async) cells and mitosis (Mit) with the peaks from Wilson *et al.*, 2010.

We observed little overlap, possibly related to the challenges to compare ChIP-seq data (and peaks) across mouse and human genomes. As an alternative, we compared our data with publicly available ChIP-seq data from human cells, namely K562 (ENCODE) and proerythroblasts⁴. This new data is now included in revised Fig. 3k. We showed high percentage of overlap, particularly for GATA2 mitotic peaks with RUNX1, PU.1, FOS, GATA1 and TAL1. GATA2 validated our integration and CTCF showed little overlap or enrichment in mitosis, which we used as negative reference. We have described these results in page 7, lines 23-25 and page 8, lines 1-2.

Fig. 3. GATA2 bookmarks a subset of interphasic target genes with roles in definitive haematopoiesis. k. Percentage of overlap between GATA2 peaks and peaks for HSPC regulators from available ChIP-seq datasets.

5. Concerning the mouse model, it is unclear why the authors state that the MD homozygous mice phenocopy the Gata2 knockout mice. It is clear that the two models differ greatly in the heterozygous mice. And if there is ‘an allelic bias in MD-Gata2 heterozygous mice resulting in preferential expression of MD over the WT allele’ please explain why there is not a defect in the CFU-C numbers in the AGM, placenta and fetal liver. Wouldn’t you expect them to be decreased as in the Gata2 heterozygous mutants?

We understand the reviewer’s comment and although there are some differences in the phenotypes of Gata2 full KO and MD-GATA2, the overall impact for definitive hematopoiesis is very similar - lethality by anemia between day E10.5 and E11.5. We believe that this is the major point that justifies using the term “phenocopy”. On a more detailed level we observe differences such as the functional yolk sacs, similar numbers of CFUs generated by MD-Gata2 heterozygous mice in MD-GATA2 mice and non-Mendelian ratios in MD-GATA2 mice.

Because of this we have modified the abstract of the paper refer to the phenotype as “partial phenocopy”. The differences in phenotype between heterozygous MD-GATA2 and GATA2 full knockout could be explained by either different requirements for mitotic bookmarking of blood progenitors throughout development or different cell cycle dependencies. Even though we observe an allelic bias in heterozygous mice the GATA2 protein is still expressed in interphase, and this may be sufficient to compensate for some developmental functions in heterozygous mice. Nevertheless, we used adult mice to evaluate the impact of degrading GATA2 in mitosis in heterozygosity. We have performed competitive transplantation of LSK-SLAM HSCs and observed a reduction in engraftment capacity of MD-Gata2 heterozygous stem cells. With these data we concluded that degrading GATA2 in mitosis from a single allele is sufficient to reveal functional defects in adult hematopoietic stem cells. We have included these data in revised Fig. 5i-k and revised Supplementary Fig. 8d, e (below). We would like to thank the reviewer to highlight this topic that greatly improved our manuscript. We will explore the impact of degrading GATA2 in adult stem cells and their progeny in more detail in a follow-up manuscript. We have included a paragraph in the discussion of the phenotype of heterozygous mice during development and the adult, in page 15, lines 12-20 as follows: “GATA2 haploinsufficient mice (*Gata2*^{+/-}) show a 2.7- and 1.6-fold decrease in the number of CFU colonies derived from E10 AGM and YS, respectively⁴⁵. Moreover, *Gata2*^{+/-} bone marrow HSPCs showed reduced engraftment capacity⁵⁶. In contrast, embryonic MD-GATA2^{+/-} HSPCs generated similar numbers of CFUs than wild-type HSPCs and had similar numbers of IAHCs. These could be explained by either a differential requirement for mitotic bookmarking of blood progenitors throughout development or different cell cycle dependencies. Nevertheless, engraftment of irradiated hosts by bone marrow heterozygous LSK-SLAM HSCs was reduced, reinforcing the requirement of GATA2 mitotic bookmarking for proper function of definitive HSCs.

Fig. 5. GATA2 bookmarking in required for definitive haematopoiesis. **i**, Percentage of donor chimerism 4w, 3m and 6m after competitive transplantation with 200 sorted LSK-SLAM (Lineage⁻Sca-1⁺Kit⁺CD150⁺CD48⁻) HSCs from WT or MD/WT adult mice and 200,000 support bone marrow cells (see Supplementary Figure 7). n(WT/WT)=6, n(MD/WT)=8. **j**, Quantification of donor contribution to myeloid and lymphoid lineages. **k**, Percentage of donor chimerism in bone marrow 6 months after transplantation. Statistical significance calculated with Kruskal-Wallis test followed by uncorrected Dunn's test.

Supplementary Fig. 8. Embryonic and adult transplantation strategies to assess the role of GATA2 at mitosis-to-G1 transition for HSPC generation and maintenance. **d**, Competitive transplantation strategy to address function of adult bone marrow HSCs. Two hundred Lineage⁻Sca-1⁺Kit⁺CD150⁺CD48⁻LSK-SLAM HSCs from a competitor CD45.1 SJL mouse or from MD/WT or WT/WT MD-*Gata2* mice were FACS purified, mixed 1:1 and injected into lethally irradiated CD45.1/2 hosts, together with 200,000 support whole bone marrow (WBM) cells. Blood was collected for analysis 4w, 3m and 6m after transplantation to assess donor engraftment (CD45.2⁺) and contribution to myeloid, B-cell and T-cell lineages. Recipients' bone marrow was analysed at the experimental end-point of 6 months. **e**, Representative gating strategy to evaluate donor chimerism (CD45.2⁺) and for assessing the percentage of myeloid (Gr1+Mac1⁺), B- (B220⁺) and T- (CD3ε⁺) donor-derived cells.

6. What do you find in whole mount stained MD-*Gata2* heterozygous embryos? Are the numbers of emergent cluster cells the same (number and size) as in the WT or the MD homozygous embryos?

In line with reviewer's comments, we have repeated the analysis with more embryos including MD/WT E10.5 embryos and quantified intra-aortic clusters. We observed an abolishment of hematopoietic clusters in the ventral region of the dorsal aorta and severe reduction at the dorsal region in MD homozygous embryos at E10.5 (Fig. 5a, b). The data was inserted in revised Figure 5, panel a and b (below). We have modified the results section in page 10, lines 19-22. Even though the number of heterozygous clusters does not differ between WT and WT/MD heterozygous embryos, heterozygous clusters appear to be smaller when compared to the WT.

Fig. 5. GATA2 bookmarking is required for definitive haematopoiesis. **a**, Immunohistochemistry images representing E10.5 wild-type (WT/WT), heterozygous (MD/WT) and homozygous (MD/MD) intra-aortic hematopoietic clusters expressing RUNX1 (red) and CD31 (white) in the ventral (V) or dorsal (D) sides of the dorsal aorta (DA). Clusters are indicated with white arrowheads. Scale bars = 150 μm. **b**, Intra-aortic hematopoietic cluster quantification per genotype. n(WT/WT)=3, n(MD/WT)=2, n(MD/MD)=2 embryos.

7. No reference is provided to the data in Eich et al. 2018 JEM concerning the levels and oscillations of Gata2 in the aortic cells undergoing endothelial to hematopoietic transition. A discussion of how these results may fit into the concept of bookmarking should be elaborated.

We thank the reviewer for highlighting this paper. Indeed, GATA2 is more expressed in S and less expressed in G1/S and G2/M⁷ and oscillations in Gata2 expression were observed during EHT¹. To contrast the concept of bookmarking with previous findings we have added a paragraph to the discussion: “It has been previously reported that GATA2 levels are higher at S-phase and lower at G1/S and G2/M⁵⁷. Using a *Gata2Venus* mouse model, GATA2 protein levels was shown to oscillate in cells undergoing EHT⁵⁸. A limitation of our study is that MD-GATA2 protein degradation *in vivo* remains to be characterized in detail throughout the cell cycle. To track MD-Gata2-induced degradation *in vivo* it would be useful to generate an MD-*Gata2Venus* mouse model, merging our approach with the one reported by Eich *et al.*⁵⁸. Nevertheless, in this study GATA2 was detected in IAHC cells undergoing mitosis⁵⁸, providing a window of opportunity for GATA2-mediated mitotic bookmarking. Whether GATA2 degradation at M-G1 results in the displacement of multimolecular TF complexes from mitotic chromatin affecting TF cooperativity in mitosis and interphase and the impact in leukaemia remains to be investigated.” In page 15, lines 21-25 and page 16, lines 1-5.

8. The authors should avoid using the term HSC (page 3, for example) and use HSPC instead.

Agreed. We have corrected the text in the revised manuscript, namely: page 6, line 2; page 7, line 20; page 8, lines 2 and 16; page 11, lines 12 and 19; page 12, lines 3 and 4; page 15, lines 6, 14-16.

Reviewer #3

The authors investigated the role of transcription factor retention during mitosis (also known as mitotic bookmarking) during developmental hematopoiesis using a mouse model system that they generated. They first show that GATA2 transcription factor remains localized to the chromatin throughout mitosis in transduced human dermal fibroblasts and mouse embryonic fibroblasts, while two other candidate TFs GFI1B and FOS are either present only in late mitosis, or not at all, respectively. Next, the authors show that the GATA2 C-terminal zinc finger site is required for the protein's mitotic localization to the chromatin. The authors conclude that since several mutated forms of GATA2 found in AML and ES patients are unable to localize to the chromatin, the DNA binding is important for the protein's function. The authors established a mouse model expressing a modified GATA2 protein that is degraded during mitosis and found that the homozygous progeny is unable to survive past definitive hematopoiesis and the embryos were lacking blood when analysed. The authors attributed the defects to the lack of GATA2 at mitotic exit and suggested that the daughter cells must be born with the GATA2 at key sites to promote hematopoiesis.

Overall the data is of high quality and the authors include a number of very nice controls that support their findings. However, there are a few places in the text where the authors overstate their conclusions. Thus, there are a few issues with the study that should be addressed prior to publication.

We would like to thank this reviewer for the helpful suggestions and for considering our manuscript of high quality including adequate controls to support our findings.

Main Points:

1. Fig. 1f: On page 4, lines 17-20, the authors state "To exclude bias derived from overexpression, we verified mitotic retention of endogenous TFs in K562 cells. As expected, the three TFs showed a similar retention profile as demonstrated in fibroblasts and HEK293T cells (Fig. 1e,f, Supplementary Fig. 1e)." However, Fig. 1f involves K562 cells overexpressing TFs, rather than endogenous. Referencing this subpanel along with Fig. 1e and the preceding sentences is misleading. It gives the impression to the reader that the data in Fig. 1f is from endogenously tagged TFs. The text should be changed to clearly identify which experiments include endogenous proteins and which include over-expressed proteins.

The presentation of the data was indeed confusing and we had restructured accordingly. We have revised the text and separated the findings with overexpression in HEK 293 T cells, fibroblasts and K562 cells (revised page 3 lines 20-25 and page 4, lines 1-7) from results with endogenous transcription factors in K562 cells by WB (revised page 4, lines 11-14). In addition, we have transferred the imaging data in K562 cells to revised Supplementary Fig. 1 to avoid misinterpretation and included new data with total endogenous protein and transcript expression in K562 (See Reviewer 1, point 1) in revised Fig. 1. We thank the reviewer for this suggestion that improved clarity.

2. Fig. 1g,h: Given the negative result, the authors should include blotting for the co-expressed proteins to ensure they are being expressed and whether they are in the cytoplasmic and chromatin fractions.

We agree with the reviewer's comment. We have now included WB analysis for the co-expressed proteins in each condition. Co-expressed proteins are localized in the same subcellular protein fraction as they were when expressed individually: GATA2 is present mainly in the chromatin-bound fraction, when compared to cytoplasmic fraction, and GFI1B and FOS are mainly present in cytoplasmic fraction. FOS was detected in conditions even when not overexpressed due to its low tissue specificity or activation by GATA2/GFI1B⁸. These data were added to Fig. 1i, j (below).

Fig. 1. GATA2 is retained at mitotic chromatin independently of cell context. i, j, Protein levels of GATA2, GFI1B and FOS in Cy and Chr fractions of mitotic HEK 293T cells when GFI1B (i) or FOS (j), were transduced individually or in combination with one or two additional TFs.

3. Fig. 2b: The Δ C-ZF mutant looks like it still binds Chr, but there is just less total protein than the other conditions. If the Δ C-ZF mutant was not able to bind Chr, then I would expect an increase in the protein in the Cytoplasmic fraction. In other words, I would have expected a reduction in the size of the band in the Chr fraction and a concomitant increase in the band in the Cytoplasmic fraction in the Δ C-ZF mutant condition (See Fig. 2d, C373R mutant for a perfect example of this). An alternative interpretation of the data in Fig. 2b is that rather than C-ZF being necessary for Chr binding, that the C-ZF is needed for protein stability. If there is less total Δ C-ZF-GATA2 protein in the cell compared to WT-GATA2, then the bands in the IP will appear weaker given equal ability to bind Chromatin. In order to distinguish between this alternative interpretation the authors could try a number of things. Can the authors provide a loading control to help normalize the bands between conditions? Can the authors include a darker exposure of this western blot so the reader can see if there is an increase in the size of the bands in the Cyto fraction? Or perhaps the authors can measure the turnover rate of each mutant to ensure they are all equally stable by doing a Cycloheximide time course experiment? Similar comments apply also to Fig. 2d.

Regarding the reviewer's concern about the overall low protein levels possibly being related with the loss of mitotic binding, we have performed WB with whole cell extracts for mCh fused Gata2 deletion constructs (included in revised Fig. 2b) and mutants (included in revised Fig 2d). We included loading controls in revised Supplementary Fig. 2a, b.

Since we do not observe less total protein in the C-ZF when compared to wild-type GATA2, we believe the results we obtain are due to C-ZF function in DNA binding and not in protein stability. We conclude that there is no correlation between absolute protein levels and mitotic bookmarking. For a more detailed answer, please see Reviewer 1, point 1.

In line with the reviewer's comment about expecting the Δ C-ZF to shift its localization from the chromatin-bound fractions to the cytoplasmic fraction, we have now looked at the soluble nucleus protein fraction, to investigate if protein could be more retained at the fraction (revised Fig 2b, d). We observed a reduction of GATA2 protein in soluble nucleus and chromatin-bound fractions in mitotic cells when compared to asynchronous cells, that is not accompanied by an increase in the cytoplasmic fraction in Δ C-ZF condition. However, we cannot rule out impact in protein stability looking at WB data alone. Nevertheless, the C-ZF is responsible for DNA binding⁹ and manipulating the C-ZF binding ability by single-point mutations revealed different retention kinetics from little to no impact in retention (R362Q and L359V), to severely impacted retention (R361L and C373R), which is consistent with publish EMSA data, where L359V shows increased DNA-binding affinity¹⁰, R362Q (AML) has modest to no impact in binding affinity and the Emberger syndrome mutations R361L and C373R result in little to no DNA binding^{11,12}. This kinetics is particularly obvious in imaging results (Fig. 2d and Supplementary Fig. 2c-j, below).

Supplementary Fig. 2. Chromatin retention of GATA2 is reduced by mutations in the C-terminal zinc finger (C-ZF). c-j, Live-cell images of HEK 293T cells overexpressing mCherry (mCh)-GATA2 (red) wild-type (WT) (c) and GATA2 proteins mutated in C-ZF in positions T354M (d), L359V (e), R361L (f), R362Q (g), C373R (h), R396Q (i) and R398W (j) in interphase (Inter) and mitosis (Pro – prophase, Meta – metaphase, Telo/Ck – Telophase/Cytokinesis). The first letter represents the wild-type amino acid, followed by the position and the replaced amino acid. DNA is marked by histone 2B (H2B)-mTurquoise (blue). Scale bars = 10 μ m. Mitotic events: n(GATA2)=219, n(T354M)=724, n(L359V)=78, n(R361L)=165, n(R362Q)=18, n(C373R)=360, n(R396Q)=151, n(R398W)=147.

Furthermore, we have attempted to perform a cycloheximide-chase assay, as per the reviewer's suggestion, with the mCherry fused GATA2 deletion constructs. For that, we have transfected HEK 293T cells with similar amounts of plasmids and evaluate protein half-life by flow cytometry. However, since our proteins are being overexpressed, this assay was suboptimal for detecting protein degradation. Notwithstanding, since we show that protein

abundance does not provide an explanation for mitotic retention kinetics, both when comparing across different transcription factors or mutated forms of GATA2 we find it unlikely that protein half-time would explain mitotic retention ability of each TF. We detailed the response to Reviewer 2 point 1.

4. Fig. 2c: The Δ C-ZF mutant overexpression has many foci in the cells in mitosis, which are not observed in interphase cells (as judged by the cells on the edge). The foci could be due to poor protein folding due to the deletion. Similar foci are observed in Fig. 2d for the C373R mutant, which the authors state in the Discussion section impacts protein folding. Could the authors comment on the appearance of the foci and whether this occurs in other cells in mitosis and whether these foci are due to poor protein folding?

Indeed, we observe protein foci which can be due to poor protein folding caused by deletion of the C-ZF or point mutations in that domain. Nevertheless, we also observe foci formation in several N-ZF mutations, namely R307W, A318T and R330Q, with no observable impact for mitotic retention (Supplementary Fig. 3, below). Therefore, protein-misfolding by itself does not explain reduction in mitotic binding. To avoid create unnecessary confusion to the reader we have deleted the sentence from discussion mentioning protein folding.

Supplementary Fig. 3. Chromatin retention of GATA2 is not impacted by mutations in the N-terminal zinc finger (N-ZF). a-f, Live-cell images of HEK 293T cells overexpressing mCherry (mCh)-GATA2 (red) proteins mutated in N-ZF in positions R293Q (a), P304H (b), R307W (c), A318T (d), L321F (e), R330Q (f) in interphase (Inter) and mitosis (Pro – prophase, Meta – metaphase, Telo/Ck – Telophase/Cytokinesis). The first letter represents the wild-type amino acid, followed by the position and the replaced amino acid. DNA is marked by histone 2B (H2B)-mTurquoise (blue). Scale bars = 10 μ m. Mitotic events: n(R293Q)=73, n(P304H)=151, n(R307W)=172, n(A318T)=296, n(L321F)=290, n(R330Q)=92.

5. Supplementary Fig. 5: The authors validated the destruction of mTurquoise signal when fused to the MD domain but then go on to fuse the MD domain to GATA2 for further experiments. It would be better to validate the MD-GATA2 protein so that they can show the exact timing of its degradation in mitosis. It is difficult to extrapolate the exact timing of protein degradation based on the FACS data. We suggest that the authors fuse the MD and the MDmut to the existing mCherry-GATA2 construct (or generate an mTurquoise-MD-GATA2 construct) and utilize live-cell imaging to demonstrate the loss of MD-GATA2 in mitosis throughout G1. This can replace the current Extended Fig. 5c as it is more relevant for their future conclusions.

We thank the reviewer for their comment. We have indeed cloned MD-GATA2-mTurquoise and MD_{mut}-GATA2-mTurquoise constructs and attempted to perform the suggested experiment. However, the constructs were not functional in our hands and require further optimization. Alternatively, we followed protein degradation before and after nocodazole arrest. Since reviewer 1 shared a similar concern, please see Reviewer 1, point 3, section (1), for a detailed answer.

6. Discussion, Page 11, lines 22-25: The authors state “Removal of the C-terminal zinc finger (C-ZF) of the DNA-binding domain abolished mitotic retention of GATA2, while deletion of the N-terminal region or the N-terminal zinc finger (N-ZF) of the DNA-binding domain did not impact mitotic retention”. The authors are presumably referencing the results from Fig. 2b,c for this statement. Binding to the DNA was not “abolished” by removal of the C-ZF. A band corresponding to the Chr bound fraction is clearly visible in Fig. 2b. Thus, the binding was not abolished, but rather reduced. The authors should consider using the word “reduced” to better describe the results in the Discussion.

Agreed. We have toned down our claim and substituted the word “abolished” with “reduced” or “reduction” in page 5, line 5 and 7, in page 12, line 22, and in the titles of Fig. 2 and Supplementary Fig.2. In addition, we have repeated WB analysis with cytoplasmic, soluble nucleus and chromatin bound fractions providing additional resolution. See Reviewer 1, point 1 for detailed response.

7. Discussion, Page 15, Line 9-10: The authors state “Overall, we have established the critical role of TFs at M-G1 transition for in vivo development”. The authors specifically test only for the role of GATA2 in in vivo development. While this study does include some data for Fos and GFI1B, those are all from in vitro experiments. Thus, I think it better to revise this

conclusion sentence to reflect that the authors only know about the role of a single TF *in vivo* from this study.

We agree with the reviewer and modified the sentence to the following: “Overall, we have established the critical role of GATA2 at M-G1 transition for *in vivo* hematopoietic development.” in page 16, line 8-9.

Minor points:

Fig. 1f: The GATA2 overexpression images lack the scale bar.

We thank the reviewer for spotting this. Fig. 1f was transferred to Supplementary Fig. 1e and a scale bar was added to the panels.

Fig. 4b: It would be a nice addition to include either here or in the Supplementary a similar table with the MDmut ratios.

We agree with the reviewer, but during the development of our project, MDmut-GATA2 mouse genotypes were not recorded the same way as the MD-GATA2 mice and in three months we wouldn't be able to address this comment to obtain a comparable table. However, we have confirmed major phenotypes in MDmut-GATA2 mice including the CFUs at E11.5, See Reviewer 1, point 3 section (2) for a detailed response.

Supplementary Fig. 5f: Its really hard to see which lines correspond to which conditions. I would suggest either color-coding the lines to the conditions, or make the shapes larger and more distinguishable.

Agreed. Previous Supplementary Fig. 5f is now Fig. 4c where the MD-GATA2 condition is shown in green to be more distinguishable.

References

1. Eich C, Arlt J, Vink CS, et al. *In vivo* single cell analysis reveals Gata2 dynamics in cells transitioning to hematopoietic fate. *J Exp Med*. 2018;215(1):233-248. doi:10.1084/jem.20170807
2. Minegishi N, Suzuki N, Kawatani Y, Shimizu R. Rapid turnover of GATA-2 via ubiquitin-proteasome protein degradation pathway. *Genes to Cells*. 2005;10(7):693-704. doi:10.1111/j.1365-2443.2005.00864.x
3. Wilson NK, Foster SD, Wang X, et al. Combinatorial transcriptional control in blood stem/progenitor cells: Genome-wide analysis of ten major transcriptional regulators. *Cell Stem Cell*. 2010;7(4):532-544. doi:10.1016/j.stem.2010.07.016
4. Pinello L, Xu J, Orkin SH, Yuan GC. Analysis of chromatin-state plasticity identifies cell-type-specific regulators of H3K27me3 patterns. *Proc Natl Acad Sci U S A*. 2014;111(3):E344–E353. doi:10.1073/pnas.1322570111
5. Pater E De, Kaimakis P, Vink CS, et al. Gata2 is required for HSC generation and survival. *J Exp Med*. 2013;210(13):2843-2850. doi:10.1084/jem.20130751
6. Rodrigues NP, Tipping AJ, Wang Z, Enver T. GATA-2 mediated regulation of normal hematopoietic stem/progenitor cell function, myelodysplasia and myeloid leukemia.

- Int J Biochem Cell Biol.* 2012;44(3):457-460. doi:10.1016/j.biocel.2011.12.004
7. Koga S, Yamaguchi N, Abe T, et al. Cell-cycle–dependent oscillation of GATA2 expression in hematopoietic cells. *Blood.* 2007;109(10):4200-4208. doi:10.1182/blood-2006-08-044149
 8. Gomes AM, Kurochkin I, Chang B, et al. Cooperative Transcription Factor Induction Mediates Hemogenic Reprogramming. *Cell Rep.* 2018;25(10):2821-2835. doi:10.1016/j.celrep.2018.11.032
 9. Yang HY, Evans T. Distinct roles for the two cGATA-1 finger domains. *Mol Cell Biol.* 1992;12(10):4562-4570.
 10. Zhang S, Ma L, Huang Q, et al. Gain-of-function mutation of GATA-2 in acute myeloid transformation of chronic myeloid leukemia. *PNAS.* 2008;105(6):2076–2081.
 11. Chong CE, Venugopal P, Stokes PH, et al. Differential effects on gene transcription and hematopoietic differentiation correlate with GATA2 mutant disease phenotypes. *Leukemia.* 2018;32(1):194-202. doi:10.1038/leu.2017.196
 12. Kazenwadel J, Betterman KL, Chong C, et al. GATA2 is required for lymphatic vessel valve development and maintenance. *J Clin Invest.* 2015;125(8):2979-2994. doi:10.1172/JCI78888DS1

REVIEWER COMMENTS

Reviewer #1 (Remarks to the Author):

The authors performed a number of new experiments and analysis to address all reviewer's comments. I believe the manuscript is significantly improved. Below are a few points that remain unaddressed or confusing.

1. Inconsistency between 1e and 1f. In Fig.1e, FOS appears to be expressed at drastically lower levels (both in cytoplasmic and chromatin extracts) compared to the other endogenous TFs (consistent with the lower mRNA levels shown in 1g), however its expression is much higher in whole-extracts in fig. 1f, although the same loading control is used (CANX). How do the authors explain the difference?

2. It is great that now the authors include Western blot analyses from all cellular fractions for each of the GATA2 mutants. The results are indeed convincing that the total levels of overexpression do not associate with the relative chromatin binding of each factor, so the differences likely reflect true biological effects. However, these results also show clearly that the main effect of each mutation is on the relative fraction of "soluble" versus "chromatin-bound" GATA2 both in asynchronous and mitotic cells. To support conclusions about specific effects on mitotic retention, a quantitation of the amounts of each factor on mitotic vs asynchronous chromatin extracts (after normalization with their respective H3) based on several WB like the one in Fig.2d would be more convincing. That becomes more critical in light of their current Suppl.Fig 2a-b, where all mutants are seemingly more retained on the mitotic chromatin fractions compared to the asynchronous, which is confusing. Again, careful quantitation will likely solve these inconsistencies.

3. The bioinformatics analysis of the GATA2 ChIP-seq data is significantly improved. A few minor comments there: (i) Why does the C3 cluster disappear from all downstream analysis? Is it replaced by the "asyn-only" peaks? The authors should clarify the terms and definitions. (ii) The gene ontology is not that informative and can be moved to the suppl. (iii) The motif analysis on the other side could be presented in an enrichment dotplot format (as they currently do with the GO) to allow direct comparison of the relative enrichments/pvalues of top enriched motifs across categories. (iv) the coloring at the heatmap of Fig3h should be inverted (red-blue scale) or a different title should be provided. As it reads now (Mit vs asyn), it suggests that mitotic peaks are enriched in all these marks and not the opposite.

4. For the in vivo experiments, I am very happy with all revisions which included the MDmut strain for all key comparisons. I also understand the challenges of performing further characterizations of the MD system in the embryos, so the in vitro characterization of the degradation/recovery kinetics should suffice. However, I strongly recommend adding in Fig 4a, an earlier timepoint after mitotic release (e.g 30m-1 hour) to show that indeed most of GATA2 is successfully degraded and then recovers by 6hr. This experiment should be straightforward in 293T. Regarding the limitation of this system, the authors should also highlight that given that some degree of degradation persists for several hours beyond mitotic exit, some of the observed effects might be due to its role in G1 and not its mitotic bookmarking function.

Reviewer #2 (Remarks to the Author):

The authors have comprehensively addressed the issues raised by this reviewer. The addition of the requested information and new results has improved the manuscript and the value of this study.

Reviewer #3 (Remarks to the Author):

The authors have addressed most of my concerns. However, I still have some issues with the response to points 3 and 5.

In Figures 2b,d it is unclear which condition the WCE blot are associated with. Are these WCE from the Asynchronous or the mitotic arrested cells? Ideally the authors should include both. The authors repeatedly cut up the western blot bands and re-assemble them in their figures making it impossible to compare bands between these cut up sections. For example, the newly included WCE bands are all presented as separate crops. I understand the motivation was likely to present the data in a more logical and organized way, but the drawback of this approach is that we cannot then compare the intensity of the bands between these separately cropped sections. The authors could have run these samples on different gels or exposed the blots differently. Thus, as currently presented, this new data does not adequately address our original point that the Δ C-ZF mutant may be less stable than the full length GATA2 protein. Typically, if the authors would like the reader to compare the intensity of bands on a western blot, they should be presented uncropped and all on the same western. The authors should show the entire blots, unaltered.

Furthermore, in Figure 4d, there is no WCE for the GATA2 control. Why is this missing? It should also be included in this figure.

The authors rebuttal about the relative distribution of the Δ C-ZF redistributing from the chromatin bound fraction to either the soluble nucleus fraction or the cytoplasm also does not address our original point. The authors would like us to compare the intensity of the bands between the Mit and the Asynchronous samples for the Δ C-ZF, but they are cropped and separated. We cannot compare the intensity of bands between two separate western blots, and the data is presented in such a way that it is impossible to know if these samples were run on the same or different western blots. To better address the point that there is a re-distribution from chromatin to either the cytoplasm or the soluble nucleus fraction, the authors could quantify the bands and plot the ratio of Chr:SN:Cy within each western blot. This would reveal the relative changes within each condition that could then be compared across blots, and would have the added benefit of averaging and statistical analysis across multiple repeats.

In new Figure 4a the author's first time point is 4 hours after nocodazole release. How do we know the protein was degraded at anaphase or the "M-G1 transition"? The authors should co-blot for cyclin B for example as a marker for mitosis/cell cycle progression. This way we can assess if the GATA2 is indeed degraded during mitosis, thus disrupting the mitotic bookmarking.

Furthermore, Figure 4a suffers from the same issue as mentioned above that the bands we are meant to compare (No release vs 4/6hr release) are cropped separately and it is not clear if they were run on the same western blots or not.

GATA2 mitotic bookmarking is required for definitive haematopoiesis

Reviewer #1 (Remarks to the Author):

The authors performed a number of new experiments and analysis to address all reviewer's comments. I believe the manuscript is significantly improved. Below are a few points that remain unaddressed or confusing.

1. Inconsistency between 1e and 1f. In Fig.1e, FOS appears to be expressed at drastically lower levels (both in cytoplasmic and chromatin extracts) compared to the other endogenous TFs (consistent with the lower mRNA levels shown in 1g), however its expression is much higher in whole-extracts in fig. 1f, although the same loading control is used (CANX). How do the authors explain the difference?

We thank the reviewer for recognizing the improvements on the manuscript and our efforts to address the concerns of the reviewers.

To clarify this point, we have repeated WBs for FOS with new mitotic enriched K562 sorted samples and WCEs. In agreement with WCEs, we have detected stronger expression of FOS in the cytoplasmic fraction of mitotic K562 cells. Although FOS transcriptional output is low in K562 cells, that is not accompanied by protein levels, indicating post-transcriptional regulation. We have included the new data in revised Figure 1e and Figure 1f (below). Loading controls in Figure 1e and 1f were obtained using the same exposure times.

Fig. 1e, Endogenous expression of GATA2, GFI1B and FOS in cytoplasmic and chromatin-bound fractions of mitotic K562 cells. H3 and CANX were used as loading controls.

Fig. 1f, Protein expression from asynchronous HDFs (overexpressed TFs, right panel) and K562 (endogenous TFs, left panel) whole-cell extracts. HDFs were transduced with mCh-TFs and the same number of mCherry+ cells were FACS sorted for western blotting. CANX and Actin were used as loading controls.

2. It is great that now the authors include Western blot analyses from all cellular fractions for each of the GATA2 mutants. The results are indeed convincing that the total levels of overexpression do not associate with the relative chromatin binding of each factor, so the differences likely reflect true biological effects. However, these results also show clearly that the main effect of each mutation is on the relative fraction of “soluble” versus “chromatin-bound” GATA2 both in asynchronous and mitotic cells. To support conclusions about specific effects on mitotic retention, a quantitation of the amounts of each factor on mitotic vs asynchronous chromatin extracts (after normalization with their respective H3) based on several WB like the one in Fig.2d would be more convincing. That becomes more critical in light of their current Suppl.Fig 2a-b, where all mutants are seemingly more retained on the mitotic chromatin fractions compared to the asynchronous, which is confusing. Again, careful quantitation will likely solve these inconsistencies.

As suggested by the reviewer, we have now quantified the chromatin-bound fractions of asynchronous and mitotic cells based on 3-5 western blots and normalization with H3 controls. To focus the analysis on the main result that DNA binding is important to mitotic chromatin retention, we performed statistical analysis on the comparison of the mitotic chromatin-bound fractions between GATA2 full length and deletions (revised Fig. 2c and 2d) and mutations (revised Fig. 2e and 2f).

Quantification of chromatin-associated GATA2 by western blotting revealed decreased GATA2 chromatin binding in both mitotic and asynchronous cells for GATA2 mutations that reduce DNA binding, namely T354M, R361L and C373R (Fig. 2e, f, Supplementary Fig. 2). However, in addition to DNA-binding perturbations, mutations in GATA2 may introduce complex conformational changes and modifications in protein stability making it difficult to reveal mitosis specific impacts with this approach. We have introduced a sentence in the results section in this regard page 5, in lines 22-24. Nevertheless, our conclusion that DNA binding is required for mitotic retention is well supported by the data regardless of the impact of individual mutations in interphase.

Below we include revised Fig. 2, including WB quantifications with statistical analysis comparing the mitotic chromatin retention between GATA2 full length and each deletion/mutant.

Fig. 2. Point mutations in GATA2 C-terminal zinc finger associated with acute myeloid leukaemia and Emberger syndrome reduce mitotic retention. **a**, Representation of GATA2 protein domains highlighting leukaemia and Emberger syndrome (ES)-associated point mutations in the N- and C- terminal zinc fingers (ZF) of the DNA-binding domain (DBD). TAD – transactivation domain. NRD – negative regulatory domain. NLS – nuclear localization signal. CML – chronic myeloid leukaemia. AML – acute myeloid leukaemia. MDS – myelodysplastic syndrome. **b**, Live-cell images of HEK 293T cells overexpressing mCherry (mCh)-GATA2 deletion (Δ) constructs (in red) excluding the N-terminal (amino acids (aa) 1-235), N-ZF (aa 287-342), C-ZF (aa 243-379) or NLS (aa 380-440) in interphase (Inter) and metaphase (Meta). Mitotic events: $n(\Delta\text{N-Terminal})=395$, $n(\Delta\text{N-ZF})=375$, $n(\Delta\text{C-ZF})=100$, $n(\Delta\text{NLS})=52$. **c**, GATA2 protein expression in whole-cell extracts (WCE), cytoplasmic (Cy), soluble nucleus (SN) and chromatin-bound (Chr) fractions of both asynchronous (A, Async) and mitotic (M, Mit) HEK 293T cells overexpressing deletion constructs. Representative blots acquired with the same exposure time for asynchronous and mitotic cells are shown. **d**, Western blot quantifications of GATA2 deletion constructs at the chromatin-bound fraction in asynchronous (A) and mitotic (M) cells after normalizing to histone 3 (H3). Comparison of mitotic chromatin retention between GATA2 and each mutant is shown. $N=3-5$. **e**, Levels of mCherry-fused GATA2 C-ZF mutant proteins in the Cy, SN and Chr protein fractions of asynchronous and mitotic HEK 293T cells. A representative cell in metaphase expressing indicated mCherry-GATA2 mutant is shown (right). Histone 2B (H2B)-mTurquoise signal (blue) indicates DNA content. Scale bars = 10 μm . **f**, Western blot quantification of GATA2 mutants at the chromatin-bound fraction in asynchronous (A) and mitotic (M) cells after normalizing to histone 3 (H3). Comparison of mitotic chromatin retention between GATA2 and each mutant is shown. $N=3-4$. **g**, Efficiency of hemogenic reprogramming of human dermal fibroblasts (% of CD9+CD49f+ cells) with GATA2 or mutants (L359V or C373R), plus GF11B and FOS. M2rtTA (M2) was used as control. $n(\text{M2})=10$, $n(\text{GATA2})=13$, $n(\text{L359V})=11$, $n(\text{C373R})=11$. Mean \pm SD is represented. Statistical significance for

western blot quantification (d, f) and hemogenic reprogramming (g) was analysed by two-way ANOVA followed by Fisher's LSD test and one-way ANOVA followed by Bonferroni's test, respectively. * $p < 0.05$, ** $p < 0.01$, *** $p < 0.001$, ns- non-significant. See Supplementary Fig. 2 for loading controls.

3. The bioinformatics analysis of the GATA2 ChIP-seq data is significantly improved. A few minor comments there:

- (i) Why does the C3 cluster disappear from all downstream analysis? Is it replaced by the “asyn-only” peaks? The authors should clarify the terms and definitions.

We would like to thank the reviewer for noting analyses related to Cluster 3. Indeed, Cluster 3 accounts for only 3% of overlapped peaks (44 peaks) between mitotic and asynchronous cells. Therefore, we only included analysis of cluster 3 in panel C of Figure 3 and removed it from analyses thereafter. We also thank the reviewer for noting that the definitions of “Async” and “Async only” were not clear. “Async” peaks take all GATA2 peaks in asynchronous cells in consideration, including bookmarked peaks. “Async only” peaks include only non-bookmarked GATA2 peaks in asynchronous cells (excluding bookmarked sites). By subtracting the mitotic peaks from GATA2 peaks in asynchronous cells we highlight the features of bookmarked sites. The definitions of “Async” and “Async only” are now clarified in Figure 3 legend (see below). To make the figure consistent throughout, we have used “Async only” peaks in all analysis except from Figure 3c and modified Figure 3a to include the number of non-bookmarked (Async only) peaks and genes. We also updated Supplementary Figure 4f and g to include Async only peaks. In addition, to improve readability, we have re-ordered the panels and modified the Results section. We believe that this Results section and our terms and definitions are now clear. Please find Figure 3 and Supplementary Figure 4 below. Modifications to the main text can be found under the results section “GATA2 bookmarks key HSPC regulators”, in pages 6 to 8 (in red).

- (ii) (ii) The gene ontology is not that informative and can be moved to the suppl.

We agreed with the reviewer's suggestion and moved gene ontology analysis to the revised Supplementary Figure 4 panel F (see below).

- (iii) (iii) The motif analysis on the other side could be presented in an enrichment dotplot format (as they currently do with the GO) to allow direct comparison of the relative enrichments/pvalues of top enriched motifs across categories.

We have now presented the motif analysis as an enrichment dot plot as suggested. This panel was included as revised Figure 3e and can be found below. We have also updated the text in the results section in page 7, lines 7-9 to “In agreement, motif discovery analysis revealed that mitotic peaks were enriched for GATA, RUNX, AP-1 and ETS motifs (Fig. 3e), suggesting that GATA2 cooperates in mitosis with hematopoietic regulators.”.

(iv) (iv) the coloring at the heatmap of Fig3h should be inverted (red-blue scale) or a different title should be provided. As it reads now (Mit vs asyn), it suggests that mitotic peaks are enriched in all these marks and not the opposite.

We agree with the reviewer and have inverted the colors to improve readability. The revised heatmap can be found in Figure 3i below.

Fig. 3. GATA2 bookmarks a subset of interphasic target genes with roles in definitive haematopoiesis. **a**, Venn diagram showing the number of ChIP-seq GATA2 peaks and genes shared between asynchronous (Async) and mitotic (Mit) K562 cells. **Async only** refers to non-bookmarked peaks and genes in asynchronous cells. **b**, Gene tracks for GATA2 binding sites at *GATA2* and *RUNX1* loci showing both bookmarked (grey) and asynchronous unique peaks. Kb – kilobase. **c**, K-means clustering of asynchronous (left) and mitotic peaks (right). The percentage of mitotic peaks overlapping with asynchronous peaks in each cluster is shown. The 42 mitotic-unique peaks are not shown. **d**, Number of GATA2 motifs in Async only peaks, mitotic peaks and mitotic clusters 1 (C1) and 2 (C2). **e**, Motif discovery analysis for GATA2 target sites per group of peaks. Coloured scale represents the adjusted *p*-value and the circle size the percentage of peaks containing a particular motif. **f**, Percentage of GATA2 peaks where motifs for relevant HSPCs regulators are also present. **g**, Percentage of overlap between GATA2 peaks and peaks for HSPC regulators from available ChIP-seq datasets. **h**, Chromatin-state enrichment heatmap representing the percentage of genome occupancy of GATA2 per group of peaks. Scale represents the percentage of peaks at each genomic segment. TSS – Transcription start site. **i**, Integration heatmap with histones marks, DNase-seq and ATAC-seq data for K562 cells (ENCODE). Scale represents the accumulated sum differences across bins between **Async only and mitotic peaks and clusters**. **j**, Histone marks and ATAC-seq profiles at peak summit (centre).

Supplementary Fig. 4. Sorting strategy for Chromatin Immunoprecipitation followed by sequencing (ChIP-seq) of mitotic K562 cells and complementary analyses. **a**, Outline of mitotic K562 fixation and FACS sorting steps. K562 were arrested with 0.2 μ g/mL nocodazole for 12-14h and fixed with 2mM Di(N-succinimidyl)

glutarate (DSG) before fixation with 1% formaldehyde (FA). After that, cells were permeabilized and stained with anti-phospho-Ser/Thr-Pro MPM-2 primary antibody and anti-mouse AF647 secondary antibody. MPM-2 antibody recognizes several phosphorylated proteins during mitosis. Cells positive for MPM-2 were FACS sorted, washed and cell pellets were snap-frozen for ChIP. **b**, Quantification of interphasic and mitotic K562 cells after double fixation. Mitotic cells were identified in the 4N peak according to propidium iodide (PI) staining. **c**, Quantification of interphasic and mitotic K562 cells after nocodazole treatment and double fixation. **d**, Gating strategy to sort MPM-2 positive nocodazole treated K562 cells after double fixation. Mitotic cell purity after sorting is shown. **e**, Gene tracks for GATA2 binding sites at *CD9* locus. Bookmarked sites are highlighted in grey. **f**, Gene Ontology (GO) biological processes (BP) for the top 1,000 gene-related peaks in non-bookmarked genes in asynchronous cells (Async only), mitotic (Mit) and mitotic clusters 1 (C1) and 2 (C2). Categories that contain more than 5 peaks per category are displayed. Coloured scale represents the adjusted *p*-value and the circle size the number of peaks per group. **g**, Gene body distribution of Mit and Async only GATA2 binding sites.

4. For the *in vivo* experiments, I am very happy with all revisions which included the MDmut strain for all key comparisons. I also understand the challenges of performing further characterizations of the MD system in the embryos, so the *in vitro* characterization of the degradation/recovery kinetics should suffice. However, I strongly recommend adding in Fig 4a, an earlier timepoint after mitotic release (e.g 30m-1hour) to show that indeed most of GATA2 is successfully degraded and then recovers by 6hr. This experiment should be straightforward in 293T. Regarding the limitation of this system, the authors should also highlight that given that some degree of degradation persists for several hours beyond mitotic exit, some of the observed effects might be due to its role in G1 and not its mitotic bookmarking function.

We thank the reviewer for acknowledging the challenges of assessing degron-mediated degradation *in vivo*. We agree with the reviewer that an early timepoint would support the characterization of degradation kinetics *in vitro*. We have performed additional experiments in this regard including early timepoints (1 and 2 hours). As we did not find differences in GATA2 degradation between 1 and 2 hours we chose to include the 2-hour timepoint in the revised manuscript to keep constant the interval between collections (2-hours). It is now clear that in our system GATA2 degradation is most efficient at 4 hours. At 2 hours GATA2 is not yet degraded and show recovery by 6 hours. We have also included the blot for Cyclin B1 using the same samples, as requested by Reviewer 3. The updated panel can be found in Fig. 4a and below.

We agree with the reviewer that degradation spillover to G1 is a limitation of the system and have added the following sentence to the discussion in page 16, lines 4-6: “This characterization is important given that some degree of GATA2 degradation may persist for several hours beyond mitotic exit potentially interfering with GATA2 function in G1.”

Fig. 4a. Western blot analysis of GATA2 and Cyclin B1 proteins in HEK 293T cells expressing GATA2 fused to the mitotic degradation (MD) domain of cyclin B1 or an inactive form (MD_{mut}) before (0h) and 2, 4 and 6 hours (h) after release from nocodazole arrest. Actin was included as loading control. Async – Asynchronous cells.

Reviewer #3 (Remarks to the Author):

The authors have addressed most of my concerns. However, I still have some issues with the response to points 3 and 5.

In Figures 2b,d it is unclear which condition the WCE blot are associated with. Are these WCE from the Asynchronous or the mitotic arrested cells? Ideally the authors should include both. The authors repeatedly cut up the western blot bands and re-assemble them in their figures making it impossible to compare bands between these cut up sections. For example, the newly included WCE bands are all presented as separate crops. I understand the motivation was likely to present the data in a more logical and organized way, but the drawback of this approach is that we cannot then compare the intensity of the bands between these separately cropped sections. The authors could have run these samples on different gels or exposed the blots differently. Thus, as currently presented, this new data does not adequately address our original point that the Δ C-ZF mutant may be less stable than the full length GATA2 protein. Typically, if the authors would like the reader to compare the intensity of bands on a western blot, they should be presented uncropped and all on the same western. The authors should show the entire blots, unaltered.

We thank the reviewer for spotting these issues. Indeed, in the previous version of the manuscript WCE blots were performed with asynchronous cells only. In the revised version of the manuscript, we show WCE of both asynchronous and mitotic cells. Please see revised Figure 2 and Reviewer 1, point 2 for additional details. The WCE blots for GATA2 deletions were acquired all in the same gel. WCE blots for GATA2 mutations were run in two gels and acquired at the same time. We have included the uncut gels for WCE of GATA2 deletions and mutations below.

Rebuttal Figure 1. WB analysis of WCE from asynchronous (A) and mitotic (M) HEK 293T overexpressing GATA2 deletion constructs. Membranes were cut and incubated with mCherry antibody (1:1000) overnight.

Rebuttal Figure 2. WB analysis of WCE from asynchronous and mitotic HEK 293T overexpressing GATA2 deletion constructs (same gel as above). Membranes were cut and incubated with Actin antibody (1:2000) overnight.

Rebuttal Figure 3. WB analysis of WCE from asynchronous (A) and mitotic (M) HEK 293T overexpressing GATA2 mutations. Membranes were cut and incubated with mCherry antibody (1:1000) overnight.

Rebuttal Figure 4. WB analysis of WCE from asynchronous and mitotic HEK 293T overexpressing GATA2 mutations (same gels as in the left panel). Membranes were cut and incubated with Actin antibody (1:2000) overnight.

We also agree with the reviewer that contiguous bands are the best way to compare mitotic and asynchronous conditions. In the revised manuscript, asynchronous samples are always shown next to mitotic samples for the same deletion/mutation. We have added the sentence “Representative blots acquired with the same exposure time for asynchronous and mitotic cells are shown” to Figure 2 legend to be more explicit that blots from the same experiment were acquired with the same exposure times.

When comparing the total levels of protein in WCE between asynchronous and mitotic cells we found a reduction of total protein levels in mitosis, however, the mitotic protein levels of the Δ C-ZF mutant levels are comparable to the mitotic levels of GATA2 full length. Therefore, protein levels do not provide an adequate explanation for reduced GATA2 retention seen in the chromatin fraction of Δ C-ZF and imaging. Regarding protein stability, we have performed cycloheximide chase in HEK 293T cells overexpressing mCh-Gata2 or the mCh-C-ZF deletion construct and did not observe increased protein degradation of the C-ZF construct, suggesting that protein stability is not being affected by the removal of the C-ZF (Rebuttal Figure 5 and 6). Nevertheless, we cannot exclude differences in protein stability and half-life as result of point mutations in GATA2 protein. However, we believe that our main conclusion that DNA binding is important to mitotic chromatin retention is still supported by the data. We have included a sentence in the results to highlight this in page 5, lines 22-24: “In addition to DNA-binding perturbations, mutations in GATA2 may introduce complex conformational changes and modification in protein stability making it difficult to reveal mitosis specific impacts with this approach.”.

Rebuttal Figure 5. Cycloheximide chase. HEK 293T cells overexpressing mCh-Gata2 (up) or mCh-C-ZF deletion construct (down) were incubated with 10ug/mL cycloheximide for 2, 4, 6 and 8 hours (h). 0h indicate the untreated control.

Rebuttal Figure 6. Actin control for mCh-Gata2 (up) and mCh-C-ZF (down) transduced cells after cycloheximide chase.

Furthermore, in Figure 2d, there is no WCE for the GATA2 control. Why is this missing? It should also be included in this figure.

Agreed. We have now included WCE of both asynchronous and mitotic samples for GATA2 control as well as mutations in revised Fig. 2e.

The authors rebuttal about the relative distribution of the Δ C-ZF redistributing from the chromatin bound fraction to either the soluble nucleus fraction or the cytoplasm also does not address our original point. The authors would like us to compare the intensity of the bands between the Mit and the Asynchronous samples for the Δ C-ZF, but they are cropped and separated. We cannot compare the intensity of bands between two separate western blots, and the data is presented in such a way that it is impossible to know if these samples were run on the same or different western blots.

To better address the point that there is a re-distribution from chromatin to either the cytoplasm or the soluble nucleus fraction, the authors could quantify the bands and plot the ratio of Chr:SN:Cy within each western blot. This would reveal the relative changes within each condition that could then be compared across blots, and would have the added benefit of averaging and statistical analysis across multiple repeats.

We agree with the reviewer that quantification of bands across multiple WBs are necessary to better compare asynchronous and mitotic samples and each condition to GATA2 full length. Reviewer 1 had a similar comment with a focus on chromatin retention after normalization to H3. In the revised manuscript (see additional details in response to Reviewer's 1 point 2), we

have quantified the bands and divided the asynchronous and mitotic (mCherry bands) chromatin (Chr) values by their respective H3 value, which we plotted on revised panel Fig. 2d (deletions) and revised Fig. 2f (mutants).

We also followed the suggestion of the reviewer and quantified the bands and plotted the ratio of Chr/SN, but excluded the Cytoplasm (Cy) fraction due to the very low GATA2 protein levels. However, we have chosen to include in the revised manuscript the Chromatin-bound GATA2 levels instead of the Chr/SN because we could normalize the Chromatin values to H3 (H3 was not detected in SN fractions), making it a fairer comparison method. We, nevertheless, include the plotted ratios below in the Rebuttal Figure 7 and 8.

Please find the detailed reply to Reviewer 1's comment number 2, together with revised Fig. 2. We are including Supplementary Fig. 2 with loading controls below.

Rebuttal Figure 7. Quantification of the ratio between mCherry-GATA2 signal of chromatin-bound fraction (Chr) and soluble nucleus (SN) in asynchronous (A) and mitotic (M) cells overexpressing deletion constructs. Comparison between asynchronous and mitotic cells is shown. Statistical significance was analyzed by two-way ANOVA followed by Fisher's LSD test. ** $p < 0.01$, *** $p < 0.001$, ns - non-significant.

Rebuttal Figure 8. Quantification of the ratio between mCherry-GATA2 signal of chromatin-bound fraction (Chr) and soluble nucleus (SN) in asynchronous (A) and mitotic (M) cells overexpressing GATA2 mutants. Comparison between asynchronous and mitotic cells is shown. Statistical significance was analyzed by two-way ANOVA followed by Fisher's LSD test. ** $p < 0.01$, ns - non-significant.

Overall, we show for deletions and point mutants that protein levels do not provide an explanation for the loss of chromatin retention. For example, R362Q shows similar protein levels in WCE extracts compared to R398W, but very different retention abilities. Moreover, live-cell imaging shows clear protein expression for all mutants and cycloheximide chase does not show increased protein degradation for the C-ZF deletion providing less support for a role of protein stability in chromatin retention.

We have revised the text related to the updated Fig 2. and Supplementary Fig. 2 to accommodate the comments from both reviewers under the results section “GATA2 mitotic retention requires DNA binding”, from page 4 to page 6 (in red).

Supplementary Fig. 2. Chromatin retention of GATA2 is reduced by mutations in the C-terminal zinc finger (C-ZF). **a, b**, Western blot analysis of actin and histone 3 loading controls for mCherry-GATA2 deletion constructs (**a**) and mutant proteins (**b**) in whole-cell extracts (WCE) and protein fractionations of asynchronous

(A, Async) and mitotic (M, Mit) HEK 293T cells. Bands were acquired using the same exposure times for asynchronous and mitotic cells, depending on the antibody and protein isolation method. Cy, cytoplasmic protein fraction. SN, soluble nucleus protein fraction. Chr, chromatin-bound protein fraction. **c-j**, Live-cell images of HEK 293T cells overexpressing mCherry (mCh)-GATA2 (red) wild-type (WT) (**c**) and GATA2 proteins mutated in C-ZF in positions T354M (**d**), L359V (**e**), R361L (**f**), R362Q (**g**), C373R (**h**), R396Q (**i**) and R398W (**j**) in interphase (Inter) and mitosis (Pro – prophase, Meta – metaphase, Telo/Ck – Telophase/Cytokinesis). The first letter represents the wild-type amino acid, followed by the position and the replaced amino acid. DNA is marked by histone 2B (H2B)-mTurquoise (blue). Scale bars = 10 μ m. Mitotic events: n(GATA2)=219, n(T354M)=724, n(L359V)=78, n(R361L)=165, n(R362Q)=18, n(C373R)=360, n(R396Q)=151, n(R398W)=147.

In new Figure 4a the author's first time point is 4 hours after nocodazole release. How do we know the protein was degraded at anaphase or the "M-G1 transition"? The authors should co-blot for cyclin B for example as a marker for mitosis/cell cycle progression. This way we can assess if the GATA2 is indeed degraded during mitosis, thus disrupting the mitotic bookmarking. Furthermore, Figure 4a suffers from the same issue as mentioned above that the bands we are meant to compare (No release vs 4/6hr release) are cropped separately and it is not clear if they were run on the same western blots or not.

We thank the reviewer for the helpful suggestions. Our initial data regarding the functionality of the degron using HEK 293T FUCCI line (Supplementary Fig. 5) shows degradation of MD-turquoise fluorescent protein at M-G1. Nevertheless, we have followed the reviewer's suggestion and checked Cyclin B1 expression before and after Nocodazole release as shown in revised Fig. 4a. Indeed, Cyclin B1 is degraded upon mitotic exit (compare no release (0h) and 2 h after nocodazole release) and is recovered in asynchronous cells. The degradation of cyclin B1 is consistent to what has been reported for U2OS cells¹, which have a similar cycling time to HEK 293T. In addition, the timing of GATA2 degradation (maximum degradation between 2-4hours) is also in a similar range of what previously published using the Cyclin B1-degron for GATA1 in an erythroblast cell line². We have also included an earlier time point (2h), as suggested by Reviewer 1, include blots for Cyclin B1 and displayed bands in a single blot. We also included a sentence in the results section regarding this issue in page 8, lines 16.18 as follows. "Cyclin B1 levels were highest before nocodazole release, in pro-metaphase, prior to degradation by the anaphase-promoting complex¹¹, and quickly reduced after release". Please see Reviewer 1 point 4 for additional details.

References:

1. Clemm von Hohenberg K, Müller S, et al. *Nat Commun.* 2022.
2. Kadauke S, Udugama MI, et al. *Cell.* 2012.

REVIEWERS' COMMENTS

Reviewer #1 (Remarks to the Author):

The authors performed many new experiments, analysis and textual changes to address the remaining concerns. All new figures are significantly improved. In my opinion, the study is ready for publication.

On minor comment is that the motif analysis presented in the revised 3e is unfortunately, relative uninformative since it shows exactly the same factors and same enrichments across all different peak categories. Since, this is the case, the authors could replace it by just presenting the most significant motifs for the bookmarked peaks, or moving it to the supplement.

Reviewer #3 (Remarks to the Author):

I thank the authors for their efforts to address the points I raised in the revision. They have successfully addressed most of my concerns, but I still have one remaining issue related to the point I raised about Gata2 degradation in Figure 4a.

The new data in Figure 4a now includes Cyclin B1 as a positive control for mitotic degradation. As currently shown, Cyclin B1 is degraded almost immediately after Nocodazole release (sometime between 0 and 2 hours) but the Gata2-MD domain is not degraded until sometime between 2 to 4 hours after nocodazole release. There is no difference in the GATA2 bands at 2 hours compared to the 0h condition, especially considering the lower actin loading control for the 2 hour time point. Thus, it appears Gata2-MD is not being degraded during Mitosis nor the M/G1 transition but likely in the middle of G1 phase. The authors new edits of the Discussion section (Page 16, line 4-6; in red) are helpful in addressing this issue. However, I would recommend the authors remove references to "GATA2-MD degradation the M-G1 transition" and instead refer to it as "GATA2-MD degradation in G1 phase." In addition, there are two specific places in the text I would like to recommend text edits to better describe their results:

1. Page 8, line 12-21: I would recommend the authors change the text to "We assessed protein levels of MD- or MDmut-GATA2 constructs before and after release from nocodazole arrest, using degradation of Cyclin B1 as a control for mitotic degradation. While Cyclin B1 was degraded immediately upon nocodazole release, we observed induced GATA2 protein degradation over time, with highest impact on MD-GATA2 degradation at 4 hours, increased protein levels after 6 hours and returning to MDmut-GATA2 levels in asynchronous cells (Fig. 4a). For better resolution of protein degradation during specific phases of the cell cycle, we fused the MD or MDmut domains to an mTurquoise fluorescent protein and observed mTurquoise signal reduction by flow cytometry in G1 phase (Supplementary Fig. 5a-c).

2. Page 16, line 4-6: I would change the text to "This characterization is important given that GATA2 is degraded several hours after other mitotically degraded proteins and is likely occurring beyond mitotic exit, potentially interfering with GATA2 function in G1 phase"

REVIEWERS' COMMENTS

Reviewer #1 (Remarks to the Author):

The authors performed many new experiments, analysis and textual changes to address the remaining concerns. All new figures are significantly improved. In my opinion, the study is ready for publication.

On minor comment is that the motif analysis presented in the revised 3e is unfortunately, relative uninformative since it shows exactly the same factors and same enrichments across all different peak categories. Since, this is the case, the authors could replace it by just presenting the most significant motifs for the bookmarked peaks, or moving it to the supplement.

We thank the reviewer for considering our study ready for publication. According to the reviewer's suggestion, we have modified Figure 3e to show the most relevant motifs for bookmarked peaks. The comparative analysis with asynchronous peaks, mitotic peaks and clusters was moved to revised supplementary Figure 4g. Modifications to the text and figure legends were introduced accordingly.

e

Name	Motif	p-value
GATA6		1e-370
SPIC		1e-33
RUNX1		1e-31
TBX20		1e-19
MEIS1		1e-16
SMAD2		1e-15
NEUROD1		1e-13
RUNX		1e-13
FOXP1		1e12
BCL11A		1e-12

Figure 3e, *De novo* motif enrichment analysis for GATA2 mitotic bookmarked target sites. Top ten motifs are shown with respective *p*-values.

Reviewer #3 (Remarks to the Author):

I thank the authors for their efforts to address the points I raised in the revision. They have successfully addressed most of my concerns, but I still have one remaining issue related to the point I raised about Gata2 degradation in Figure 4a.

The new data in Figure 4a now includes Cyclin B1 as a positive control for mitotic degradation. As currently shown, Cyclin B1 is degraded almost immediately after Nocodazole release (sometime between 0 and 2 hours) but the Gata2-MD domain is not degraded until sometime between 2 to 4 hours after nocodazole release. There is no difference in the GATA2 bands at 2 hours compared to the 0h condition, especially considering the lower actin loading control for the 2 hour time point. Thus, it appears Gata2-MD is not being degraded during Mitosis nor the M/G1 transition but likely in the middle of G1 phase. The authors new edits of the Discussion section (Page 16, line 4-6; in red) are helpful in addressing this issue. However, I would recommend the authors remove references to “GATA2-MD degradation the M-G1 transition” and instead refer to it as “GATA2-MD degradation in G1 phase.” In addition, there are two specific places in the text I would like to recommend text edits to better describe their results:

1. Page 8, line 12-21: I would recommend the authors change the text to “We assessed protein levels of MD- or MDmut-GATA2 constructs before and after release from nocodazole arrest, using degradation of Cyclin B1 as a control for mitotic degradation. While Cyclin B1 was degraded immediately upon nocodazole release, we observed induced GATA2 protein degradation over time, with highest impact on MD-GATA2 degradation at 4 hours, increased protein levels after 6 hours and returning to MDmut-GATA2 levels in asynchronous cells (Fig. 4a). For better resolution of protein degradation during specific phases of the cell cycle, we fused the MD or MDmut domains to an mTurquoise fluorescent protein and observed mTurquoise signal reduction by flow cytometry in G1 phase (Supplementary Fig. 5a-c).
2. Page 16, line 4-6: I would change the text to “This characterization is important given that GATA2 is degraded several hours after other mitotically degraded proteins and is likely occurring beyond mitotic exit, potentially interfering with GATA2 function in G1 phase”

We thank the reviewer for acknowledging our efforts to address the all the points raised. Regarding data on GATA2 degradation presented in Figure 4a, we have repeated the western blots with more protein (7,5ug) and used a non-stripped membrane to detect Cyclin B1 for better visualization. Moreover, we have performed cell cycle analysis of HEK 293T cells before (0h) and after (2h, 4h and 6h) nocodazole release to better characterize the cell cycle status of the population of cells after nocodazole release. The new results of the repeated western blot and the cell cycle profile can be found below.

Rebuttal Figure 1. Western blot analysis of GATA2 and cyclin B1 proteins in HEK 293T cells expressing MD-GATA2 or MD_{mut}-GATA2 before (0h) and 2, 4 and 6 hours (h) after release from nocodazole arrest. Actin was included as loading control. Async – Asynchronous cells.

Supplementary Fig. 5a, Cell cycle analysis of asynchronous and nocodazole treated HEK 293T cells before (0h) and 2, 4 and 6 hours (h) after release from nocodazole arrest. PI – propidium iodide.

The repetition of the western blot confirmed the degradation results of MD-GATA2 at 4 hours previously included in the manuscript. Cell cycle analysis of HEK 293 T cells showed that at 4h hours after nocodazole release the cell population is still exiting mitosis with 40% of cells in G2/M, so we cannot say that degradation is only happening exclusively in G1 as suggested. MD-GATA2 degradation might not exactly follow Cyclin B1 degradation profile in this setting due to its larger size or due to overexpression bias. Although the degradation dynamics for GATA2 seems delayed, our collective data (including the flow cytometry analysis present in Supplementary Figure 5) shows that degradation of proteins fused to MD

domain occurs at mitosis-to-G1 transition and is prolonged in G1, as described in our manuscript and according to previous use of the cyclin B1 mitotic degron^{11,14,16}.

We have included the cell cycle profiles in revised supplementary Figure 5a and have modified the highlighted sentences by the reviewer to:

1) “We assessed protein levels of MD- or MD_{mut}-GATA2 constructs before and after release from nocodazole arrest, using degradation of Cyclin B1 as a control for mitotic degradation. While Cyclin B1 was quickly degraded after nocodazole release, we observed induced GATA2 protein degradation over time, with highest impact on MD-GATA2 degradation at 4 hours, increased protein levels after 6 hours and returning to MD_{mut}-GATA2 levels in asynchronous cells (Fig. 4a, Supplementary Fig. 5a). For better resolution of protein degradation during specific phases of cell cycle, we fused the MD or MD_{mut} domains to an mTurquoise fluorescent protein and observed mTurquoise signal reduction by flow cytometry at both mitosis and G1 phases (Supplementary Fig. 5b-d).”

2) “This characterization is important given that GATA2 degradation persists for several hours beyond mitotic exit potentially interfering with GATA2 function in G1.”